

# Mapping long-term evapotranspiration losses in the catchment of the shrinking Lake Poopó

Juan Torres-Batlló[1], Belén Martí-Cardona[1], Ramiro Pillco-Zolá[2]

[1]Department of Civil and Environmental Engineering, University of Surrey, Guildford, UK
    [2]Instituto de Hidráulica e Hidrología, Universidad Mayor de San Andrés, La Paz, Bolivia

**Abstract.**

Lake Poopó is located in the Andean Mountain Range Plateau or Altiplano. A generalised decline in the lake water level has been observed since 2001, coinciding roughly with an intensification of agriculture exploitations such as quinoa crops. Several factors have been blamed for the lake recession, including climate change, increased farming, mining abstractions and
population growth. Being an endorheic catchment, evapotranspiration (ET) losses are expected to be the main water output mechanism. This study used a time series of more than 1000 satellite data based products to map ET and vegetation index trends in the Poopó catchment between 2001 and 2014. The aim was to explore the links between ET, vegetation, land use and the lake recession. The years 2015 and 2016 were excluded of the analysis due to the strong impact of El Niño phenomenon over the study area, which could have masked long term temporal trends related to land use.

We quantified the ET losses and vegetation indices for the main cover types in the Poopó catchment and their temporal trends in the study period. It became obvious that cultivated areas were the ones which had experienced the largest increase in water consumption, although they were not in all instances the land covers with the largest losses. This quantification provides essential information for the sustainable planning of water resources and land uses in the catchment.

We also collected on-site and satellite precipitation data. When integrated over the entire catchment, the overall ET losses
showed a sustained increasing trend at an average rate of 3.2 mm yr$^{-1}$. Rainfall water inputs followed a similar trend, with a slightly higher increasing rate of 5.2 mm yr$^{-1}$. Based on these results and from the point of view of the catchment water balance, the ET loss intensification derived from crop expansion has been compensated by the increase in precipitation. Consequently, this study found no clear link between the agriculture intensification and the Lake Poopó recession in the analysed period.



# 1. Introduction

The sustainable management of fresh water resources faces the challenges of climate change and the need for increased food production to sustain the growing population. These pressures have played a decisive role in the shrinkage of several large endorheic lakes in the world (Wurtsbaugh et al., 2017), such as Lake Urmia in Iran (Eimanifar and Mohebbi, 2005; AghaKouchak et al., 2015), the Aral Sea in Kazahkstan and Uzbekistan (Micklin, 1988; Micklin 2007), Lake Chad in Central Africa (Gao et al., 2011) or the Great Salt Lake in U.S.A. (Wurtsbaugh et al., 2017).

The Lake Poopó, in the Andean Altiplano, is a large, endorheic saline lake, which supports livestock and fishing activities of local population. The lake´s water level has suffered a long-term decreasing trend (Arsen et al., 2014; Cretaux at al., 2011 Satgé et al., 2017). It dried out completely in December 2015 after a strong El Niño event, although flooded again and partially recovered a few months later. Several studies have investigated the Lake Poopó decline (Zola and Bengtsson, 2006; Canedo et al., 2016; Abarca- Del- Rio et al., 2012). These studies pointed at several causes undermining the lake balance: the decreasing flow discharge in the Desaguadero River, which contributes about 60% of the lake´s annual inflow (Zola and Bengtsson, 2006; Molina Carpio et al., 2014); a rapid expansion of the arable land in the catchment (Jacobsen et al., 2011; Satgé et al., 2017); the increased evaporation rates due to climate warming (Abarca- Del-Rio et al., 2012); the intensification of mining activities (Andreucci and Radhuber, 2017). Being an endorheic basin, evapotranspiration (ET) losses are expected to be the main output mechanism for the surface and groundwater storage in Poopó catchment. Earth observation enables the spatial reconstruction of vegetation and ET changes.

Earth observation satellites have imaged the Earth periodically since the 70´s. These images can be processed to derive a wide range of spatial information of paramount relevance for water resources monitoring, such as extent of water bodies (Huang et al., 2018; Marti-Cardona et al., 2010; Marti-Cardona et al., 2013), rainfall (Satgé et al., 2019; Zambrano-Bigiarini et al., 2017; Ayehu et al., 2018) inland waters surface temperature (Marti- Cardona et al., 2019; Liu et al., 2015), vegetation indexes (Duethmann and Blöschl, 2018; Eckert et al., 2015; de Jong et al., 2011), or per-pixel evapotranspiration values (Running et al., 2017; Marti-Cardona et al., 2016; Karimi & Bastiaanssen, 2015; Mo et al., 2017).

Numerous studies in recent years have used satellite data based products to relate vegetation and ET changes to water resources variability (Tan et al., 2018; Zhang et al., 2018; Bouchez et al., 2016; López et al., 2017; Mohebzadeh and Fallah, 2019). For instance, Omute et al., (2012) demonstrated the link between NDVI and water level fluctuations in Lake Victoria. Deus and Gloaguen (2013) highlighted the dependence of Lake Mantra extent to climate fluctuations and the usefulness of remote sensing datasets for water resource management.





This paper explores spatiotemporal trends in vegetation, ET and precipitation within the DP catchment. The aim is to investigate links between land use, ET losses and the Lake Poopó shrinkage. For this purpose, ET and vegetation index trends are estimated per cover type and geographical area.

## 2. Study Area and Period

The Andean Altiplano is the largest plateau on the Earth, after the Tibetan Plateau. The Altiplano has an approximate area of 192,000 km², distributed over the countries as follows: Chile (4%), Peru (26%) and Bolivia (70%) and has an average elevation of 4,000 m. The highest point is Mount Sajama (6,542 m) and the lowest is at the bottom of the Lake Titicaca (3,533 m) (OEA, 1996). It represents one of the poorest regions in South America with a total population of 2.2 million (OEA, 1996). The region includes the TDPS system with a total area of 143,900 km² (Satgé et al., 2015) at an average elevation of 3,810 m.

The TDPS system is an endorheic catchment that comprises a physiographic region of South America located between latitudes and longitudes from 14° S to 21° S and from 71° W to 66° W. The geographical limits of the system are: Cordillera Occidental to the west, the Cordillera Real (separates the catchments of the Amazon) to the east, Lake Titicaca to the north and the Uyuni Salar to the south. This research will focus on the Desaguadero-Poopó system (DP) within the TDPS system, which is the surface catchment contributing runoff to Lake Poopó with an area of approximately 58,000 km². The DP is essentially an

endorheic system, although rarely some overspill discharges towards the Salar the Uyuni can occur. Below Lake Titicaca, the Desaguadero River conveys water from Titicaca to Poopó being the adjacent area of this the lowest point of the DP system with 3,600 m approximately.

Annual precipitation varies with the latitude, ranging from about 750 mm yr$^{-1}$ in the north to 160 mm yr$^{-1}$ in the arid south.

Rainfall is very unevenly distributed over the year with 95% of the annual precipitation falling between October and March, while July and August are the driest months. The average annual temperatures are around 10°C, achieving their minimum during the months of June and July and their peaks in the months of December and January. As it is a plain situated at high altitude it undergoes large thermal amplitudes from -5 °C at night to 25°C at noon. Because of the altitude, the high solar radiation generates intense potential evapotranspiration of 1,000-1,500 mm yr$^{-1}$(OEA, 1996). According to Pillco- Zolá et al.,

(2019), the monthly average relative humidity varies between 52% and 68%. In terms of vegetation, the 65% of the total vegetation area is covered by grassland followed by alpine vegetation (12%), crops (11%), bareland (9%) and wetland (3%). The most common crops in the Altiplano are potatoes and quinoa (Canedo et al., 2016; Jacobsen, 2011). This is the area of the planet with the largest quinoa production.

With the aim to explore catchment changes that may have impacted the Lake Poopó in the last decades, this study analysed the vegetation and evaporation spatiotemporal trends in the DP system for the period 2001-2014. The years 2015 and 2016 were excluded from the analysis due to the occurrence of an extreme El Niño phenomenon, which significantly altered the hydrological long term trends. The trends were analysed considering the entire time series, and also focusing on the wettest





months, January and February, and the driest ones, July and August. The overall and seasonal trends were mapped over the entire DP system, and also examined independently for the main land cover types.

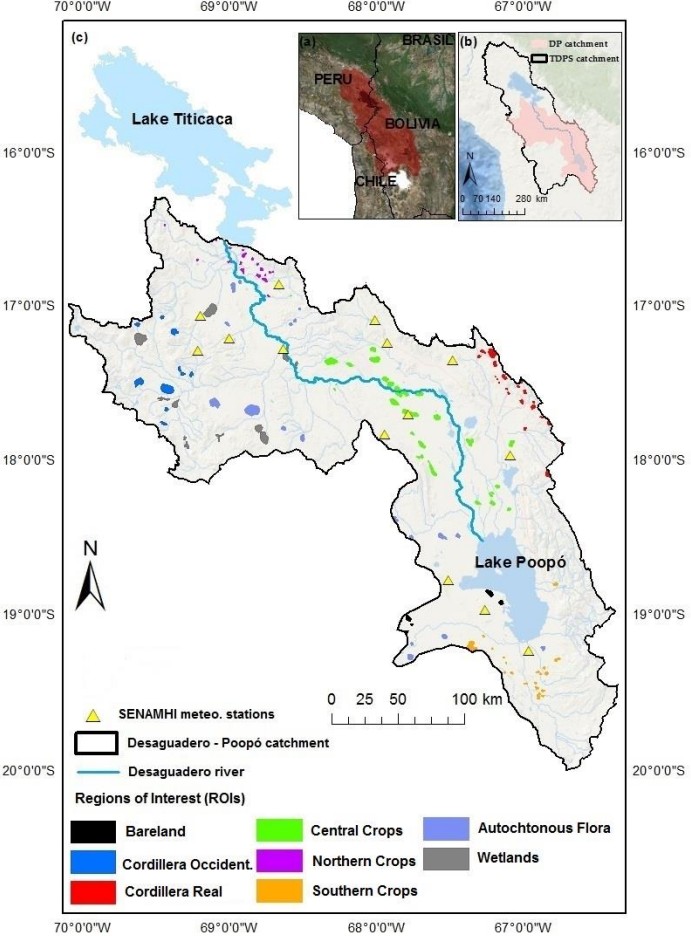

**Figure 1.** Study area with (a) the location of the TDPS system within the South American continent; (b) DP system within the TDPS and (c) DP system, ROIs representative of land cover types, and location of meteorological stations.



## 3. Methodology

### 3.1 Experimental Data

The remote sensing products and on-site records used in this study are listed below. Most of the analysis was carried out using

the Google Earth Engine (GEE) platform (Gorelick et al., 2017), which enabled the effective access and processing of large spatio-temporal data sets.

### 3.1.1 Satellite data

This study used imagery and products derived from the Moderate Resolution Imaging Spectrometer (MODIS). Two identical MODIS instruments are installed on board the Terra and Aqua satellites of the National American Space Agency (NASA), each imaging the entire Earth twice daily.  Optimal quality pixels from consecutive MODIS images are selected and used to produce vegetation index maps at 8- day intervals, made available as products MOD13Q1 and MYD13Q1 (Didan, 2015). For the analysis of vegetation trends in the DP system, this study used the normalized difference vegetation index (NDVI,

Rouse Jr, 1974) in those products. Table 1 provides an overview of the used dataset.

Global evapotranspiration (ET) estimates are also derived from MODIS images combined with ancillary datasets made available at 8-day intervals as product MOD16A2 (Mu et al., 2011; Running et al., 2017). The algorithm for the data collection is based on the logic of Penman-Monteith equation (Penman, 1948; Monteith, 1965) including daily meteorological data along

with MODIS data. The ET estimates account for the evaporation from canopy and soil, and for the transpiration through plant´s stomata. The magnitude of the measurements is in (mm day $^{-1}$).

MODIS surface reflectance (Vermote, 2015) data was acquired for 16 May 2012, at the end of the wet season in a rainy year, to generate a water and flood areas mask for the DP system.


**Table 1.** Datasets used in this study.

| PRODUCT NAME | MOD16A2 | MOD13Q1 | MYD13Q1 | MOD09A1 | SRTM | CHIRPS |
|---|---|---|---|---|---|---|
| **Physical magnitude** | ET | NDVI | NDVI | Reflectance | Terrain elevation | Precipitation |
| **Spatial resolution** | 500 m | 250 m | 250 m | 500 m | 30 m | 0.05 arc degrees |
| **Temporal interval** | 8 days | 8 days | 8 days | 8 days | N/A | Daily |
| **Number of products used:** | | | | 16 - May 2012 | Mosaic of 10 | 5113 |
| **Entire study period** | 643 | 322 | 288 | | granules | |
| **Wet season** | 112 | 56 | 48 | | | |
| **Dry season** | 112 | 56 | 52 | | | |



### 3.1.2 Land cover maps

Two land cover maps of the DP system were used: the *Mapa de Cobertura y Uso Actual de la Tierra en Bolivia* (COBUSO) (UTNIT, 2011) and the GlobeLand30 (Chen et al., 2015). Commissioned by the Bolivian Government, COBUSO is a country-
wide, 30 m spatial resolution land cover map derived from 60 Landsat 5 scenes acquired between 2006 and 2010.  This map includes 21 different land cover classes over the DP system that have been grouped according to their similarity to a total of 10 classes through conventional GIS techniques. Table 2 presents the grouping of classes.

The GlobeLand30 is a global land cover map at 30 m resolution made by the National Geomatics Center of China. The accuracy
in the area of study of GlobeLand30 is lower than the one produced by COBUSO-2010. Zones with irrigated systems were derived from (Canedo et al., 2016) where have been mapped.

### 3.1.3 Digital Elevation Model

A digital elevation model of the study area was obtained from the Shuttle Radar Topography Mission (SRTM) data (Farr et al., 2007). The SRTM is an international project led by the National Geospatial-Intelligence Agency (NGA) and the National Aeronautics and Space Administration (NASA). The mission was performed in February 2000 and the data was acquired using single-pass interferometry. This model was used for delineating the DP watershed boundaries.

### 3.1.4 Precipitation data

Precipitation data were obtained from on-site meteorological stations and from satellite rainfall estimations. Monthly rainfall data were acquired at fourteen meteorological stations of the Servicio Nacional de Meteorología e Hidrología de Bolivia (SENAMHI). Figure 1 shows the location of these stations.
In order to obtain rainfall data from areas not adequately gauged by meteorological stations, data from the "Climate Hazards Group InfraRed Precipitation with Station data (CHIRPS)" was used. CHIRPS is a global rainfall dataset at 0.05 arc degree resolution based on satellite and in-situ precipitation measurements, developed by the U.S. Geological Survey (USGS) and the Climate Hazards Group (Funk et al., 2015).The CHIRPS precipitation data was successfully validated in a peripheral DP region by Satgé et al., 2019; Canedo et al., 2018. Here, monthly accumulated rainfall from the 14 on-site meteorological
stations and from the CHIRPS data was compared. Those CHIRPS pixels coinciding with the location of the meteorological stations were selected and the monthly accumulation was calculated for those, using a time series of 5,111 daily rainfall observations. The comparison between both data sets showed a Spearman's correlation coefficient (CC) of 0.80 with significant results and bias of 0.51 mm month$^{-1}$ (Fig. 2).  Given the agreement between both data sets, the regional analysis of precipitation was carried out using CHIRPS data, which available for the entire DP system.

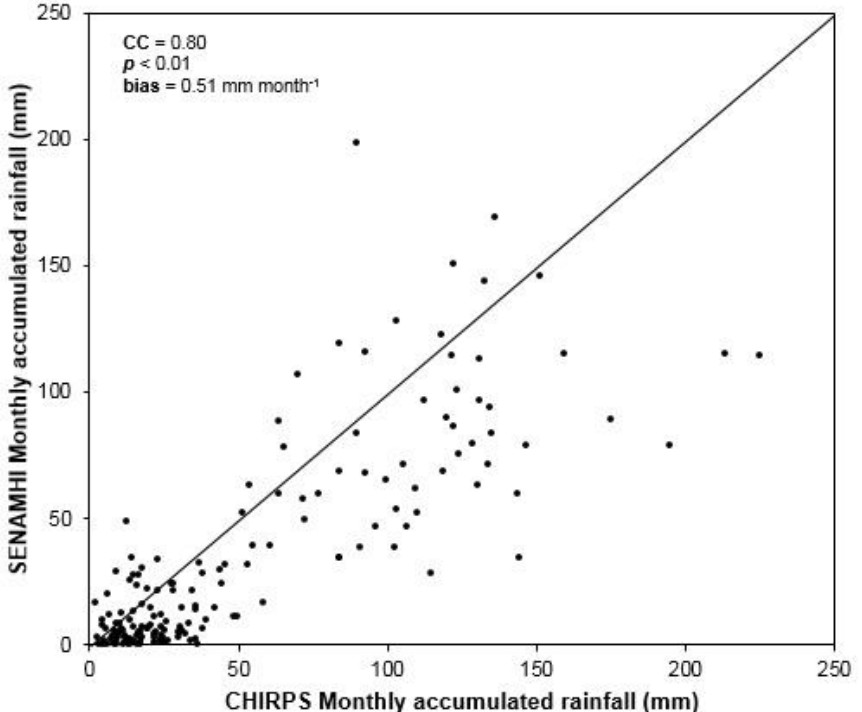

**Figure 2.** Scatter plot of monthly SRE (CHIRPS) versus rain gauges (SENAMHI).

### 3. 2 Mapping vegetation and evapotranspiration dynamics

A MODIS surface reflectance image from 16 May 2012 was used to calculate the Modified Normalized Difference Water Index (MNDWI) (Xu, 2006) over the entire DP system. The MNDWI map was thresholded in order to identify likely waterlogged pixels and produce a water mask. This mask was applied to the MODIS NDVI products prior to any analysis. It was not applied to the ET products since a water mask is already incorporated in their derivation.

10 Per pixel averages of NDVI and ET losses were calculated for the study period, using the entire data set of the corresponding MODIS products, after applying the water mask.

Per pixel ET and NDVI temporal trends were assessed by calculating the Sen´s slope (Sen, 1968) for the entire study period, and for the wet and dry seasons. The Sen´s slope is robust to outlier values (Neeti and Eastman, 2011) and has become a
15 common alternative to the Ordinary Least Squares (OLS) method in the last decades for spatio-temporal NDVI analysis (Fernandes and Leblanc, 2005; Wu et al., 2013). The equation for Sen´s slope is as follows:

$$Q_i = Median\left\{\frac{x_j - x_i}{j - i}\right\}, \text{for } i = 1, \dots, N \qquad (1)$$


where $x_j$ and, $x_i$ were the value at time $i$ and $j$ ($j > i$) respectively and N is the number of time periods representing the totality of the observations per pixel. The median of the slopes is used to characterize the trend. The significance of the time series trends was evaluated by the non-parametric Mann-Kendall test (Mann, 1957; Kendall, 1957) commonly used in the significance analysis of hydro-meteorological data series (Fang et al., 2016) through the assessment of the *p*-value (Fensholt and Proud, 2012).

NDVI and ET average values and trends were analysed together with the digital elevation model to identify physiographical regions exhibiting different temporal patterns within the DP.

## 3.3 Land cover characterisation

In order to identify the NDVI and ET dynamics of the main land covers in the DP system, eight broad cover categories were defined based on the COBUSO-2010 and GlobeLand30 classifications, and on the observed spatial trends of both parameters. These categories are briefly described below.

- Northern Crops corresponds to cropland areas located in the northern part of the system, which were under exploitation long before 2001, the start of the present study period.
- Central Crops represents agriculture areas located in the central valley of the DP system on alluvial soils and rather flat topography following the course of the Desaguadero River.
- Southern Crops are located south of Lake Poopó, in areas where literature reported expansion of quinoa crops in the study period.
- Cordillera Real encompasses croplands and alpine vegetation areas located above 4200 m. Both covers are included in the same category, since their mixture occurs at a scale difficult to separate at the ET product resolution. A high ratio of irrigated systems (Canedo et al., 2016) has been observed in this area.
- Cordillera Occidental includes scattered Alpine vegetation in the arid North West mountain range.
- Autochthonous Flora encompasses Andean puna and Andean grassland. It is the most abundant cover type.
- Wetland areas are located in the northern and north-western part of the system due to the high availability of water. These areas represent flood zones with peculiar ecosystems.
- Bare land groups sand and lacustrine deposits with almost no vegetation. The highest extent of bareland is found in the arid south and southwest of the DP system.

Regions of interest were delineated for each of these categories (Fig. 1) and validated by site visits and by visual inspection of multi-temporal, high resolution Google Earth images. Table 2 indicates the land cover classes converted from the previous





grouping of classes of COBUSO included in the ROIs defined for this study. The ROI´s were used to extract per-class NDVI and ET values and trends from the maps produced according to Section 3.2.

**Table 2.** ROIs resulting from the land cover clusters extracted from COBUSO-2010 and GlobeLand30.

**COBUSO-2010 Land Cover map**

| ROIs | Crops | Wetland | Andean Puna | Andean Grassland | Andean Forests | Sand Deposit | Sault Deposit | Lacustrine Deposit | Alpine Grassland | Scattered Alpine Veg. |
|---|---|---|---|---|---|---|---|---|---|---|
| Central Crops | X | | | | | | | | | |
| Northern Crops | X | | | | | | | | | |
| Southern Crops | X | | | | | | | | | |
| Wetland | | X | | | | | | | | |
| Bareland | | | | | | X | X | X | | |
| Cordillera Occidental | | | | | | | | | X | X |
| Cordillera Real | X | | | | X | | | | X | X |
| Autochtonous Flora | | | X | X | | | | | | |

## 4. Results and discussion

### 4.1 Average vegetation and evapotranspiration maps

NDVI values averaged for the study period (Fig.3a) range between 0.10 and 0.35 for virtually the entire DP system. The greenest area, with NDVI values from 0.25 to 0.35, is located at the northern part of the catchment and corresponds to dense agriculture exploitations and floodable areas.  The smallest vegetation indices are found in the lowest elevation areas of the central and southern plateau, and also in the mountainous northwest.

Average ET losses range from 145 to 550 mm yr$^{-1}$ (Fig.3b). They appear to be closely related to the terrain elevation, increasing more rapidly with elevation over the Cordillera Real.   The maximum ET values are observed in high elevation areas of the Cordillera Real, followed by the northern most part, where NDVI values are highest.  The lowest ET losses occur in the drier south west, with average annual values under 250 mm yr$^{-1}$.





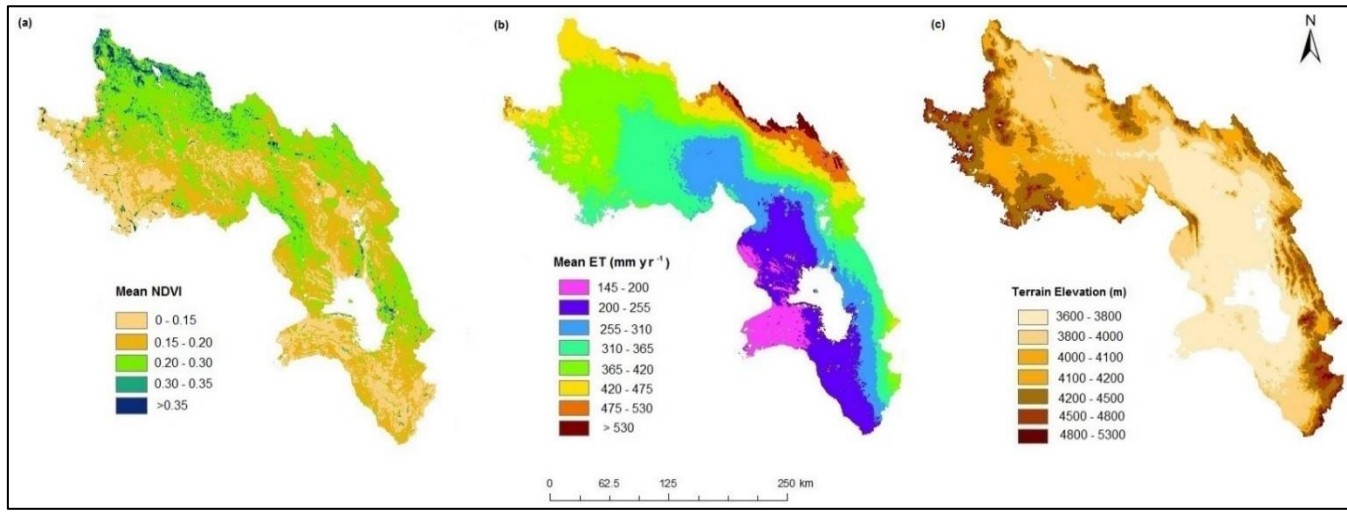

**Figure 3.** Spatial distribution of mean NDVI and ET for the period 2001-2014 and terrain elevation of the DP system. (a) Mean NDVI; (b) Mean ET; (c) Terrain elevation.

## 4.2. Spatio-temporal trends in ET and NDVI

Yearly NDVI and ET averages over the entire DP system show a timid overall increasing trend of about 0.001 yr$^{-1}$ and 4.3 mm yr$^{-1}$ (Fig. 4), respectively, consistent with the average precipitation trend. A spatial analysis of the vegetation temporal

10   trends (Fig. 5) shows that these slow changes occurred over most of the catchment. Two clusters of trends can be identified: one corresponds to areas around Charahuaito (see.Fig.5b), in the southwest part of Aroma province, exhibiting the largest positive NDVI change rate; the second cluster is located at the north end of the system, close to Machaca, and exhibits a negative trend. Both clusters correspond to the areas with the highest mean NDVI (Fig. 3). The spatial distribution of ET temporal trends reveals slow increases over a large central part of the system (Fig. 5), while the clearer negative trends

15   concentrate on the mountainous northwest area. Not clear general correlations between the NDVI and ET trends have been identified, suggesting that the NDVI is not the single main driver for the generalized increase in ET losses.





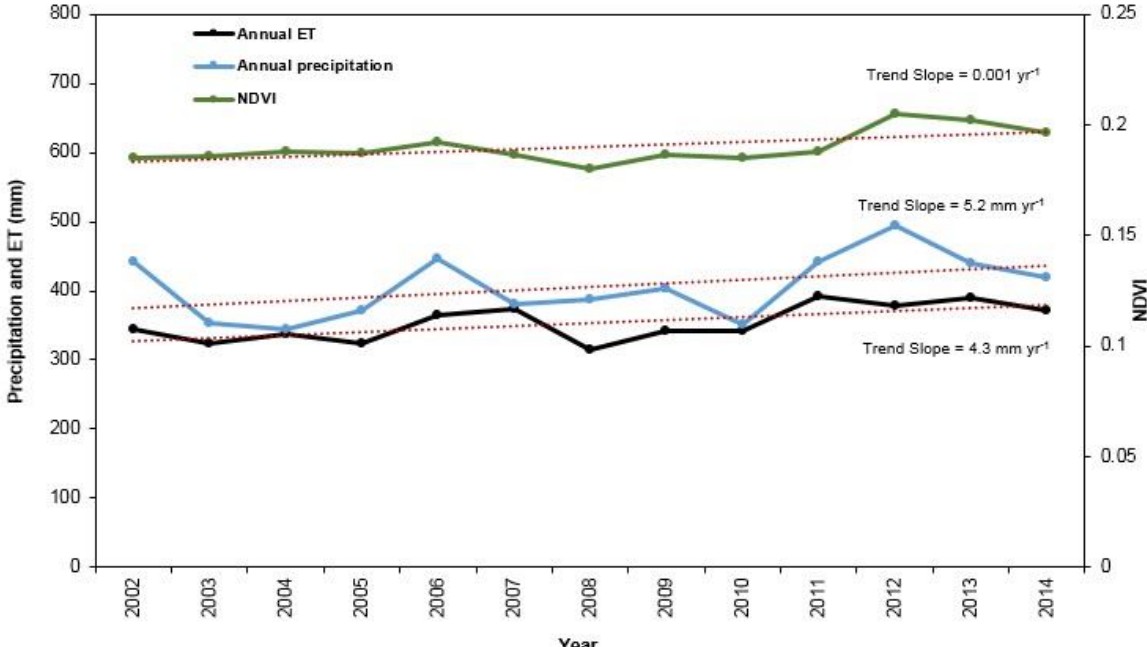

**Figure 4.** Evolution of annual average NDVI, ET and precipitation (CHIRPS) in the DP system.

When temporal trends are analysed by season, accused, statistically significant NDVI and ET increasing trends arise in the wet
5 season, contrasting with almost no changes or slightly negative ones in the dry season (Table 3, Fig. 5). In the wet season,
NDVI increments occur largely over areas of high NDVI. A general spatial consistency between increases in NDVI and ET
can be observed (Fig. 5), indicating that during the wet season, ET increases are more related to vegetation than for the rest of
the year. However, the limited degree of correlation between both trends reinforces the hypothesis that NDVI is not the single
main driver of ET changes in the DP system.

**Table 3.** Regional annual ET mean changes in the DP System from 2001-2014. NDVI mean change for the study period 2001-2014. Number
of significant pixels obtained from the non-parametric Mann Kendall test.

| Period | Mean ET annual change ( mm yr$^{-1}$) | % of significant pixels | Mean NDVI change | % of significant pixels |
|---|---|---|---|---|
| **Wet season** | 7.3 | 68% | 0.045 | 61% |
| **Dry season** | -1.0 | 1% | 0.004 | 55% |

Some differences were observed in the spatial distribution of increasing ET trends compared to the ones reported by Satgé et al., (2017). The latter authors observed statistically significant ET increases in the north of the catchment (see hatched area in Fig. 5). In this research, the highest increases were located in the central and eastern part of the system. It should be noticed that the analysis by Satgé et al., (2017) used 180 monthly MODIS ET version 5 products for the period 2000-2014,

5    while this study used a total of 643 8-day MODIS ET v6 from 2001 to 2014.

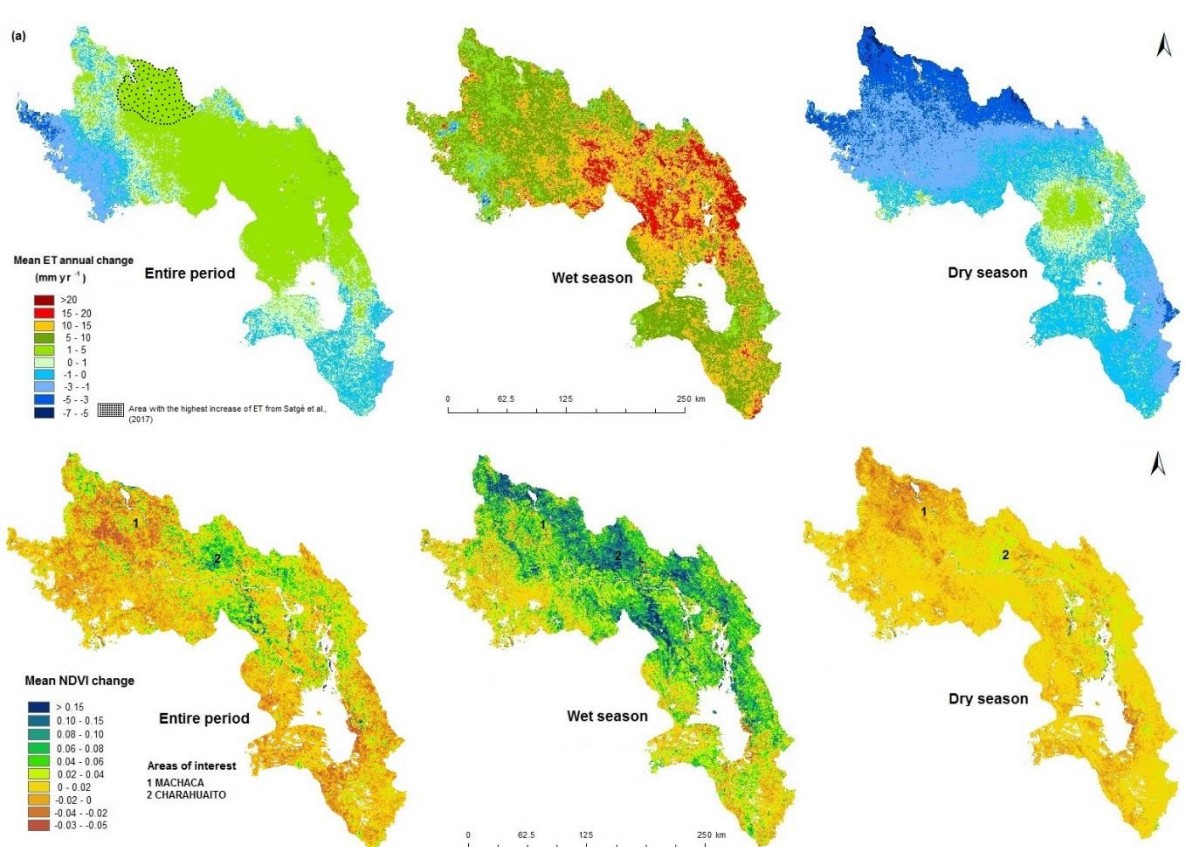

**Figure 5.** NDVI and ET spatio-temporal trends over the DP system for the period 2001-2014: (a) ET annual mean change (mm yr[-1]) and (b) NDVI mean change for the entire study analysis.

### 4.3.1 Per land cover analysis of NDVI and ET trends

Trends were positive in most of the scenarios except for the dry season, where results showed a low negative trend for all the

15   land covers except for Cordillera Real. The land cover with higher increases in most of the analysis was Central crops with 3.3 mm yr[-1], 13.1 mm year[-1] and - 0.3 mm year[1] in annual, wet and dry season respectively. Central crops was the land cover



showing the most significant results with $p = 0.02$ for the entire period and $p < 0.01$ for the wet season. The Cordillera Real was the land cover with greater results after Central Crops with a total of 3.8 mm yr$^{-1}$ and 12.6 mm yr$^{-1}$ for the annual and wet season respectively showing no changes during the dry season with no significance. Southern and Northern Crops experienced similar trends than Central Crops with lower increases for the annual and wet season with negative trends during the dry months. The analysis of the annual period in areas without human activity revealed almost no changes for Bareland and Autochthonous Flora. The trend of these land covers changed slightly during the wet months with some increases around (3.9 - 4.2 mm yr$^{-1}$). During the dry season, Bareland and Autochthonous Flora showed a decreasing trend. Wetland and Cordillera Occidental were the only land cover with an annual decrease trend of -0.3 mm yr$^{-1}$ and - 1.0 mm yr$^{-1}$ respectively.

The positive trends that characterize all those land covers without human activity might be connected to environmental causes such as, for instance, changes in temperature (López-Moreno et al., 2016) , the increase of glacial meltwater runoff due to climate change in the last decades (Cook et al., 2016) or with the exposed rainfall dynamic in previous sections (see.Fig.4). The gap between cropland areas and the rest of land covers could be related to non-environmental factors. The annual period showed high increases in ET located in areas with high annual water consumption such as Cordillera Real, contrasting with the highest decreases of ET located around Cordillera Occidental and Wetlands where annual water consumption remains high.

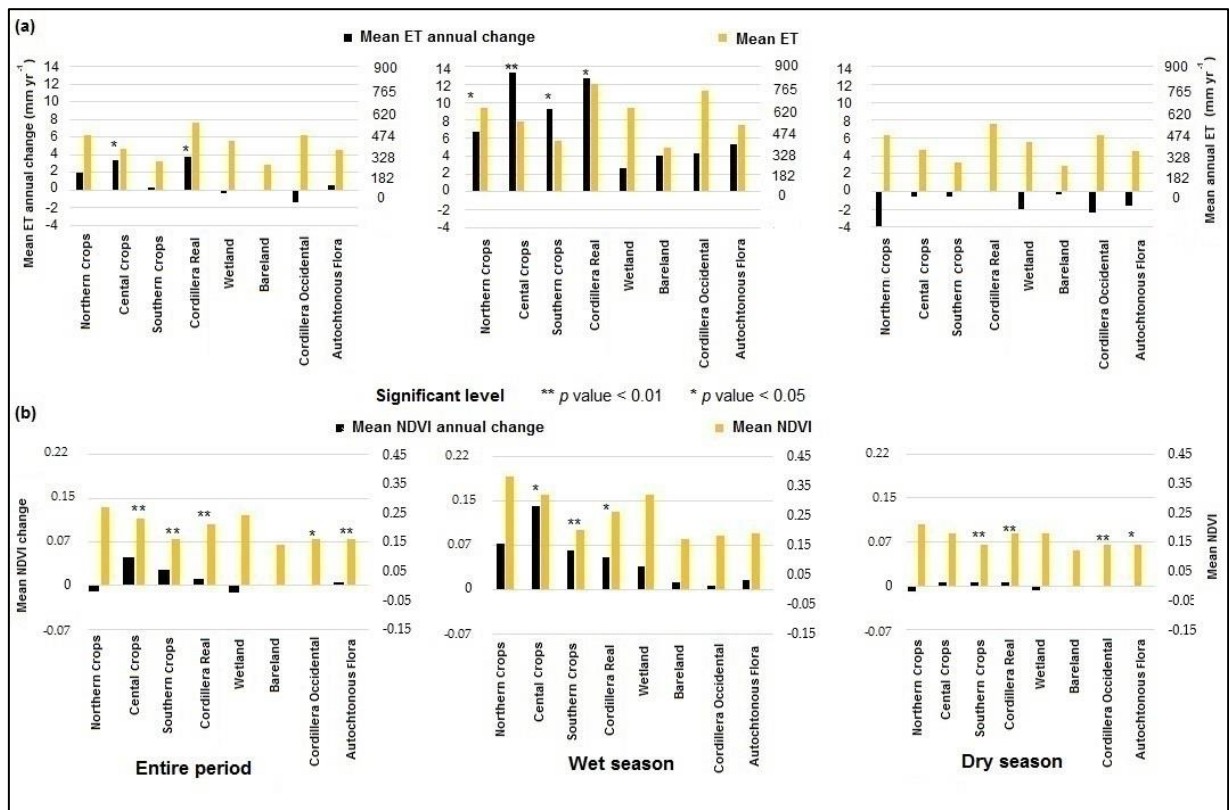

**Figure 6.** ET and NDVI annual land cover trends with a mean of the values corresponding to the entire period and the significance level. (a) Mean ET annual changes (mm yr$^{-1}$) and mean ET (mm yr$^{-1}$) over different land covers and (b) Mean NDVI change and mean NDVI for the same land covers as in ET analysis.





The results of the vegetation land cover analysis revealed interesting and comparable patterns of seasonal vegetation trends depending on the land cover type**.** Over the period 2001-2014 land cover NDVI trends were generally positive across the DP system (see.Fig.6b), especially during the wet season. As in the ET analysis, the land cover presenting higher increases was

Central Crops. Greater increases for Northern Crops and Southern Crops than for Cordillera Real were noticed during the wet season. Wetland was the land cover without human activity presenting higher increases during the wet season followed by Autochthonous flora. Cordillera Occidental and Bareland did not present significant changes in their dynamics for the entire period. This situation could be related to the lack of vegetation in those areas. No remarkable changes were noticed during the dry season in all the land covers. Wetland and Northern Crops were the only land covers showing negative trends during the

entire period. The higher increases in cropland areas, especially during the wet season when vegetation is developing and needs more amount of water, may be related to some changes in the water management of the area such as increase of irrigation or increase of cropland surface.

### 5. Conclusions

With the aim to explore catchment changes that may have impacted the Lake Poopó water balance, this study has undertaken a comprehensive spatiotemporal analysis of changes in vegetation and evapotranspiration losses over the DP system for the period 2001-2014, using more than 1200 satellite products. Temporal trends of these parameters have also been quantified for the main land cover classes in the area.

The analysis of vegetation changes throughout the entire study period shows slight positive and negative trends distributed over the catchment. Two clusters of consistent trends can be identified: one corresponds to areas around Charahuaito and exhibits the largest positive NDVI change rates; the second cluster, close to Machaca, shows decreasing trends. Both clusters correspond to the highest NDVI areas, indicating that the clearest, large-scale vegetation changes occurred in the most

vegetated areas. The spatial distribution of ET temporal trends reveals slow increases over a large central part of the system, while the clearest negative trends concentrate on the mountainous northwest area. The last result agrees with the recent finding by Satge et al., (2017).

The seasonal analysis of temporal trends revealed striking differences between wet and dry seasons. ET and NDVI trends in

the dry season are either slightly negative or approximately null, while the wet season shows accused increasing and significant trends over more than 60% of the catchment area. This increasingly uneven seasonal dynamics are presumably relate to comparable trends in weather and precipitation patterns and are currently under investigation.

ET losses and their trends have been estimated for the main land covers in the DP catchment. Their values indicate that the land covers with higher water consumption are: Cordillera Real, Wetlands, Cordillera Occidental, Northern Crops and Central Crops with average values of 500, 410, 410, 365 and 310 mm yr$^{-1}$ respectively. This quantification of water consumption per cover type provides crucial information for the sustainable planning of agriculture exploitations and water resources use in the
DP system.

Among the analysed land cover classes, only those including crops, i.e. Northern, Central and Southern Crops, plus Cordillera Real, have experienced an increase in NDVI and evaporation losses, while natural covers showed either constant or decreasing trends. The larger increase in vegetation and ET losses over agricultural regions, strongly suggest that cropping practices
exacerbated water losses in these areas.

The NDVI and ET values averaged annually over the DP system increased at a mean rate of 0.001 yr$^{-1}$ and 3.2 mm yr$^{-1}$, which yields a mean NDVI and annual ET increments of 0.14 and 45 mm for the 14-year study period. Water inputs into the system due to precipitation increased at a mean rate of 5.2 mm yr$^{-1}$, exceeding the ET rise rate. On the other hand, no neat spatial
correlations between NDVI and ET trends have been identified, indicating that the NDVI is not the single main driver for the increase in ET losses. These results indicate that, despite the intensification of agriculture in the DP system between 2001 and 2014 increased the ET losses, these cannot be the directly linked to the Lake Poopó shrinkage.

**Acknowledgements** The authors express their gratitude to USGS, the Climate Hazard Group (CHG), and Google Earth Engine for the availability of the valuable MODIS and CHIRPS datasets and to SENAMHI (Bolivia) for bring monthly in situ precipitation data.

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
