# Peer review of "Mapping long-term evapotranspiration losses in the catchment of the shrinking Lake Poopó"

_Hydrology and Earth System Sciences, 2019_

## Referee Comment (RC1) · Anonymous Referee #1 · 15 Jul 2019

Based on the works cited in the introduction of this study, the catchment under investigation represents an ideal case study on the challenges facing sustainable water resources management due to competing water use demands, coupled with climate-related water scarcity. Many such catchments suffer from lack of ground-based observations of key hydrological variables and even where available their spatial distribution is often inadequate to capture the spatial variability of hydrological processes. Thus, efforts by the authors in this study to utilize remote sensing datasets to analyze the spatial-temporal dynamics of hydrological processes is commendable. However, I have a negative recommendation for this paper based on critical issues highlighted below;

i. The title of the paper, "Mapping long-term evapotranspiration losses in the catchment of the shrinking Lake Poopo" is conflicting with the aim of the study based on the con-

[Figure]

clusion of the study which suggests that the aim was to link agricultural intensification to the declining lake levels.

ii. The introduction is poorly written with no coherent link between ideas presented in the paragraphs. The gap based on previous studies is not well articulated. Thus the scientific contribution and aim of the study are not clear. Besides, the introduction is too short, and the literature review is inadequate

iii. Declining lake water levels is a pertinent issue in this catchment, and as highlighted in the short literature review, there are several possible causes. As such, analyzing trends in precipitation, evapotranspiration, and vegetation dynamics without a proper catchment water balance analysis may not reveal the real cause of lake water level decline. It is a bit strange that quantitative analysis related to lake water levels does not suffice, not even in Fig. 4 on trend analysis.

iv. The study analyzes the effects of agricultural intensification on the catchment water balance, yet only a static land cover map is used for over ten years. Based on this, it is not possible to identify areas where land use changed over ten years.

v. Statistical results of the validation of the unified land cover map are not presented.

vi. The choice of a single date image (P7 – L6) to derive a water mask is not justified. Why this particular date? The water, mask should be time-varying to reflect seasonal changes in the extent of water bodies.

vii. Based on the precipitation validation results presented in Fig. 2, a consistent over-estimate by the CHIRPS product is observed. A bias correction scheme is necessary; otherwise the increase in precipitation from the analysis may be due to the biases from the rainfall product used. Analysis such as the double mass curve analysis is also necessary to identify possible errors in datasets or human-induced changes in rainfall run-off relationships. This could act as a good starting point for the catchment water balance analysis.

viii. The Mann-Kendall test is central to this study, yet there is barely any literature in this study on this method, other alternatives, and why this method, in particular, is suited for this study. Also, the scenarios under which the method was applied are not described. Were the analysis done under the assumption of short term persistence or long term persistence? Tabular presentation of the trend analysis results is missing.

Other technical issues include;

i. Use the word general instead of generalized (P1-L12), (P10-L16), the word decisive (P2-L4) is also not appropriate in this context. Use "areas prone to flooding" not "floodable" (P9-L13).

ii. Obvious (P1-L21), timid (P10-L8), accused (p11-L4) are not appropriate words in scientific writing.

iii. Improper usage of articles and prepositions e.g. of instead of from (P1-L18), article the in "the agriculture" (P1-L28), "the Salar the Uyuni" (P3-L15), "the 65%" (P3-L26) among others

iv. Sentence structure such as in P3-L15, P6-L10, P11-L4, P14-L30 makes it difficult to understand the authors' message.

v. Some statements are ambiguous e.g. "The last results agree with the findings . . ..." (P14-L26). Better refer to results in section x or y.

vi. General grammar editing is needed.

vii. Cross-reference using table or figure numbers and not below or above e.g. (P5-L4)

viii. Some acronyms are used without prior definitions e.g. DP (P3-L1), TDPS (P3-L9).

ix. No consistency in the formatting of tables. Table 1, 2 and 3 are all formatted differently.

x. Some figures have very small fonts in the legend e.g. Fig. 5.

xi. Some sections such as 3.3 are misplaced. To maintain flow in the document this section should be in section 3.1.2

---

## Short Comment (SC1) · 17 Jul 2019

Dear Referee #1,

Thank you for your useful comments to the manuscript. We, the authors, believe that the majority of the technical issues raised were actually addressed during the development of the study and have a rigorous justification, but they were poorly explained in the manuscript. These issues include: why a single date image was used to derive a water mask or how the land cover changes in the study period were considered. A thorough validation of the CHIRPS product was also carried out but is not properly reflected in the manuscript.

Other pointed issues stem from the inadequate introduction, which may create some

misunderstanding about the aims of the study. Actually, we never aimed to develop a water balance of the lake at this stage, since we are currently missing important data for that purpose. The analysis presented here focused on the comparison of temporal trends in vegetation, ET and precipitation. These trends are presented spatially distributed and per land cover type. Absolute water storage figures are intentionally avoided. We are now addressing a thorough revision of the Introduction section for a more comprehensive literature review and a clear statement of the study aims and scientific contribution.

We would like to emphasise the large number and appropriateness of the used satellite products, our assurance on the rigour of the analysis undertaken and the significance of some of the results. We do hope to appropriately address all the weaknesses identified by the reviewer within the next days. Our sincere thanks again for raising those.

Sincerely, The authors

---

## Referee Comment (RC2) · Anonymous Referee #2 · 12 Aug 2019

This manuscript uses remotely sensed data products to help evaluate the role of evapotranspiration (ET) in the recession of Lake Poopo. The trends of precipitation, ET, and the normalized difference vegetation index (NDVI) were carefully analyzed, and it was found that ET was not the main contributor for this change. However, there is a lack of further analysis about the causes, while the vegetation trend has been given more attention than necessary.

1) Section 2: Although the manuscript is about evaluating whether ET has caused the shrinking of Lake Poopo, the text about Lake Poopo and its changes is minimal. The authors are strongly suggested to offer more info on this regard (such as time series of lake level or area).

2) Earlier in the manuscript, it was assumed that ET losses have increased due to

increased cultivated crops. However, the analysis of ET changes in Figure 6 focuses on the ET rate for different crop types at given locations. In Figure 1, only representative ROIs of land cover types are indicated. Showing the land cover land use change (LCLUC) during the study period will help to justify the argument that the area of cultivated crops has increased. Otherwise one can argue that the increased ET is primarily due to increased precipitation, instead of LCLUC.

3) Results from Figure 4 suggest that the overall precipitation has increased more than that from the ET. Consequently, it was concluded that there is no clear link between agricultural intensification and the lake recession. Given these, I don't think the NDVI trend analyses (Figure 5 and Figure 6) are very relevant in this study. Rather, readers would likely be more interested in finding out the true driver for the depletion of the Lake.

4) From a water budget perspective, it is expected that the river discharge into the lake should have also increased. Therefore, the authors are suggested to show the observed river discharge, which directly affects the lake size. Has the river discharge increased? If so, why did the lake shrink? If not, how to explain its trend against the increasing trends from precipitation and ET?

5) Since the ET values in the cultivated agriculture areas have increased more than the other areas, it is conceivable that the irrigation withdraw has played an important role in the depletion. Some discussion on this would help to better understand the results from this study.

6) Page 3, Line 9: the acronym TDPS needs to be defined.

7) Page 6, Line 31: The '5,111 daily rainfall observations' doesn't match the number '5,113' in Table 1.

8) Page 6, Lines 32-33: change 'both' to 'the two'

9) Page 7, Lines 6-9: what is the basis of choosing the MODIS reflectance data on

5/16/2012 for the mask? How does the reflectance based mask differ from the ET based mask?

10) Page 12, first paragraph: As the authors pointed out, the results from Stage et al. (2017) and this study are different because different versions of MODIS ET at two temporal resolutions were used. To what degree would the results and conclusions be affected by the uncertainties associated with the MODIS NDVI and ET products? It would be interesting to add another set of results using MODIS monthly ET v6.

11) Page 12, section number '4.3.1' should be '4.3'.

12) Please discuss the errors and uncertainties associated with the datasets employed in this study.

13) Figure 5: It is unclear how the 'ET annual mean change' was calculated for the wet season and the dry season. Were they first calculated for two months and then scaled up to annual?
* * *

---

## Author Comment (AC1) · 3 Sep 2019

Journal: HESS
Title: Mapping long-term evapotranspiration losses in the catchment of the shrinking Lake Poopó.
Author(s): Juan Torres-Batlló et al.
MS No.: hess-2019-187
MS Type: Research article

**Response to Anonymous Referee #1**

**Based on the works cited in the introduction of this study, the catchment under investigation represents an ideal case study on the challenges facing sustainable water resources management due to competing water use demands, coupled with climate related water scarcity. Many such catchments suffer from lack of ground-based observations of key hydrological variables and even where available their spatial distribution is often inadequate to capture the spatial variability of hydrological processes. Thus, efforts by the authors in this study to utilize remote sensing datasets to analyze the spatial-temporal dynamics of hydrological processes is commendable. However, I have a negative recommendation for this paper based on critical issues highlighted below;**

We sincerely thank the anonymous Referee #1 for their careful review of our manuscript. We have substantially revised our manuscript with special attention to the critical issues identified by the Referee and believe that our study aims and results are much better conveyed now.

The Referee's comments are indicated in bold font, followed by our response in normal font. Blue colour shows new sentences or paragraphs as modified in the manuscript. All the answers are referenced (PX-LX) to the new manuscript which is attached at the end of the document.

Additionally, all analyses presented in the manuscript have been re-processed to the period 2002-2014, for consistency. In the original version, some had been undertaken for 2001-2014 and others for 2002-2014. The modification of the analysis period have not introduced significant changes in the results.

**Major comments:**

**i.     The title of the paper, "Mapping long-term evapotranspiration losses in the catchment of the shrinking Lake Poopo" is conflicting with the aim of the study based on the conclusion of the study which suggests that the aim was to link agricultural intensification to the declining lake levels.**

Actually, the study aim was not "to link agricultural intensification to the declining lake levels". Our aim was to assess vegetation, ET and precipitation spatio-temporal trends over the Desaguadero-Poopó (DP) system, in order to identify changes that can help to explain the lake's decline. The study was timely, since it exploited the recently improved NASA's ET product, and recent cloud-based processing capabilities, which enabled an updated and more detailed (data richer) spatio-temporal analysis of those parameters.

We believe that the misunderstanding regarding the study aims was due to a poor statement of our study aims. In order to address this issue, we have profoundly revised the Introduction to the manuscript (please, see answer to comment ii) and clarified the study aims. We have also changed the title to the one below, better aligned to the research aims:

"Mapping evapotranspiration, vegetation and precipitation trends in the catchment of the shrinking Lake Poopó."

**ii.    The introduction is poorly written with no coherent link between ideas presented in the paragraphs. The gap based on previous studies is not well articulated. Thus the scientific**

**contribution and aim of the study are not clear. Besides, the introduction is too short, and the literature review is inadequate**

As mentioned above, we have profoundly revised the Introduction section to provide a comprehensive literature review and to clearly state the current gap, and the novel and timely scientific contribution of the study. Please, see the new Introduction in the revised version of the manuscript (P2-L1).

iii. **Declining lake water levels is a pertinent issue in this catchment, and as highlighted in the short literature review, there are several possible causes. As such, analyzing trends in precipitation, evapotranspiration, and vegetation dynamics without a proper catchment water balance analysis may not reveal the real cause of lake water level decline. It is a bit strange that quantitative analysis related to lake water levels does not suffice, not even in Fig. 4 on trend analysis.**

In fact, we haven't attempted to develop a water balance for the lake or its catchment, since important hydrological data are missing. For example, there is limited data about aquifer level evolution in the study period and on the groundwater exchange with the lake. The time series of flow discharge in the Desaguadero River, the main water input into Lake Poopó, presents significant gaps and uncertainties. Other challenges for the lake water balance are related to the contribution from minor ungauged tributaries and to the evaporation losses. A catchment-wide water balance would also require the quantification of water storage in the form of snow/ice and in the aquifer. We are currently working on the quantification and simulation of some of those water balance terms, but the study presented here focused on one piece of the puzzle, on the detailed temporal analysis of the spatially distributed terms:  ET, precipitation and vegetation. We don't attempt to explain the issues affecting Lake Poopó, but to exploit state-of-the-art spatiotemporal data to improve our quantitative knowledge of one of the factors that has been highlighted by recent literature as a possible driver of the lake's decline: the increased crop ET losses.  Our revised Introduction tries to clarify this goal of our study, humbler than a water balance, as may have seemed.

Following the Referee's comment, we have added the trend of Lake Poopo's volume in Fig. 4 (P14-L17). We believe that these trends provide important information regarding the role of the increased ET losses over the lake's decline, since they reveal that, at the catchment scale, precipitation inputs have increased at the same or even faster rate than the ET losses. This result seems to lessen the responsibility of increased cropping practices and associated ET losses from the Lake Poopó water loss.

[Figure]

**Figure 4.** Evolution and temporal trends of annual ET and precipitation and mean NDVI over the entire DP system, and volume change in Lake Poopó.

**iv.** **The study analyzes the effects of agricultural intensification on the catchment water balance, yet only a static land cover map is used for over ten years. Based on this, it is not possible to identify areas where land use changed over ten years.**

Actually, the study analysed the increase in ET and NDVI over the entire catchment and, additionally, per land cover type. But we did not attempt to quantify agricultural intensification. In order to associate mean ET and NDVI magnitudes and trends to different land cover types and explored their relative trends, we defined a number of polygons characteristic of those types (regions of interest, ROIs). Special care was taken in the definition of the ROIs, so that their land use type didn't change over the study period. Consequently, the observed land-cover ET or NDVI trends are associated to a stationary, land cover use.

We revised Section 3.2 of the manuscript (P10-L6), to clarify this point by adding the paragraph as follows:

"Regions of interest (ROIs) were delineated for each of these categories (Fig. 1) and validated by site visits and by visual inspection of multi-temporal, high resolution Google Earth images. Google Earth Engine Timelapse (Gorelick et al., 2017), was used to ensure that the ROI's land cover types had not changed during the study period. The ROIs were used to extract per-class NDVI and ET annual values and trends as explained in Section 3.3." (P10-L6)

**v.** **Statistical results of the validation of the unified land cover map are not presented.**

Actually, we have realised that the explanation on the unified land cover map is not needed in this manuscript. We used the land cover map for the delineation of the regions of interest, representative of the 8 core land cover types, which were based on the COBUSO-2010 classification only. We have consequently simplified significantly section "Land cover maps", reason why we have merged it with section "Digital elevation model" into the new section "3.1.3 Topography model and land cover map" (P9-L1), included below. We apologise for the unnecessary explanations included in the initial version of the manuscript.

"3.1.3 Topography model and land cover map (P9-L1)

Topographic data for the DP system was obtained from the Shuttle Radar Topography Mission (SRTM) terrain elevation model (Farr et al., 2007). The terrain topography was used to delineate the DP watershed boundaries, to inform the delineation of control regions and to interpret the study results.
The land cover map Mapa de Cobertura y Uso Actual de la Tierra en Bolivia, COBUSO-2010 (UTNIT, 2011), was used to inform the delineation of the control regions as explained in Section 3.2. Commissioned by the Bolivian Government, COBUSO-2010 is a country-wide, 30 m spatial resolution land cover map derived from 60 Landsat 5 scenes acquired between 2006 and 2010. This map contains 21 different land cover types within the DP system."

**vi.** **The choice of a single date image (P7 – L6) to derive a water mask is not justified. Why this particular date? The water, mask should be time-varying to reflect seasonal changes in the extent of water bodies.**

Yes, we totally agree that the time-varying mask would have been more accurate. The reason why we used the 16-May 2012 image to produce the MNDWI mask is that it was acquired at the end of the wettest season in the study period (see Fig. 4). Consequently, we expected this image to capture the water bodies and prone-to-flood areas at a large water extent condition. Additionally, using a static water mask has advantages in terms of simplicity in the processing of such a large data set, and in the use of a constant number of NDVI values for per-pixel time series (ET products are already masked for water presence).

However, we acknowledge that the static water mask may introduce errors in a few pixels close to areas prone to flooding. This point has been acknowledged in the paper as follows (P10-L16):

"The May 2012 image was selected for the derivation of a water mask because it was acquired at the end of an extremely wet season, so it is expected to capture water bodies and prone-to-flood areas at a close-to-peak water extent condition. A constant water mask, as opposed to a time-varying one, was used for simplicity, although it may miss few, sporadic water-logged pixels." (P10-L16)

vii.     **Based on the precipitation validation results presented in Fig. 2, a consistent overestimate by the CHIRPS product is observed. A bias correction scheme is necessary; otherwise the increase in precipitation from the analysis may be due to the biases from the rainfall product used. Analysis such as the double mass curve analysis is also necessary to identify possible errors in datasets or human-induced changes in rainfall run-off relationships. This could act as a good starting point for the catchment water balance analysis.**

The reason why no bias correction was applied is that the study discussion and conclusions rely on the precipitation trends. No absolute precipitation values are compared against ET, only their temporal tendencies. Regardless of their accuracy, CHIRPS and other satellite derived products are consistent with themselves, which implies that their trends are robust.

We are currently working on the exhaustive analysis of precipitation regimes throughout the DP system, for which double mass curve analysis are being undertaken. Hopefully, this will appear in a near-future research output.

In response to the Referee's comment:

In the revised Introduction, we have emphasized the focus on trends of our analysis, as opposite to absolute values.  In Section 3.1.2 Precipitation Data (P8-L1), we have included a more detailed comparison of CHIRPS versus ground-based rainfall measurements, performed for each rainfall station. This comparison shows a strong consistency.  We have clarified under this section that a bias correction is not applied to the CHIRPS data, because the study relies on their trends, as opposite to their absolute retrievals.

"Precipitation data were collected from on-site meteorological stations and from satellite rainfall estimation products. Monthly precipitation data were obtained from twenty-one meteorological stations of the Servicio Nacional de Meteorología e Hidrología de Bolivia (SENAMHI). Figure 1 shows the location of these stations, which is sparse and not well representative of high elevation areas. In order to retrieve measurements from areas poorly gauged by meteorological stations, satellite precipitation estimations from CHIRPS were used. CHIRPS is a global rainfall dataset at 0.05 arc degree resolution, derived from satellite and in-situ precipitation measurements, developed by the U.S. Geological Survey (USGS) and the Climate Hazards Group (Funk et al., 2015).  CHIRPS precipitation data were successfully validated in areas adjacent to the DP system by Satgé et al. (2019) and Canedo et al. (2018). Here, monthly accumulated precipitation from each on-site meteorological station was compared to the one provided by its corresponding CHIRPS pixel. Fig. 2 depicts the scatter plot for the two data sets, with showing a Spearman's correlation coefficient (CC) of 0.87 with significant results and bias of 2.25 mm month$^{-1}$ (Fig. 2).  Given the agreement between the two data sets, the regional analysis of precipitation was carried out using CHIRPS data, which provides full coverage of the DP system. A bias correction was not applied since only the precipitation temporal trends, and not their absolute values, were needed for the purpose of this study. The lack of precipitation ground data for the validation of the CHIRPS product over the highest parts of the DP system is acknowledged as a source of uncertainty, and current efforts are in place to improve their gauging." (P8-L1)

viii.    **The Mann-Kendall test is central to this study, yet there is barely any literature in this study on this method, other alternatives, and why this method, in particular, is suited for this study. Also, the scenarios under which the method was applied are not described. Were the analysis done under the assumption of short term persistence or long term persistence? Tabular presentation of the trend analysis results is missing.**

In Section 3.4 "Retrieval and mapping of vegetation, evapotranspiration and precipitation dynamics" (P10-L21) and 3.4.1 "Calculation of trends and their significance" (P11-L1) we have included detailed explanations and references for the calculation of Sen's slope, Mann-Kendall (MK) test and significance levels, and we have justified the suitability of these methods. The standard MK test was used under the assumption of long term persistence, which has also been indicated in the revised paper. Please, find our revisions to this comment below in blue.

**3.4 Retrieval and mapping of vegetation, evapotranspiration and precipitation dynamics (P10-L22)**

Per pixel annual averages of NDVI, ET losses and precipitation were calculated for the study period, using the entire data set of the corresponding MODIS products. Per pixel ET, NDVI and precipitation temporal trends and their significance were assessed by calculating the Sen´s slope estimator (Sen, 1968) and Mann-Kendall (MK) test (Mann, 1957; Kendall, 1975) as explained in the Section 3.4.1. Both, trend slope and MK test were applied to per-pixel data using the entire study period, and also considering the wet and dry seasons independently. The approach of the study is based on how physical processes change over time. This is the reason why the trend analysis was carried out under the assumption that the datasets follow a long-term persistence.

**3.4.1 Calculation of trends and their significance (P11-L1)**

Tests for the detection of trends in time series can be classified as parametric and non-parametric methods. Parametric tests are applied assuming that the observed data follows a normal distribution while non-parametric trend test only requires the data to be independent (Gocic and Trajkovic, 2013). The Sen´s slope is a non-parametric method, robust to outlier values (Yue et al., 2002; Neeti and Eastman, 2011; Guay et al., 2014) which has become a common alternative to the Ordinary Least Squares (OLS) method in the last decades for spatio-temporal vegetation and hydro-climatological analysis (Yue et al., 2002; Fernandes and Leblanc, 2005; Wu et al., 2013; Gocic and Trajkovic, 2013). The equation for Sen´s slope is as follows:

$$Q = Median\left\{\frac{x_j - x_i}{j - i}\right\}, \text{for } i = 1, \dots, N \tag{1}$$

Where $Q$ is the slope, $x_j$ and $x_i$ are the values at time $j$ and $i$ ($j > i$) respectively and N is the number of time periods representing the totality of the observations per pixel. Each point is compared with all its next data points and the median of the slopes is used to characterize the trend. The significance of the time series trends was evaluated by the non-parametric Mann-Kendall (MK) test (Mann, 1957; Kendall, 1975). The Mann-Kendall is one of the widely used non-parametric test to detect monotonic trends of hydro-meteorological data series (Dinpashoh et al., 2011; Shadmani et al., 2012; Fensholt and Proud, 2012; Fang et al., 2016, Fathian et al., 2016). Due to the study analyses hydrological data which is not normally distributed, the non-parametric methods are well suited for the detection of monotonic trends. The following equations show how to calculate the test statistic $S$ and the test statistic $Z$ from the Mann-Kendall test:

$$S = \sum_{i=1}^{n-1}\sum_{j=i+1}^{n} sgn(x_j - x_i) \tag{2}$$

$$sgn(x_j - x_i) = \begin{cases} -1, & \text{if} \quad x_j - x_i < 0 \\ 0, & \text{if} \quad x - x_i = 0 \\ +1, & \text{if} \quad x - x_i > 0 \end{cases} \tag{3}$$

where $n$ is the length of the dataset, $x_j$ and $x_i$ represent data points in time series $j$ and $i$, respectively ($j > i$). The difference of the magnitude from each pair of values is compared ($x_j$ - $x_i$ ) when ($j > i$) following Eq.(3). *Sgn ($x_j - x_i$)* is the function sign whose value can be -1, 0 or 1. A positive $S$ value means that the trend is increasing while a negative $S$ value means that the trend is decreasing. The variance of $S$ is estimated by the following equation:

$$Var\,(S) = \frac{\left[n(n-1)(2n+5)-\sum_{j-1}^{g} t_j(t_j-1)(2t_j+5)\right]}{18} \tag{4}$$

where $g$ is the number of the tied groups in the data and $t_j$ is the number of data points in each tied group. The significance of the trend can be found calculating $Z$ value which is acquired as follows:

$$Z = \begin{cases} \frac{S-1}{\sqrt{Var(S)}}, S > 0 \\ \quad 0 \;, \quad S = 0 \\ \frac{S+1}{\sqrt{Var(S)}}, S < 0 \end{cases} \tag{5}$$

In the present study, the significant level of $Z$ was assessed at 99%, 95% and 90% interval of confidence for $|Z|$ values higher than 2.58, 1.96 and 1.64 respectively. Goodman (2001) shows in Table 1 "Bayesian Interpretation of P-values" the correspondence between Z-score and the p-value. The Mann-Kendall test was assessed in all the NDVI, ET and precipitation spatio-temporal trend analysis.

**Other technical issues include:**

i.       **Use the word general instead of generalized (P1-L12), (P10-L16), the word decisive (P2-L4) is also not appropriate in this context. Use "areas prone to flooding" not "floodable" (P9-L13).**

We apologise for these wording errors and thank the Referee #1 for pinpointing them. The word generalized has been deleted (P14-L14) and replaced by general in (P1-L12). The adjective "decisive" has been eliminated in the revised Introduction. "Floodable" has been replaced by "areas prone to flooding" (P12-L17).

"A general decline in the lake water level has been observed in the last two decades…" (P1-L12)

"…is located in the northern part of the catchment and contains dense agricultural developments and areas prone to flooding." (P12-L17)

ii.       **Obvious (P1-L21), timid (P10-L8), accused (p11-L4) are not appropriate words in scientific writing.**

The word "obvious" has been replaced by "clear" (**P1-L22**). The words "timid" and "accused" have been deleted from the manuscript.

"It became clear that cultivated areas were the ones which had experienced the largest increase in water consumption…" (P1-L22)

iii.       **Improper usage of articles and prepositions e.g. of instead of from (P1-L18), article the in "the agriculture" (P1-L28), "the Salar the Uyuni" (P3-L15), "the 65%" (P3-L26) among others**

The expressions highlighted by the Referee #1 have been corrected. Additionally, the language use in manuscript has undergone a thorough review.

"The years 2015 and 2016 were excluded from the analysis due to..." (P1-L18)

"...no clear link between vegetation development and ET indicating that..." (P1-L29)

"The DP is an endorheic system with rare overspill discharge events towards Salar de Uyuni" (P5-L3)

"%. In terms of vegetation, 65% of the total vegetation..." (P5-L13)

**iv.      Sentence structure such as in P3-L15, P6-L10, P11-L4, P14-L30 makes it difficult to understand the authors' message.**

The sentence in P3-L15 (old manuscript) has been replaced by:

"The DP is an endorheic system with rare overspill discharge events towards Salar de Uyuni" (P5-L3)

The sentence in P6-L10 from the old manuscript has been deleted.

The sentence in P11-L4 from the old manuscript has been replaced by:

"When considering seasonal temporal trends, the wet season presents statistically significant increases in NDVI, ET and precipitation whereas the dry season shows no changes or slight decreases." (P16-L7)

The sentence in P14-L30 from the old manuscript has been replaced by:

"The seasonal analysis of temporal trends revealed striking differences between wet and dry seasons. The trends of NDVI, ET and precipitation in the dry season are either slightly negative or approximately null, whereas the wet season shows increasing trends which are significant over more than 60% of the catchment area. " (P19-L28)

**v.       Some statements are ambiguous e.g. "The last results agree with the findings ..." (P14-L26). Better refer to results in section x or y.**

We modified the sentence to be more specific. It now reads as:

"The clearest negative trends concentrate on the mountainous northwest area, consistent with the finding of Satge et al., (2017)." (P19-L25)

**vi.      General grammar editing is needed.**

A thorough grammar revision has been undertaken by a language specialist.

**vii.     Cross-reference using table or figure numbers and not below or above e.g. (P5-L4)**

Amended. It is now in **P6-L8**.

"The remote sensing products and on-site records used in this study are listed in Table 1." (P6-L8)

**viii.    Some acronyms are used without prior definitions e.g. DP (P3-L1), TDPS (P3-L9).**

We have carefully revised all acronyms so they are defined at their first use:

"The highest increases in ET were located in the central and northern part of the Desaguadero-Poopó (DP) system..." (P4-L3)

"The region includes the Titicaca-Desaguadero-Poopó-Salares (TDPS) system with a total area of 143,900 km²..." (P4-L30)

"Regions of interest (ROIs) were delineated for each of these categories..." (P10-L6)

"...and Mann-Kendall (MK) test (Mann, 1957; Kendall, 1975) as explained in the Section 3.4.1." (P10-L25)

**ix.      No consistency in the formatting of tables. Table 1, 2 and 3 are all formatted differently.**

We have modified the tables to keep the same formatting throughout the manuscript.

Table 1: P7-L24

Table 2: P17-L1

Table 3: P18-L1

**x.      Some figures have very small fonts in the legend e.g. Fig.5**

All the figures have been modified in order to improve their visualization (see attached new version of the manuscript)

**xi.      Some sections such as 3.3 are misplaced. To maintain flow in the document this section should be in section 3.1.2**

We agree with the Referee's point but, at the same time, we think that incorporating the entire Section 3.3 into section 3.1.2, would mix up the sections purely on "experimental data" with those presenting the methods applied to the data (3.2, etc.).

In order to address the Referee's comment, we have re-arranged the mentioned sections, so they are consecutive. They now read as follows:

**3.1.3 Topography model and land cover cap** (P9-L1)

Topographic data for the DP system was obtained from the Shuttle Radar Topography Mission (SRTM) terrain elevation model (Farr et al., 2007). The terrain topography was used to delineate the DP watershed boundaries, to inform the delineation of control regions and to interpret the study results. The study used the version 3.0 with 30 m of spatial resolution and an absolute vertical accuracy of ~9 m (90 % confidence) or better (Rodriguez et al. 2005).
The land cover map *Mapa de Cobertura y Uso Actual de la Tierra en Bolivia*, COBUSO-2010 (UTNIT, 2011), was used to inform the delineation of the control regions as explained in Section 3.2. Commissioned by the Bolivian Government, COBUSO-2010 is a country-wide, 30 m spatial resolution land cover map derived from 60 Landsat 5 scenes acquired between 2006 and 2010. This map contains 21 different land cover types within the DP system.

**3.2 Definition of control areas for the main land cover types** (P9-L13)

In order to associate the mean ET and NDVI magnitudes and trends to the main land cover types in the DP system and to explore their relative trends, eight broad land cover categories were defined based on the COBUSO-2010 classes. These categories and the COBUSO-2010 classes they correspond to are briefly described below.

- Northern Crops corresponds to cropland areas located in the northern part of the system, which were under exploitation long before 2002, the beginning of the study period. North Crops comprises Multiple Crops extracted from COBUSO-2010.

- Central Crops represents agricultural areas located in the central valley of the DP system on alluvial soils and rather flat topography following the course of the Desaguadero River, grouping Multiple Crops from COBUSO-2010. Besides Multiple Crops, some residual pixels belonging to Central Crops are classified as Semi-arid Grassland and Wetland in COBUSO-2010.

- Southern Crops are located south of Lake Poopó and encompass the COBUSO-2010 land cover of Multiple Crops.

- Cordillera Real groups Multiple Crops, Sub-humid Andean Forest, Scattered Vivacious High-Andean Vegetation and Semi-arid Grassland located above 4200 m. All the covers are included in the same category, since their mixture occurs at a scale difficult to separate at the ET product resolution. An abundance of irrigation systems has been reported in this area by Canedo et al., 2016 together with an expansion of the mining activity (Perreault, 2013).

- Cordillera Occidental includes Scattered Vivacious High-Andean Vegetation and Scattered Puna in Sand areas in the North West mountain range.

- Autochthonous Flora encompasses Semi-arid Grassland, Scattered Puna in Sand areas, Scattered Vivacious High-Andean Vegetation and Sand Deposits in rare occurrences.

- Wetland areas are located in the northern and north-western part of the system due to the high availability of water. These areas represent flood zones with peculiar ecosystems and encompasses Wet Grasslands.

- Bare land groups Sand, Sault and Lacustrine deposits with almost no vegetation. The highest extent of bareland is found in the arid south and southwest of the DP system.

Regions of interest (ROIs) were delineated for each of these categories (Fig. 1) and validated by site visits and by visual inspection of multi-temporal, high resolution Google Earth images.  Google Earth Engine Timelapse (Gorelick et al., 2017), was used to ensure that the ROI's land cover types had not changed during the study period. The ROIs were used to extract per-class NDVI and ET annual values and trends as explained in Section 3.3.

**Mapping evapotranspiration, vegetation and precipitation trends in the catchment of the shrinking Lake Poopó.**

Juan Torres-Batlló[1], Belén Martí-Cardona[1], Ramiro Pillco-Zolá[2]

[1]Department of Civil and Environmental Engineering, University of Surrey, Guildford, UK
[2]Instituto de Hidráulica e Hidrología, Universidad Mayor de San Andrés, La Paz, Bolivia

**Abstract.**

Lake Poopó is located in the Andean Mountain Range Plateau or Altiplano. A general decline in the lake water level has been observed in the last two decades, coinciding roughly with an intensification of agriculture exploitation such as quinoa crops. Several factors have been blamed for the shrinkage in the extent of the lake, including climate change, increased farming, mining abstractions and population growth. Being an endorheic catchment, evapotranspiration (ET) losses are expected to be the main water output mechanism. This study used a time series of more than 5000 satellite data-based products to map ET, vegetation index (NDVI) and precipitation trends in the Poopó catchment between 2002 and 2014. The aim was to explore links between ET, vegetation, precipitation, land use and the lake decline. The years 2015 and 2016 were excluded from the analysis due to the strong impact of El Niño phenomenon over the study area, which could have masked long term temporal trends related to land use.

We quantified the ET and NDVI magnitudes and trends over the lake Poopó catchment and for areas characteristic of the main cover types during the study period. It became clear that cultivated areas were the ones which had experienced the largest increase in water consumption, although they were not in all instances the land covers with the largest losses. This quantification provides essential information for the sustainable planning of water resources and land uses in the catchment.

We also collected on-site and satellite precipitation data. When integrated over the entire catchment, the overall ET losses showed a sustained increasing trend at an average rate of 4.3 mm yr[-1]. Rainfall water inputs followed a similar trend, with a higher increasing rate of 5.2 mm yr[-1]. Based on these results and from the point of view of the catchment water resources, the ET loss intensification derived from crop expansion has been compensated by the increase in precipitation. Furthermore, no clear spatial correlation was found between ET and NDVI temporal trends. Consequently, this study found no clear link between vegetation development and ET indicating that the NDVI is not the single main driver for the increase in ET losses in the Poopó catchment during the period analysed.

**1. Introduction**

Endorheic or terminal catchments, i.e. those with no surface drainage to rivers or oceans, constitute 25% of the world's continental area (excluding Greenland and Antarctica) and are mostly located in semi-arid and arid climates (Hostetler, 1995). Given the lack of outlet drainage, the surface water storage in endorheic catchments depends critically on the equilibrium among precipitation, evapotranspiration (ET) and groundwater exchange (Hammer, 1986; McCarthy et al., 2001; Wang et al., 2018). This equilibrium is affected, for example, by increases in agricultural land and irrigation, which intensifies ET water losses. ET and evaporation from open water surfaces are also exacerbated by warming temperatures. Climate change combined with human activity has caused severe perturbances in the water balance of several endorheic basins around the world, leading to the shrinkage of terminal lakes in these basins (Wurtsbaugh et al., 2017; Wang et al., 2018), such as Lake Urmia in Iran (Eimanifar and Mohebbi, 2005; AghaKouchak et al., 2015), the Aral Sea in Kazahkstan and Uzbekistan (Micklin, 1988; Micklin, 2007), Lake Chad in Central Africa (Gao et al., 2011) and the Great Salt Lake in the USA (Wurtsbaugh et al., 2017).

The Altiplano, in the Andean Mountain Range, is an endorheic, semi-arid catchment and the second largest mountain plateau in the world. Several studies have highlighted the vulnerability of the Altiplano's water resources to climate change. For example, López-Moreno et al. (2015) and Hunziker et al., (2018) documented temperature increases in the Altiplano in the order of 0.20°C decade$^{-1}$ for the periods 1965-2012 and 1981-2010, respectively. Hoffmann and Requena (2012) estimated the impact of long-term temperature increments on water resources in the northern part of the Altiplano and concluded that it would lead to a dramatic reduction of water in lakes, rivers, glaciers and wetlands, especially during the dry season. Using time-series of satellite images, Cook et al. (2016) revealed a 43% reduction of the total glacial cover in the Bolivian Andean Mountains in the period 1986-2014, which has given rise to new and potentially dangerous proglacial lakes. Vuille et al. (2008) projected a decrease in water availability due to the retreat of Tropical Andean glaciers and the consequent decrease of their reservoir effect. Using global and regional climate models, Thibeault et al. (2010), Seth et al., (2010) and Urrutia and Vuille (2009) predicted changes in precipitation patterns that would lead to the intensification of floods and droughts in the area. In addition to climate change, anthropogenic activities have increased the pressure on the Altiplano's water resources and are likely to exacerbate it further in the future. For example, annual quinoa yield, for which the Altiplano is a major producer, escalated from 28,500 to 75,500 tons in Bolivia between 2008 and 2014 (INE, 2019). Satgé et al. (2017) showed increasing ET rates over cropland in the Altiplano between 2000 and 2014, suggesting its connection to the agriculture intensification. According to Buytaert. and De Bièvre (2012) the projected population growth in the tropical Andes will increase water demand by up to 50% in 2050. Based on population projections and glacio-hydrological simulations, Kinouchi et al., (2019) revealed several scenarios of increase in water demand in Bolivia from 2011 to 2039 of magnitudes ranging from 15% to 53%. Mining,

which is one of the main economic activities in the Altiplano, has increased significantly during the last decade leading to the release of heavy metals and other pollutants into the water bodies thus impairing soil productivity, local fauna and human water use (Perreault, 2013; Andreucci and Radhuber, 2017).

5 Lake Poopó, at the South East of the Andean Altiplano, is a shallow saline lake which supports livestock and fishing activities of local communities (Zola and Bengtsson, 2006; Zola and Bengtsson, 2007; Satgé et al., 2017). More than half of Lake Poopo's annual input comes from the Desaguadero River, which originates from Lake Titicaca's surcharge. Apart from rare overspills towards the Laca Jahuira River, at the south end of the lake (Revollo, 2001; Zola and Bengtsson, 2007), the major water loss mechanism of Lake Poopó is evaporation. The Desaguadero River and Poopó's catchments undergo pronounced

10 wet and dry seasons, causing a rapid recharge of Lake Poopó during the wet season, and a slower decline of water levels as the evaporation exceeds the inflows in the dry season (Zola and Bengtsson, 2006). The annual maximum and minimum lake extents strongly depends on the annual precipitation, as shown by Satgé, 2017. As a result of this dependency, Lake Poopó dried completely on several occasions in the past after severe dry episodes, e.g. in the early 1940´s, 1970s, mid-1990´s and late 2015 (Zola and Bengtsson, 2006).

Apart from the above mentioned specific extreme dry periods, some authors have claimed a declining trend in Lake Poopó water storage in the last two decades (Arsen et al., 2014; Satgé et al., 2017). The latter author highlighted the clear decoupling of the lake extent from the annual precipitation in 2013 and 2014. Several studies have related the lake's decline to the abovementioned pressures over the Altiplano water resources: rapid expansion of arable land in the catchment and an

20 associated increase of ET losses (Satgé et al., 2017); the decreasing flow discharge in the Desaguadero River, presumably linked to irrigation uses (Zola and Bengtsson, 2006; Abarca- Del- Rio et al., 2012; Molina Carpio et al., 2014), and the increased evaporation due to climate warming (Abarca- Del- Rio et al., 2012; López-Moreno et al., 2015).

The lack of substantial ground-based hydro-meteorological observations and the large extent of the study area lead to the use

25 of remote sensing data techniques to map and monitor the Altiplano water resources trends. Earth observation satellites have imaged the Earth periodically since the 1970s. These images can be processed to derive a wide range of spatial information of paramount relevance for water resource monitoring, such as the extent of water bodies (Campbell and Wynne, 2011; Alsdorf et al., 2007; Marti-Cardona et al., 2010; Marti- Cardona et al., 2013; Huang et al., 2018), the spatial distribution of rainfall (Kidd, 2001; Espinoza et al., 2009; Bookhagen and Burbank, 2010), inland water surface temperature (Schneider and Hook,

30 2010; Crosman and Horel, 2009; Liu et al., 2015; Marti-Cardona et al., 2019), vegetation indexes (Chen et al., 2006; Xie et al., 2008; Hermann et al., 2005; de Jong et al., 2011; Eckert et al., 2015; Duethmann and Blöschl, 2018) and ET dynamics (Courault et al., 2005; Li et al., 2009; Karimi and Bastiaanssen, 2015; Marti- Cardona et al., 2016; Mo et al., 2017; Running et al., 2017) among others.

Satgé et al. (2017) used monthly satellite ET products to conduct a spatio-temporal assessment of ET losses in the Altiplano for the period 2000-2014 and pointed at the ET increase from cropland as a key driver of Lake Poopó's decline. The highest increases in ET were located in the central and northern part of the Desaguadero-Poopó (DP) system, despite the maximum cropland expansion being reported at the south and southwest of Lake Poopó by Jacobsen et al. (2011) and Ayaviri and Vallejos (2014). So far, there is no integrated annual and seasonal assessment of precipitation, ET and vegetation tendencies for the study area.

The study presented here undertook for the first time an integrated assessment of vegetation, ET losses and precipitation spatio-temporal trends over the DP system. The study exploits the improved ET product from NASA's Moderate Resolution Imaging Spectroradiometer (MODIS, MOD16A2.006 Global Terrestrial Evapotranspiration 8-Day Global 500m; Running et al., 2017) together with MODIS Terra and Aqua vegetation index product (MOD13Q1.006 and MYD13Q1.006 16-Day Global 250m; Didan, 2015) and Climate Hazards Group InfraRed Precipitation with Station Data (CHIRPS ;Funk et al., 2014) precipitation product. The processing was conducted using the new cloud-based Google Earth Engine, which enabled the integration of the complete time series of daily and 8-day products for the period of analysis, involving over 5000 raster data sets. Temporal trends of the vegetation amount, ET and precipitation were mapped for the entire study period, which encompassed the sharp increase in quinoa yield. The trends for the wet and dry seasons were also analysed independently for the first time, as these were as these were expected to intensify by Thibeault et al. (2010), Seth et al., (2010) and Urrutia and Vuille (2009). An additional novelty of this study is the characterization of ET and vegetation amount trends and absolute values for the broad land uses and geographical areas in the Altiplano. Although absolute values must be taken with caution due to their limited validation over the study area, their temporal trends are robust, and they represent the relative water consumption among different land covers. This information is of paramount relevance for sustainable land use planning and management of water resources in the Altiplano. It is worth noting that, although final figures of precipitation inputs and ET outputs are provided for the entire system, a water balance was not attempted and the main conclusions are drawn from the comparison of the reliable temporal trends of those parameters.

**2. Study area and period**

The Andean Altiplano is the largest plateau on the Earth, after the Tibetan Plateau. The Altiplano has an approximate area of 192,000 km², distributed over the countries as follows: Chile (4%), Peru (26%) and Bolivia (70%) and has an average elevation of 4,000 m. The highest point is Mount Sajama (6,542 m) and the lowest is at the bottom of Lake Titicaca (3,533 m) (OEA, 1996). It represents one of the poorest regions in South America with a total population of 2.2 million (OEA, 1996). The region includes the Titicaca-Desaguadero-Poopó-Salares (TDPS) system with a total area of 143,900 km² (Satgé et al., 2015) at an average elevation of 3,810 m. The TDPS system is an endorheic catchment that comprises a physiographic region of South America located between latitudes and longitudes from 14° S to 21° S and from 71° W to 66° W. The geographical limits of the system are: the Cordillera Occidental to the west, the Cordillera Real (separates the

catchments of the Amazon) to the east, Lake Titicaca to the north and Salar de Uyuni to the south. This research will focus on the DP system within the TDPS system, which is the surface catchment contributing runoff to Lake Poopó with an area of approximately 58,000 km$^2$. The DP is an endorheic system with rare overspill discharge events towards Salar de Uyuni (Revollo, 2001; Zola and Bengtsson, 2007). Below Lake Titicaca, the Desaguadero River conveys water from Titicaca to

5    Poopó, which is the adjacent area to this, the lowest point of the DP system, at an altitude of approximately 3,600 m.

Annual precipitation varies with the latitude, ranging from about 750 mm yr$^{-1}$ in the north to 160 mm yr$^{-1}$ in the arid south. Rainfall is very unevenly distributed over the year with 95% of the annual precipitation falling between October and March, while July and August are the driest months. The average annual temperatures are around 10°C, achieving their minimum

10    during the months of June and July and their peaks in the months of December and January. As it is a plain situated at high altitude it undergoes large thermal amplitudes from -5 °C at night to 25°C at noon. Because of the altitude, the high solar radiation generates intense potential evapotranspiration of 1,000-1,500 mm yr$^{-1}$(OEA, 1996). According to Pillco- Zolá et al., (2019), the monthly average relative humidity varies between 52% and 68%. In terms of vegetation, 65% of the total vegetation area is covered by grassland followed by alpine vegetation (12%), crops (11%), bareland (9%) and wetland (3%). The most

15    common crops in the Altiplano are potatoes and quinoa (Jacobsen, 2011; Canedo et al., 2016) being the largest quinoa production area in the planet.

This study analysed the vegetation, evapotranspiration and precipitation spatio-temporal trends in the DP system for the period 2002-2014. The years 2015 and 2016 were excluded from the analysis due to the occurrence of an extreme El Niño

20    phenomenon, which significantly altered the hydrological trends. The trends were analysed considering the entire time series, and also focusing on the wettest months, January and February, and the driest ones, July and August. The overall and seasonal trends were mapped over the entire DP system, and also examined independently for the main land cover types.

[Figure]

**Figure 1.** Study area with (a) the location of the TDPS system within the South American continent; (b) DP system within the TDPS and (c) DP system, ROIs representative of land cover types, and location of meteorological stations.

**3. Methodology**

**3.1 Experimental data**

The remote sensing products and on-site records used in this study are listed in Table 1. Most of the analysis were carried out using the Google Earth Engine (GEE) platform (Gorelick et al., 2017), which enabled the effective access and processing of large spatio-temporal data sets.

**3.1.1 Satellite data**

Two identical MODIS instruments are installed on board the Terra and Aqua satellites of the National American Space Agency (NASA), each imaging the entire Earth twice daily. Optimal quality pixels (i.e. with no sub-pixel cloud, low aerosol nor sensor view angle < 30 degrees) from consecutive MODIS images are selected and used to produce vegetation index maps at 8- day intervals, made available from products MOD13Q1 and MYD13Q1 (Didan, 2015a, b). For the analysis of vegetation trends in the DP system, this study used the Normalized Difference Vegetation Index (NDVI, Rouse Jr, 1974). The accuracy of the MOD13Q1 and MYD13Q1 NDVI values for clear pixels is reported as ± 0.025 (Gao et al., 2003).

Global evapotranspiration (ET) estimates are also derived from MODIS images combined with ancillary datasets made available at 8-day intervals as product MOD16A2 (Mu et al., 2011; Running et al., 2017). These ET estimates account for the actual (as opposed to potential) evaporation from canopy and soil, and for the real transpiration through plant´s stomata, and are provided in mm day$^{-1}$. This study used the ET product MOD16A2 version 6, which improves the previous version 5 in spatial resolution, from 1000 m to 500 m, and in the accuracy of near real-time surface weather data. It has been validated for 232 watersheds and 46 eddy flux towers (Running et al., 2017), and its accuracy has been assessed with an average mean absolute error of about 24 %, within the 10-30% range of accuracy of ET observations (Running et al., 2017; Jian et al., 2004; Kalma et al., 2008). Regarding the ET and NDVI products' accuracy, it is important to note that they undergo strict, frequent calibrations to ensure consistent measurements, and hence provide robust, reliable temporal trends.

MODIS surface reflectance (Vermote, 2015) data was acquired for 16 May 2012, at the end of the wettest season in the study period, to generate a water mask. Table 1 provides an overview of the satellite data used.

**Table 1.** Datasets used in this study.

| | DATASETS | | | | | |
|---|---|---|---|---|---|---|
| PRODUCT NAME | MOD16A2 | MOD13Q1 | MYD13Q1 | MOD09A1 | SRTM | CHIRPS |
| Physical magnitude | ET | NDVI | NDVI | Reflectance | Terrain elevation | Precipitation |
| Spatial resolution | 500 m | 250 m | 250 m | 500 m | 30 m | 0.05 arc degrees |
| Temporal interval | 8 days | 16 days | 16 days | 8 days | N/A | Daily |
| Number of products used | | | | 16-May 2012 | Mosaic of 10 granules | |
| Entire study period | 599 | 299 | 288 | | | 4748 |
| Wet season | 105 | 52 | 48 | | | 770 |
| Dry season | 104 | 52 | 52 | | | 806 |

**3.1.2 Precipitation data**

Precipitation data were collected from on-site meteorological stations and from satellite rainfall estimation products. Monthly precipitation data were obtained from twenty-one meteorological stations of the Servicio Nacional de Meteorología e
5  Hidrología de Bolivia (SENAMHI). Figure 1 shows the location of these stations, which is sparse and not well representative of high elevation areas. In order to retrieve measurements from areas poorly gauged by meteorological stations, satellite precipitation estimations from CHIRPS were used. CHIRPS is a global rainfall dataset at 0.05 arc degree resolution, derived from satellite and in-situ precipitation measurements, developed by the U.S. Geological Survey (USGS) and the Climate Hazards Group (Funk et al., 2015). CHIRPS precipitation data were successfully validated in areas adjacent to the DP system
10  by Satgé et al. (2019) and Canedo et al. (2018). Here, monthly accumulated precipitation from each on-site meteorological station was compared to the one provided by its corresponding CHIRPS pixel. Fig. 2 depicts the scatter plot for the two data sets, with showing a Spearman's correlation coefficient (CC) of 0.87 with significant results and bias of 2.25 mm month$^{-1}$ (Fig. 2). Given the agreement between the two data sets, the regional analysis of precipitation was carried out using CHIRPS data, which provides full coverage of the DP system. A bias correction was not applied since only the precipitation temporal
15  trends, and not their absolute values, were needed for the purpose of this study. The lack of precipitation ground data for the validation of the CHIRPS product over the highest parts of the DP system is acknowledged as a source of uncertainty, and current efforts are in place to improve their gauging.

[Figure]

**Figure 2.** Scatter plot of monthly satellite rainfall estimations (CHIRPS) versus rain gauges (SENAMHI).

**3.1.3 Topography model and land cover cap**

Topographic data for the DP system was obtained from the Shuttle Radar Topography Mission (SRTM) terrain elevation model (Farr et al., 2007). The terrain topography was used to delineate the DP watershed boundaries, to inform the delineation of control regions and to interpret the study results. The study used the version 3.0 with 30 m of spatial resolution and an absolute vertical accuracy of ~9 m (90 % confidence) or better (Rodriguez et al. 2005).

The land cover map *Mapa de Cobertura y Uso Actual de la Tierra en Bolivia*, COBUSO-2010 (UTNIT, 2011), was used to inform the delineation of the control regions as explained in Section 3.2. Commissioned by the Bolivian Government, COBUSO-2010 is a country-wide, 30 m spatial resolution land cover map derived from 60 Landsat 5 scenes acquired between 2006 and 2010. This map contains 21 different land cover types within the DP system.

**3.2 Definition of control areas for the main land cover types**

In order to associate the mean ET and NDVI magnitudes and trends to the main land cover types in the DP system and to explore their relative trends, eight broad land cover categories were defined based on the COBUSO-2010 classes. These categories and the COBUSO-2010 classes they correspond to are briefly described below.

- Northern Crops corresponds to cropland areas located in the northern part of the system, which were under exploitation long before 2002, the beginning of the study period. Northern Crops comprises Multiple Crops extracted from COBUSO-2010.
- Central Crops represents agricultural areas located in the central valley of the DP system on alluvial soils and rather flat topography following the course of the Desaguadero River, grouping Multiple Crops from COBUSO-2010. Besides Multiple Crops, some residual pixels belonging to Central Crops are classified as Semi-arid Grassland and Wetland in COBUSO-2010.
- Southern Crops are located south of Lake Poopó and encompass the COBUSO-2010 land cover of Multiple Crops.
- Cordillera Real groups Multiple Crops, Sub-humid Andean Forest, Scattered Vivacious High-Andean Vegetation and Semi-arid Grassland located above 4200 m. All the covers are included in the same category, since their mixture occurs at a scale difficult to separate at the ET product resolution. An abundance of irrigation systems has been reported in this area by Canedo et al., 2016 together with an expansion of the mining activity (Perreault, 2013).
- Cordillera Occidental includes Scattered Vivacious High-Andean Vegetation and Scattered Puna in Sand areas in the North West mountain range.
- Autochthonous Flora encompasses Semi-arid Grassland, Scattered Puna in Sand areas, Scattered Vivacious High-Andean Vegetation and Sand Deposits in rare occurrences.

- Wetland areas are located in the northern and north-western part of the system due to the high availability of water. These areas represent flood zones with peculiar ecosystems and encompasses Wet Grasslands.
- Bare land groups Sand, Sault and Lacustrine deposits with almost no vegetation. The highest extent of bareland is found in the arid south and southwest of the DP system.

Regions of interest (ROIs) were delineated for each of these categories (Fig. 1) and validated by site visits and by visual inspection of multi-temporal, high resolution Google Earth images. Google Earth Engine Timelapse (Gorelick et al., 2017), was used to ensure that the ROI's land cover types had not changed during the study period. The ROIs were used to extract per-class NDVI and ET annual values and trends as explained in Section 3.3.

**3.3 Masking water pixels**

A MODIS surface reflectance image from 16 May 2012 was used to calculate the Modified Normalized Difference Water Index (MNDWI) (Xu, 2006) over the entire DP system. A safe threshold of -0.3 was applied in order to identify likely

15  waterlogged pixels and produce the MNDWI water mask. This mask was applied to the MODIS NDVI products prior to any analysis. It was not applied to the ET products since a water mask is already incorporated in their derivation. The May 2012 image was selected for the derivation of a water mask because it was acquired at the end of an extremely wet season, so it is expected to capture water bodies and prone-to-flood areas at a close-to-peak water extent condition. A constant water mask, as opposed to a time-varying one, was used for simplicity, although it may miss few, sporadic water-logged pixels.

**3.4 Retrieval and mapping of vegetation, evapotranspiration and precipitation dynamics**

Per pixel annual averages of NDVI, ET losses and precipitation were calculated for the study period, using the entire data set of the corresponding MODIS products. Per pixel ET, NDVI and precipitation temporal trends and their significance were

25  assessed by calculating the Sen´s slope estimator (Sen, 1968) and Mann-Kendall (MK) test (Mann, 1957; Kendall, 1975) as explained in the Section 3.4.1. Both, trend slope and MK test were applied to per-pixel data using the entire study period, and also considering the wet and dry seasons independently. The approach of the study is based on how physical processes change over time. This is the reason why the trend analysis was carried out under the assumption that the datasets follow a long-term persistence viewpoint.

NDVI, ET and precipitation average values and trends were mapped over the entire DP system and analysed together with the digital elevation model to identify physiographical regions exhibiting different temporal patterns. Average values and trends of NDVI, ET and precipitation were averaged within the ROIs representative of the main land cover types, to identified differential behaviours among them.

**3.4.1 Calculation of trends and their significance**

Tests for the detection of trends in time series can be classified as parametric and non-parametric methods. Non-parametric tests do not presume the data distribution, only require them to be statistically independent (Gocic and Trajkovic, 2013). The Sen´s slope is a non-parametric method to measure data trends, robust to outlier values (Yue et al., 2002; Neeti and Eastman, 2011; Guay et al., 2014) which has become a common alternative to the Ordinary Least Squares (OLS) method in the last decades for spatio-temporal vegetation and hydro-climatological analysis (Yue et al., 2002; Fernandes and Leblanc, 2005; Wu et al., 2013; Gocic and Trajkovic, 2013). The Sen´s slope is calculated as follows:

$$Q = Median\left\{\frac{x_j - x_i}{j - i}\right\}, \text{for } i = 1, \dots, N \tag{1}$$

Where $Q$ is the slope, $x_j$ and $x_i$ are the values at time $j$ and $i$ ($j > i$) respectively and N is the number of time periods representing the totality of the observations per pixel. Each point is compared with all the next data points and the median of the slopes is used to characterize the trend. The significance of the time series trends was evaluated by the non-parametric MK test (Mann, 1957; Kendall, 1975). The MK is one of the widely used non-parametric test to detect monotonic trends of hydro-meteorological data series (Dinpashoh et al., 2011; Shadmani et al., 2012; Fensholt and Proud, 2012; Fang et al., 2016, Fathian et al., 2016). Due to the study analyses hydrological data which is not normally distributed, the non-parametric methods are well suited for the detection of monotonic trends. The following equations show how to calculate the test statistic $S$ and the test statistic $Z$ from the MK test:

$$S = \sum_{i=1}^{n-1}\sum_{j=i+1}^{n} sgn(x_j - x_i) \tag{2}$$

$$sgn(x_j - x_i) = \begin{cases} -1, & if \quad x_j - x_i < 0 \\ 0, & if \quad x - x_i = 0 \\ +1, & if \quad x - x_i > 0 \end{cases} \tag{3}$$

where $n$ is the length of the dataset, $x_j$ and $x_i$ represent data points in time series $j$ and $i$, respectively ($j > i$). The difference of the magnitude from each pair of values is compared ($x_j$ - $x_i$ ) when ($j > i$) following Eq.(3). *Sgn* ($x_j - x_i$) is the function sign whose value can be -1, 0 or 1. A positive $S$ value means that the trend is increasing while a negative $S$ value means that the trend is decreasing. The variance of $S$ is estimated by the following equation:

$$Var(S) = \frac{\left[n(n-1)(2n+5) - \sum_{j-1}^{g} t_j(t_j - 1)(2t_j + 5)\right]}{18} \tag{4}$$

where $g$ is the number of the tied groups in the data and $t_j$ is the number of data points in each tied group. The significance of the trend can be found calculating $Z$ value which is acquired as follows:

$$Z = \begin{cases} \dfrac{S-1}{\sqrt{Var(S)}}, & S > 0 \\ 0, & S = 0 \\ \dfrac{S+1}{\sqrt{Var(S)}}, & S < 0 \end{cases} \tag{5}$$

In the present study, the significant level of $Z$ was assessed at 99%, 95% and 90% interval of confidence for $|Z|$ values higher than 2.58, 1.96 and 1.64 respectively. Goodman (2001) shows in Table 1 "Bayesian Interpretation of P-values" the correspondence between Z-score and the p-value. The MK test was assessed in all the NDVI, ET and precipitation spatio-temporal trend analysis.

**4. Results and discussion**

**4.1 Average NDVI, ET and precipitation maps**

NDVI values averaged for the study period (Fig. 3a) range between 0.10 and 0.35 for virtually the entire DP system. The greenest area, with NDVI values from 0.25 to 0.35, is located in the northern part of the catchment and contains dense agricultural developments and areas prone to flooding. The smallest vegetation indices are found in the lowest elevation areas of the central and southern plateau, and also in the mountainous north-west, contrasting with the greenness in the mountainous eastern part of the system.

Average ET losses range from 145 to 550 mm yr$^{-1}$ (Fig. 3b). They appear to be closely related to the terrain elevation, increasing more rapidly with elevation over the Cordillera Real. The maximum ET values are observed in high elevation areas of the Cordillera Real, followed by the northernmost part, where NDVI values are highest. The lowest ET losses occur in the drier south-west, with average annual values under 250 mm yr$^{-1}$.

A north–south gradient is observed in the spatial distribution of precipitation, with annual values ranging from 750 mm yr$^{-1}$ (in the north to 230 mm yr$^{-1}$ in the south (Fig. 3c). Detail analysis of precipitation patterns in the DP system can be found in Garreaud, et al., 2003; Pillco et al., 2007.

[Figure]

**Figure 3.** Spatial distribution of mean NDVI, ET and precipitation for the period 2002-2014 and terrain elevation of the DP system. (a) Mean NDVI; (b) Mean ET; (c) Mean precipitation and (d) Terrain elevation.

**4.2. Spatio-temporal trends in NDVI, ET and precipitation**

Fig. 4 depicts the mean annual NDVI, the annual accumulated ET and annual precipitation integrated over the entire DP system for every year in the study period together, with the change of water volume in Lake Poopó. The ET and precipitation time

series show a mean increasing trend of 4.3 mm yr$^{-1}$ and 5.2 mm yr$^{-1}$ at a significance confidence levels of 90% and 95%, respectively. This result strongly suggests that, from a catchment point of view, the increased ET water losses have been compensated by increases in precipitation during the analysed period. The catchment-wide NDVI exhibits a slower and not-significant increasing trend, mimicking to some degree the rainfall temporal pattern.

Fig. 5 illustrates the spatial distribution of the analysed temporal changes and their per-pixel significance. Fig. 5.a shows that per-pixel vegetation changes across the entire period were small over most of the catchment. Two clusters of NDVI trends can be identified: one corresponds to the areas around Charahuaito (see Fig 5.a) in the southwest part of Aroma province, exhibiting the largest positive NDVI change rate; the second cluster is located at the north end of the system, close to Machaca, and

10    exhibits a negative trend. Both clusters correspond to the areas with the highest mean NDVI (Fig. 3).

The spatial distribution of ET temporal trends reveals highly significant increments over a large central part of the system (Fig. 5.b), while the clearer negative trends are concentrated on the mountainous north-west area. Higher increasing trends were observed in precipitation than in ET over the DP system, especially in the eastern and central parts of the system. No

15    clear spatial correlations between the NDVI and ET trends have been identified, suggesting that the increase in vegetation is not the single main driver for the increase in ET losses.

[Figure]

**Figure 4.** Evolution and temporal trends of annual ET and precipitation and mean NDVI over the entire DP system, and volume change in Lake Poopó.

[Figure]

**Figure 5.** Spatial distribution of NDVI, ET and precipitation trends during the study period. Mean NDVI change (a) was calculated multiplying the trend by the time period. The spatial distribution of ET (b) and precipitation (b) represents annual changes (mm yr$^{-1}$). Figures (d), (e) and (f) show the spatial distribution of the significance level of the trends for the period 2002-2014 in the DP system. Green colour represents pixels without significance in their results. The difference in the pixel size is due to the different spatial resolution of the products.

Some differences were observed in the spatial distribution of increasing ET trends when compared with those reported by Satgé et al. (2017). The latter authors observed statistically significant ET increases in the north of the catchment (see hatched area in Fig. 5b). In this research, the highest increases were observed over a larger, central part of the system. Significant increases were also observed in the south, where the expansion of quinoa crops was reported by Jacobsen et al. (2011). Some

differences between the studies by Satgé et al. (2017) and the one presented here that may explain the different observations include: Satge et al. (2017) used approximately 180 ET products at monthly intervals for the period 2000–2014, while this study used a total of 599 ET products at eight-day intervals, from 2002 to 2014. Furthermore, Satge et al. (2017)'s analysis used the MODIS ET product version 5, at a spatial resolution of 1000 m, whereas this research benefitted from the latest version 6 of the same product, at a spatial resolution of 500 m.

When considering seasonal temporal trends, the wet season presents statistically significant increases in NDVI, ET and precipitation whereas the dry season shows no changes or slight decreases (Table 3, Fig. 6). In the wet season, NDVI increments occur largely over areas of high mean NDVI. However, the limited degree of correlation between NDVI and ET trends reinforces the hypothesis that NDVI is not the single main driver of ET changes in the DP system. The greatest precipitation increases were observed in mountainous areas over the wet season. The increased rainfall is partially transported via the stream network or aquifer to the lowlands, which could explain the observed ET increase in the central part.

[Figure]

**Figure 6.** NDVI, ET and precipitation spatio-seasonal trends over the DP system for the period 2002-2014: NDVI mean change for (a) wet season and (d) dry season; ET mean annual change (mm yr$^{-1}$) for (b) wet season and (e) dry season and precipitation mean annual change (mm yr$^{-1}$) for (c) wet season and (f) dry season.

**Table 2.** Regional annual ET mean changes in the DP System from 2002-2014. NDVI mean change for the study period 2002-2014. Statistics obtained from the non-parametric Mann Kendall test.

| | STATISTICS OF SEASONAL SPATIO-TEMPORAL ANALYSIS | | | | | |
| | Wet period | | | Dry period | | |
| | NDVI | ET | Precipitation | NDVI | ET | Precipitation |
|---|---|---|---|---|---|---|
| **Sen's slope** | 0.049 | 10.7 mm yr$^{-1}$ | 13.26 mm yr$^{-1}$ | 0.002 | -0.7 mm yr$^{-1}$ | 0.26 mm yr$^{-1}$ |
| ***p* value** | 0.06* | 0.03** | 0.06* | 0.02** | 0.24 | 0.15 |
| **% of significant pixels** | 51% | 66% | 40% | 70% | 20% | 28% |

5 \* Significant at 90% confidence level

\*\*Significant at 95% confidence level

**4.3 Per land cover analysis of ET and NDVI trends**

10 ET trends were positive in most of the scenarios except for the dry season (Fig. 7). The land cover with the highest increases was Central Crops, with trends of 5.7 mm yr$^{-1}$, 16.3 mm yr$^{-1}$ for the entire period and wet season, respectively. Southern Crops and Cordillera Real showed the next highest results after Central Crops, with significant increases during the entire period and wet season, contrasting with no changes during the dry season. Northern Crops experienced the highest decreases during the dry season. The analysis of the entire period revealed almost no changes for Wetlands and Cordillera Occidental. The trend of

15 these land covers changed slightly during the wet months, with increases around 7–8.3 mm yr$^{-1}$. During the dry season, Wetland and Cordillera Occidental showed decreasing trends. Significant increases of ET during the wet season were observed in Bareland and Autochthonous Flora. Table 3 shows the significance confidence level of the land cover trends.

The positive trends observed over natural land covers is probably related to environmental factors, such as warmer temperatures (Hunziker et al., 2018; López-Moreno et al., 2016), increases in glacial meltwater runoff (Vuille et al., 2008;

20 Cook et al., 2016) or changes in the rainfall patterns discussed in previous sections (see Fig. 4). The difference between cropland areas and the rest of the land covers could be related to non-environmental factors. Bolivia is one of the largest exporting countries of quinoa. In the last two decades, the production of quinoa in Bolivia has increased significantly (INE, 2019) responding to a growing international demand. Because the study focuses on static ROIs, the larger increases of ET in cropland areas than in the rest of the land covers may be related to changes in the irrigation management, in order to meet

25 international demand. However, there is a significant general increase of ET affecting all the land covers. The analysis of mean annual water losses per land cover type showed Eastern Cordillera, Cordillera Occidental and Northern Crops as the land covers with the highest ET mean annual values (Fig. 7).

**Table 3.** Result of the MK test.

| LAND COVERS | STATISTICS OF LAND COVER SPATIO-TEMPORAL ANALYSIS | | | | | |
|---|---|---|---|---|---|---|
| | NDVI | | | ET | | |
| | Entire period | Wet period | Dry period | Entire period | Wet period | Dry period |
| **Bareland** | 0.42 | 0.34 | 0.36 | 0.01** | 0.03** | 0.38 |
| **Wetlands** | 0.3 | 0.33 | 0.32 | 0.21 | 0.33 | 0.44 |
| **Cordillera Occ.** | 0.24 | 0.28 | 0.01** | 0.36 | 0.32 | 0.24 |
| **Cordillera Orient.** | 0.03** | 0.06* | 0.00** | 0.03** | 0.26 | 0.36 |
| **Autochthonous Flora** | 0.01** | 0.16 | 0.03** | 0.04** | 0.09* | 0.21 |
| **Northern Crops** | 0.14 | 0.09* | 0.22 | 0.11 | 0.18 | 0.34 |
| **Central Crops** | 0.00** | 0.00** | 0.08* | 0.00** | 0.02** | 0.48 |
| **Southern Crops** | 0.00** | 0.00** | 0.00** | 0.02** | 0.02** | 0.42 |

\* Significant at 90% confidence level

\*\* Significant at 95% confidence level

[Figure]

5    **Figure 7.** ET and NDVI annual land cover trends with a mean of the values corresponding to the entire period and the significance level. (a) Mean ET annual changes (mm yr$^{-1}$) and mean ET (mm yr$^{-1}$) over different land covers and (b) Mean NDVI change and mean NDVI for the same land covers as in ET analysis.

The results of the vegetation land cover analysis revealed interesting and comparable patterns of seasonal vegetation trends. Over the period 2002–2014, land cover NDVI trends were generally positive across the DP system (see Fig. 7b), especially during the wet season. As in the ET analysis, the land cover presenting the highest increases was Central Crops. Greater increases for Northern Crops and Southern Crops than for Cordillera Real were noticed during the wet season. Wetland was the land cover without human activity that presented higher increases during the wet season, followed by Autochthonous flora. Cordillera Occidental and Bareland did not present significant changes in their dynamics for the entire period. The situation could be related to the lack of vegetation in those areas. No remarkable changes were noticed during the dry season in all the land covers. Wetland and Northern Crops were the only land covers showing negative trends during the entire period. The higher increases in cropland areas, especially during the wet season when vegetation is developing and needs a greater amount of water, may be related to some changes in the water management of the area.

**5. Conclusions**

With the aim of exploring catchment changes that may have impacted the Lake Poopó water balance, this study has undertaken a comprehensive spatiotemporal analysis of changes in vegetation, evapotranspiration losses and precipitation over the DP system for the period 2002-2014, using more than 5000 satellite products. Temporal trends of these parameters have also been quantified for the main land cover classes in the area.

The analysis of vegetation changes throughout the entire study period shows slight positive and negative trends distributed over the catchment. Two clusters of consistent trends can be identified: one corresponds to areas around Charahuaito and exhibits the largest positive NDVI change rates; the second cluster, close to Machaca, shows decreasing trends. Both clusters correspond to the highest NDVI areas, indicating that the clearest, large-scale vegetation changes occurred in the most vegetated areas. The spatial distribution of ET temporal trends reveals significant increases over a large central part of the system. The clearest negative trends concentrate on the mountainous northwest area, consistent with the finding of Satgé et al., (2017).

The seasonal analysis of temporal trends revealed striking differences between wet and dry seasons. The trends of NDVI, ET and precipitation in the dry season were either slightly negative or approximately null, whereas the wet season showed increasing trends which are significant over more than 60% of the catchment area. This increasingly uneven seasonal dynamics are presumably related to comparable trends in weather and precipitation patterns and are currently under investigation.

ET losses and their trends have been estimated for the main land covers in the DP catchment. Their values indicate that the land covers with higher water consumption are: Cordillera Real, Cordillera Occidental, Northern Crops, Wetlands and Central

Crops with average values of 500, 410, 410, 370 and 310 mm yr$^{-1}$ respectively. This quantification of water consumption per cover type provides crucial information for the sustainable planning of agriculture exploitation and water resource use in the DP system.

5  Among the analysed land cover classes, only those including crops, i.e. Central and Southern Crops, plus Cordillera Real, have experienced an increase in NDVI and evapotranspiration losses, while natural covers showed either constant or decreasing NDVI trends together with increases in ET. The larger increase in vegetation and ET losses over agricultural regions, strongly suggest that cropping practices exacerbated water losses in these areas.

10  The NDVI and ET values averaged annually over the DP system increased at a mean rate of 0.001 yr$^{-1}$ and 4.3 mm yr$^{-1}$, which yields a mean NDVI and annual ET increments of 0.14 and 56 mm for the 14-year study period.  Water inputs into the system due to precipitation increased at a mean rate of 5.2 mm yr$^{-1}$, exceeding the ET rise rate. On the other hand, no neat spatial correlations between NDVI and ET trends have been identified, indicating that the NDVI is not the single main driver for the increase in ET losses. These results indicate that, despite the intensification of agriculture in the DP system between 2002 and 15  2014 increased the ET losses, these cannot be the directly linked to the Lake Poopó shrinkage. These results urge the investigation of trends in other lake water balance terms, such as the flow discharge in the Desaguadero River other tributaries, the evaporation losses and the water exchange between the lake and the aquifer.

20  **Acknowledgements** The authors express their gratitude to USGS, Climate Hazard Group, and Google Earth Engine for the availability of the MODIS and CHIRPS datasets, and to SENAMHI (Bolivia) for the in situ precipitation data.

**References**

25  Abarca-Del-Rio, R., Crétaux, J. F., Berge-Nguyen, M., & Maisongrande, P.: Does Lake Titicaca still control the Lake Poopó system water levels? An investigation using satellite altimetry and MODIS data (2000–2009). Remote sens. lett., 3(8), 707-714, doi: 10.1080/01431161.2012.667884  2012.

AghaKouchak, A., Norouzi, H., Madani, K., Mirchi, A., Azarderakhsh, M., Nazemi, A., Nasrollahi, N., Farahmand, A., 30  Mehran, A. and Hasanzadeh, E.: Aral Sea syndrome desiccates Lake Urmia: call for action. J.Great Lakes Res., 41(1), 307-311, doi: 10.1016/j.jglr.2014.12.007, 2015.

Alsdorf, D.E., Rodríguez, E. and Lettenmaier, D.P.: Measuring surface water from space. Rev. Geophys, 45(2), doi: 10.1029/2006RG000197 2007

Andreucci, D., & Radhuber, I.: Limits to "counter-neoliberal" reform: Mining expansion and the marginalisation of post-extractivist forces in Evo Morales's Bolivia. Geoforum., 84, 280-291, doi: 10.1016/j.geoforum.2015.09.002, 2017.

5    Arsen, A., Crétaux, J. F., Berge-Nguyen, M., & del Rio, R. A.: Remote sensing-derived bathymetry of Lake Poopó. Remote Sens-Basel., 6(1), 407-420, doi: 10.3390/rs6010407, 2014.

Bookhagen, B. and Burbank, D.W.: Toward a complete Himalayan hydrological budget: Spatiotemporal distribution of snowmelt and rainfall and their impact on river discharge. J. Geophys. Res., 115(F3), doi:10.1029/2009JF001426 2010

Buytaert, W. and De Bièvre, B.: Water for cities: The impact of climate change and demographic growth in the tropical Andes. Water Resources Res., 48(8), doi: doi.org/10.1029/2011WR011755 2012.

Campbell, J.B. and Wynne, R.H.:. Introduction to remote sensing. Guilford Press, USA, 2011.

Canedo, C., Pillco Zolá, R., & Berndtsson, R.: Role of Hydrological Studies for the Development of the TDP System. Water-Sui., 8(4), 144. doi: 10.3390/w8040144, 2016.

Canedo Rosso, C., Hochrainer-Stigler, S., Pflug, G., Condori, B., and Berndtsson, R.: Early warning and drought risk
20    assessment for the Bolivian Altiplano agriculture using high resolution satellite imagery data, Nat. Hazards Earth Syst. Sci. Discuss., https://doi.org/10.5194/nhess-2018-133, 2018.

Chen, J., Liao, A., Cao, X., Chen, L., Chen, X., He, C., Han, G., Peng, S., Lu, M. and Zhang, W.: Global land cover mapping at 30 m resolution: A POK-based operational approach. Int. Soc. Photogramme., 103, 7-27, doi:
25    doi.org/10.1016/j.isprsjprs.2014.09.002, 2015.

Chen, X.L., Zhao, H.M., Li, P.X. and Yin, Z.Y.: Remote sensing image-based analysis of the relationship between urban heat island and land use/cover changes. Remote Sens. of Environment, 104(2), pp.133-146, doi: 10.1016/j.rse.2005.11.016, 2006.

30    Cook, S. J., Kougkoulos, I., Edwards, L. A., Dortch, J., and Hoffmann, D.: Glacier change and glacial lake outburst flood risk in the Bolivian Andes. Cryosphere., 10, 2399-2413, https://doi.org/10.5194/tc-10-2399-2016, 2016.

Courault, D., Seguin, B. and Olioso, A.: Review on estimation of evapotranspiration from remote sensing data: From empirical to numerical modeling approaches. Irrig. Drain, 19(3-4), pp.223-249, 2005.

Crétaux, J., Jelinski, W., Calmant, S., Kouraev, A., Vuglinski, V., & Bergé-Nguyen, M. et al.: SOLS: A lake database to monitor in the Near Real Time water level and storage variations from remote sensing data. Adv. Space Res., 47(9), 1497-1507, doi: 10.1016/j.asr.2011.01.004, 2011.

Crosman, E.T. and Horel, J.D.: MODIS-derived surface temperature of the Great Salt Lake. Remote Sens. of Environment, 113(1), pp.73-81, doi: 10.1016/j.rse.2008.08.013,2009.

de Jong, R., de Bruin, S., de Wit, A., Schaepman, M. E., & Dent, D. L.: Analysis of monotonic greening and browning trends

10    from global NDVI time-series. Remote Sens. of Environment, 115(2), 692-702, doi: 10.1016/j.rse.2010.10.011, 2011.

D. Ayaviri Nina and P. Vallejos Mamani.: Cambio climático y seguridad alimentaria, un análisis en la producción agrícola. CienciAgro, 3(1): 59 – 70, 2014.

15    Didan, K.: MOD13Q1 MODIS/Terra Vegetation Indices 16-Day L3 Global 250m SIN Grid V006 [Data set]. NASA EOSDIS LP DAAC, doi: 10.5067/MODIS/MOD13Q1.006, 2015a.

Didan, K.: MYD13Q1 MODIS/Aqua Vegetation Indices 16-Day L3 Global 250m SIN Grid V006 [Data set]. NASA EOSDIS LP DAAC, doi: 10.5067/MODIS/MOD13Q1.006, 2015b.

Dinpashoh, Y., Jhajharia, D., Fakheri-Fard, A., Singh, V.P. and Kahya, E.: Trends in reference crop evapotranspiration over Iran. J. Hydrol, 399(3-4), pp.422-433, doi: https://doi.org/10.1016/j.jhydrol.2011.01.021 2011.

Duethmann, D. and Blöschl, G.: Why has catchment evaporation increased in the past 40 years? A data-based study in Austria,

25    Hydrol. Earth Syst. Sci., 22, 5143-5158, https://doi.org/10.5194/hess-22-5143-2018, 2018.

Eckert, S., Hüsler, F., Liniger, H., & Hodel, E.: Trend analysis of MODIS NDVI time series for detecting land degradation and regeneration in Mongolia. J. Arid. Environments, 113, 16-28, doi: 10.1016/j.jaridenv.2014.09.001, 2015.

30    Eimanifar, A. and Mohebbi, F.: Urmia Lake (northwest Iran): a brief review. Saline systems.,3(1), 5, doi: 10.1186/1746-1448-3-5 2005.

Espinoza Villar, J.C., Ronchail, J., Guyot, J.L., Cochonneau, G., Naziano, F., Lavado, W., De Oliveira, E., Pombosa, R. and Vauchel, P.: Spatio-temporal rainfall variability in the Amazon basin countries (Brazil, Peru, Bolivia, Colombia, and

Ecuador). International Journal of Climatology: Int. J. Climatol., 29(11), pp.1574-1594, doi: https://doi.org/10.1002/joc.1791, 2009.

Fang, N. F., Chen, F. X., Zhang, H. Y., Wang, Y. X., and Shi, Z. H.: Effects of cultivation and reforestation on suspended
5    sediment concentrations: a case study in a mountainous catchment in China, Hydrol. Earth Syst. Sci., 20, 13-25, https://doi.org/10.5194/hess-20-13-2016, 2016.

Farr, T.G., Rosen, P.A., Caro, E., Crippen, R., Duren, R., Hensley, S., Kobrick, M., Paller, M., Rodriguez, E., Roth, L. and Seal, D.,: The shuttle radar topography mission. Rev. Geophys., 45(2), doi: 10.1029/2005RG000183, 2007.

10   Fathian, F., Dehghan, Z., Bazrkar, M.H. and Eslamian, S.: Trends in hydrological and climatic variables affected by four variations of the Mann-Kendall approach in Urmia Lake basin, Iran. Hydrolog. Sci. J, 61(5), pp.892-904, doi: https://doi.org/10.1080/02626667.2014.932911, 2016.

Fensholt, R. and Proud, S.R.: Evaluation of earth observation based global long term vegetation trends—Comparing GIMMS and MODIS global NDVI time series. Remote Sens. Environ., 119, 131-147, doi: 10.1016/j.rse.2011.12.015, 2012.

Fernandes, R. and Leblanc, S.G.: Parametric (modified least squares) and non-parametric (Theil–Sen) linear regressions for predicting biophysical parameters in the presence of measurement errors. Remote Sens. Environ., 95(3), 303-316, doi: 10.1016/j.rse.2005.01.005, 2005.

20   Funk, Chris, Pete Peterson, Martin Landsfeld, Diego Pedreros, James Verdin, Shraddhanand Shukla, Gregory Husak, James Rowland, Laura Harrison, Andrew Hoell & Joel Michaelsen. "The climate hazards infrared precipitation with stations—a new environmental record for monitoring extremes". Scientific Data 2, 150066, doi:10.1038/sdata.2015.66, 2015.

Gao, X., Huete, A.R. and Didan, K.: Multisensor comparisons and validation of MODIS vegetation indices at the semiarid
25   Jornada experimental range. IEEE T. Geosci. Remote, 41(10), pp.2368-2381, doi: 10.1109/TGRS.2003.813840, 2003.

Gao, H., Bohn, T. J., Podest, E., McDonald, K. C., & Lettenmaier, D. P.: On the causes of the shrinking of Lake Chad. Environm. Res. Lett. 6(3), 034021, doi: 10.1088/1748-9326/6/3/034021 2011.

30   Garreaud, R., Vuille, M. and Clement, A.C.: The climate of the Altiplano: observed current conditions and mechanisms of past changes. Palaegeogr. Palaeocl, 194(1-3), pp.5-22, doi: 10.1016/S0031-0182(03)00269-4 ,2003.

Gocic, M. and Trajkovic, S.: Analysis of changes in meteorological variables using Mann-Kendall and Sen's slope estimator statistical tests in Serbia. Global Planet. Change, 100, pp.172-182, doi: https://doi.org/10.1016/j.gloplacha.2012.10.014, 2013.

Goodman, S. N.: Of P-values and Bayes: a modest proposal. Epidemiology, 12(3), 295-297, 2001.

Gorelick, N., Hancher, M., Dixon, M., Ilyushchenko, S., Thau, D. and Moore, R.: Google Earth Engine: Planetary-scale geospatial analysis for everyone. Remote Sens. Environ., 202,18-27, doi: 10.1016/j.rse.2017.06.031, 2017.

Guay, K.C., Beck, P.S., Berner, L.T., Goetz, S.J., Baccini, A. and Buermann, W.: Vegetation productivity patterns at high

10 northern latitudes: a multi-sensor satellite data assessment. Glob. Change Biol, 20(10), pp.3147-3158, doi: https://doi.org/10.1111/gcb.12647 ,2014.

Hammer, U.T.: Saline lake ecosystems of the world (Vol. 59). Springer Science & Business Media, Netherlands, 1986.

15 Herrmann, S.M., Anyamba, A. and Tucker, C.J.: Recent trends in vegetation dynamics in the African Sahel and their relationship to climate. Global Environ. Change, 15(4), pp.394-404, doi: https://doi.org/10.1016/j.gloenvcha.2005.08.004, 2005.

Hoffmann, D. and Requena, C.: Bolivia en un mundo 4 grados más caliente: Escenarios sociopolíticos ante el cambio climático

20 para los años 2030 y 2060 en el altiplano norte. Fundación PIEB, Programa de Investigación Estratégica en Bolivia, Bolivia, 2012.

Hostetler, S.W. Hydrological and Thermal Response of Lakes to Climate: Description and Modeling; Springer: Berlin/Heidelberg, Germany, 1995.

Huang, C., Chen, Y., Zhang, S., & Wu, J.: Detecting, extracting, and monitoring surface water from space using optical sensors: A review. Rev. Geophys., 56(2), 333-360, doi: 10.1029/2018RG000598, 2018.

Hunziker, S., Brönnimann, S., Calle, J., Moreno, I., Andrade, M., Ticona, L., Huerta, A. and Lavado-Casimiro, W.: Effects of

30 undetected data quality issues on climatological analyses. Clim. Past, 14(1), pp.1-20, doi:, https://doi.org/10.5194/cp-14-1-2018, 2018.

Instituto Nacional de Estadística. Estadísticas económicas; Instituto Nacional de Estadística, Ed.; The Bolivian National Institute of Statistics: La Paz, Bolivia, 2015. (In Spanish).

Jacobsen, S.: The Situation for Quinoa and Its Production in Southern Bolivia: From Economic Success to Environmental Disaster. J. Agron.Crop.Sci., 197(5), 390-399, doi: 10.1111/j.1439-037x.2011.00475.x, 2011.

Jiang, L., Islam, S., and Carlson, T. N.: Uncertainties in latent heat flux measurement and estimation: implications for using a simplified approach with remote sensing data. Can. J. Remote Sens., 30(5), 769-787, doi: 10.5589/m04-038, 2004.

5  Kalma, J. D., McVicar, T. R., and McCabe, M. F.: Estimating land surface evaporation: A review of methods using remotely sensed surface temperature data. Surv.Geophys., 29(4-5), 421-469, doi: 10.1007/s10712-008-9037-z, 2008.

Karimi, P. and Bastiaanssen, W. G. M.: Spatial evapotranspiration, rainfall and land use data in water accounting – Part 1: Review of the accuracy of the remote sensing data, Hydrol. Earth Syst. Sci., 19, 507-532, https://doi.org/10.5194/hess-19-507-2015, 2015.

Kendall, M. G.: Rank Correlation Methods – Griffin, London, UK, 202 pp., 1975.

Kidd, C.: Satellite rainfall climatology: A review. Int. J. Climat, 21(9), pp.1041-1066, doi: https://doi.org/10.1002/joc.635, 2001.

Li, Z.L., Tang, R., Wan, Z., Bi, Y., Zhou, C., Tang, B., Yan, G. and Zhang, X.: A review of current methodologies for regional
15  evapotranspiration    estimation    from    remotely    sensed    data. Ah.    S.    Sens., 9(5),    pp.3801-3853,    doi: https://doi.org/10.3390/s90503801, 2009.

López-Moreno, J., Morán-Tejeda, E., Vicente-Serrano, S., Bazo, J., Azorin-Molina, C., & Revuelto, J. et al.: Recent temperature variability and change in the Altiplano of Bolivia and Peru. Int. J. Climatol., 36(4), 1773-1796. doi:
20  10.1002/joc.4459, 2015.

Liu, G., Ou, W., Zhang, Y., Wu, T., Zhu, G., Shi, K., & Qin, B.: Validating and mapping surface water temperatures in Lake Taihu: Results from MODIS land surface temperature products. IEEE. J. Sel. Top. Appl. 8(3), 1230-1244, doi: 10.1109/JSTARS.2014.2386333, 2015.

Martí-Cardona, B., Prats, J., & Niclòs, R.:. Enhancing the retrieval of stream surface temperature from Landsat data. Remote Sens. of Environment, 224, 182-191, doi: 10.1016/j.rse.2019.02.007, 2019

Martí-Cardona, B., Pipia, L., Rodríguez Máñez, E., & Hans Sánchez, T.): Teledetección de la Evapotranspiración y Cambio
30  de Cubiertas en la Cuenca del Río Locumba, Perú. In XXVII Congreso Latinoamericano de Hidráulica, IAHR, Lima, Peru, 28-30 Sep. 2016.

Martí-Cardona, B., Dolz-Ripollés, J., & López-Martínez, C.: "Wetland inundation monitoring by the synergistic use of ENVISAT/ASAR imagery and ancillary spatial data", Remote Sens. of Environment, 139(12), 171–184. doi:10.1016/j.rse.2013.07.028, 2013.

5   Martí-Cardona, B., López-Martínez, C., Dolz-Ripollés, J., & Bladé-Castellet, E.: "ASAR polarimetric, multi- incidence angle and multitemporal characterization of Doñana wetlands for flood extent monitoring", Remote Sens. of Environment, 114(11), 2802–2815. doi:10.1016/j.rse.2010.06.015, 2010.

Mann, H.: Nonparametric Tests Against Trend. Econometrica., 13(3), 245. doi: 10.2307/1907187, 1957.

10   McCarthy, J.J., Canziani, O.F., Leary, N.A., Dokken, D.J. and White, K.S. eds.: Climate change 2001: impacts, adaptation, and vulnerability: contribution of Working Group II to the third assessment report of the Intergovernmental Panel on Climate Change (Vol. 2). Cambridge University Press., UK, 2001.

Micklin, P.: Desiccation of the Aral Sea: A Water Management Disaster in the Soviet Union. Sci., 241, 1170-1176. doi: 10.1126/science.241.4870.1170, 1988.

Micklin, P.: The Aral sea disaster. Annu. Rev. Earth Planet. Sci., 35, 47-72. doi: 10.1146/annurec.earth.35.031306.140120, 2007.

Mo, X., Chen, X., Hu, S., Liu, S., and Xia, J.: Attributing regional trends of evapotranspiration and gross primary productivity 20   with remote sensing: a case study in the North China Plain, Hydrol. Earth Syst. Sci., 21, 295-310, https://doi.org/10.5194/hess-21-295-2017, 2017.

Molina Carpio, J., Satgé, F. and Pillco Zola, R..: Los recursos hídricos del sistema TDPS. Available online: http://horizon.documentation.ird.fr/exl-doc/pleins_textes/divers14-09/010062840.pdf, 2014.

Monteith, J. L. et al.: Evaporation and environment, in: Symp. Soc. Exp. Biol, vol. 19, 4, 1965.

Mu, Q., Zhao, M., and Running, S. W.: Improvements to a MODIS global terrestrial evapotranspiration algorithm, Remote Sens. Environ., 115, 1781–1800, doi: 10.1016/j.rse.2011.02.019, 2011.

Neeti, N. and Eastman, J.R.: A contextual mann-kendall approach for the assessment of trend significance in image time series. T. GIS., 15(5), 599-611, doi: doi.org/10.1111/j.1467-9671.2011.01280.x, 2011.

OEA (Organización de los Estados Americanos).: Diagnóstico ambiental del sistema Titicaca-Desaguadero-Poopó-Salar de Coipasa (sistema TDPS) Bolivia-Perú. Departamento Regional y Medio Ambiente. OEA, Washington, D.C. USA, 1996.

Penman, H. L.: Natural evaporation from open water, bare soil and grass, in: Proc. R. Soc. Lond. A, vol. 193, 120–145, The Royal Society, 1948.

Perreault, T.: Dispossession by accumulation? Mining, water and the nature of enclosure on the Bolivian Altiplano. Antipode, 45(5), pp.1050-1069, doi: https://doi.org/10.1111/anti.12005 2013.

Pillco, R., Uvo, C.B., Bengtsson, L. and Villegas, R.: Precipitation variability and regionalization over the Southern Altiplano, Bolivia. *Int. J. Climatol.*, pp.149-164, 2007.

Pillco Zolá, R., Bengtsson, L., Berndtsson, R., Martí-Cardona, B., Satgé, F., Timouk, F., Bonnet, M.-P., Mollericon, L., Gamarra, C., and Pasapera, J.: Modelling Lake Titicaca's daily and monthly evaporation, Hydrol. Earth Syst. Sci., 23, 657-668. doi: 10.5194/hess-23-657-2019, 2019.

Revollo, M.M.: Management issues in the Lake Titicaca and Lake Poopo system: importance of developing a water budget. Lakes & Reservoirs: Research & Management, 6(3), pp.225-229, doi: https://doi.org/10.1046/j.1440-1770.2001.00151.x, 2001.

Rodriguez, E., C.S. Morris, J.E. Belz, E.C. Chapin, J.M. Martin, W. Daffer, S.Hensley.: An assessment of the SRTM topographic products, Technical Report JPL D-31639, Jet Propulsion Laboratory, Pasadena, California, 143 pp. available on http://www2.jpl.nasa.gov/srtm/SRTM_D31639.pdf, 2005.

Rouse, J.W., Haas, R.H., Schell, J.A., Deering, D.W.: Monitoring vegetation systems in the great plains with erts. In: Freden, S.C., Mercanti, E.P., Becker, M.A.: Third Earth Resources Technology Satellite 1 Symposium. NASA, Washington, DC, 1974.

Running, S., Mu, Q., Zhao, M. (2017). MOD16A2 MODIS/Terra Net Evapotranspiration 8-Day L4 Global 500m SIN Grid V006 [Data set]. NASA EOSDIS Land Processes DAAC, doi: 10.5067/MODIS/MOD16A2.006

Satgé, F., Bonnet, M. P., Timouk, F., Calmant, S., Pillco, R., Molina, J., ... & Garnier, J.: Accuracy assessment of SRTM v4 and ASTER GDEM v2 over the Altiplano watershed using ICESat/GLAS data. Int. J. Remote Sens., 36(2), 465-488.doi: 10.1080/01431161.2014.999166, 2015

Satgé, F., Espinoza, R., Zolá, R. P., Roig, H., Timouk, F., Molina, J., Garnier, J., Calmant, F., Seyler, F. and Bonnet, M. P.: Role of climate variability and human activity on Poopó Lake droughts between 1990 and 2015 assessed using remote sensing data. Remote Sens-Basel., 9(3), 218. doi: 10.3390/rs9030218, 2017.

5   Satgé, F., Ruelland, D., Bonnet, M.-P., Molina, J., and Pillco, R.: Consistency of satellite-based precipitation products in space and over time compared with gauge observations and snow- hydrological modelling in the Lake Titicaca region, Hydrol. Earth Syst. Sci., 23, 595-619, https://doi.org/10.5194/hess-23-595-2019, 2019.

Sen, P.K.: Estimates of the regression coefficient based on Kendall's tau. J. Am. Stat. Assoc., 63(324), 1379-1389.doi: 10   10.2307/2285891, 1968.

Schneider, P. and Hook, S.J.: Space observations of inland water bodies show rapid surface warming since 1985. Geophys. Res. Lett., 37(22), doi: https://doi.org/10.1029/2010GL045059, 2010.

Seth, A., Thibeault, J., Garcia, M. and Valdivia, C.: Making sense of twenty-first-century climate change in the Altiplano: 15   Observed trends and CMIP3 projections. Ann.Assoc Am. Geogr., 100(4), pp.835-847, doi: https://doi.org/10.1080/00045608.2010.500193, 2010.

Shadmani, M., Marofi, S. and Roknian, M.: Trend analysis in reference evapotranspiration using Mann-Kendall and Spearman's Rho tests in arid regions of Iran. Water Resourc. Manag., 26(1), pp.211-224, doi: 10.1007/s11269-011-9913-z, 2012.

20   Thibeault, J.M., Seth, A. and García, M.: Changing climate in the Bolivian Altiplano: CMIP3 projections for temperature and precipitation extremes. J. Geophys. Res-Atmos. 115(D8), doi: https://doi.org/10.1029/2009JD012718, 2010.

Urrutia, R. and Vuille, M.: Climate change projections for the tropical Andes using a regional climate model: Temperature and precipitation simulations for the end of the 21st century. J. Geophys. Res-Atmos., 114(D2), doi: https://doi.org/10.1029/2008JD011021, 2009.

25   UTNIT.: Mapa de cobertura y uso actual de la tierra, Bolivia. COBUSO-2010. Unidad Tecnica Nacional de Informacion de la Tierra. Retreived from http://cdrnbolivia.org/geografia-fisica-nacional.htm, 2011.

Vermote, E.: MOD09A1 MODIS/Terra Surface Reflectance 8-Day L3 Global 500m SIN Grid V006 [Data set]. NASA EOSDIS LP DAAC. doi: 10.5067/MODIS/MOD09A1.006, 2015.

Vuille, M., Francou, B., Wagnon, P., Juen, I., Kaser, G., Mark, B.G. and Bradley, R.S.: Climate change and tropical Andean glaciers: Past, present and future. Earth-sci. Rev., 89(3-4), pp.79-96, doi: https://doi.org/10.1016/j.earscirev.2008.04.002, 2008.

5 Wang, J., Song, C., Reager, J.T., Yao, F., Famiglietti, J.S., Sheng, Y., MacDonald, G.M., Brun, F., Schmied, H.M., Marston, R.A. and Wada, Y.: Recent global decline in endorheic basin water storages. Nat. Geosci. 11(12), p.926, doi: https://doi.org/10.1038/s41561-018-0265-7, 2018.

Wu, Z., Wu, J., Liu, J., He, B., Lei, T. and Wang, Q.: Increasing terrestrial vegetation activity of ecological restoration program 10 in the Beijing–Tianjin Sand Source Region of China. Ecol. Eng., 52,37-50, doi: 10.1016/j.ecoleng.2012.12.040, 2013.

Wurtsbaugh, W. A., Miller, C., Null, S. E., DeRose, R. J., Wilcock, P., Hahnenberger, M.,... & Moore, J.: Decline of the world's saline lakes. Nat. Geosci. 10(11), 816. doi: 10.1038/ngeo3052, 2017.

Xie, Y., Sha, Z. and Yu, M.: Remote sensing imagery in vegetation mapping: a review. J. Plant Ecol., 1(1), pp.9-23, doi: 15 https://doi.org/10.1093/jpe/rtm005, 2008.

Xu, H.: Modification of normalised difference water index (NDWI) to enhance open water features in remotely sensed imagery. Int. J. Remote Sens., 27(14), 3025-3033. doi: 10.1080/01431160600589179, 2006

20 Yue, S., Pilon, P. and Cavadias, G.: Power of the Mann–Kendall and Spearman's rho tests for detecting monotonic trends in hydrological series. J. Hydrol., 259(1-4), pp.254-271, doi: https://doi.org/10.1016/S0022-1694(01)00594-7,2002.

Zola, R.P. and Bengtsson, L.: Long-term and extreme water level variations of the shallow Lake Poopó, Bolivia. Hydrolog. Sci. J., 51(1), 98-114. doi: 10.1623/hysj.51.1.98, 2006.

Zolá, R.P. and Bengtsson, L.: Three methods for determining the area–depth relationship of Lake Poopó, a large shallow lake in Bolivia. Lakes & Reservoirs: Research & Management, 12(4), pp.275-284, doi:https://doi.org/10.1111/j.1440-1770.2007.00344.x, 2007.

---

## Author Comment (AC2) · 3 Sep 2019

Journal: HESS
Title: Mapping long-term evapotranspiration losses in the catchment of the shrinking Lake Poopó.
Author(s): Juan Torres-Batlló et al.
MS No.: hess-2019-187
MS Type: Research article

**Response to Anonymous Referee #2**

**This manuscript uses remotely sensed data products to help evaluate the role of evapotranspiration (ET) in the recession of Lake Poopo. The trends of precipitation, ET, and the normalized difference vegetation index (NDVI) were carefully analyzed, and it was found that ET was not the main contributor for this change. However, there is a lack of further analysis about the causes, while the vegetation trend has been given more attention than necessary.**

We sincerely thank the anonymous Referee #2 for their careful review of our manuscript. We have substantially revised our manuscript with special attention to the critical issues identified by the Referee and believe that our study aims and results are much better conveyed now.

The Referee's comments are indicated below in bold font, followed by our response in normal font. Blue colour shows new sentences or paragraphs as modified in the manuscript. All the answers are referenced (PX-LX) to the new manuscript which is attached at the end of the document.

Additionally, all analyses presented in the manuscript have been re-processed to the period 2002-2014, for consistency. In the original version, some had been undertaken for 2001-2014 and others for 2002-2014. The modification of the analysis period has not introduced significant changes in the results.

1. **Section 2: Although the manuscript is about evaluating whether the ET has caused the shrinking of Lake Poopo, the text about Lake Poopo and its changes is minimal. The authors are strongly suggested to offer more info on this regard (such as time series of lake level or area).**

We completely agree that the original manuscript provided insufficient background on the lake catchment issues. It didn't state the study aims clearly enough, either. Our sincere apologies for these omissions.

In the current version of the paper, we have profoundly revised the Introduction section in order to provide a thorough literature review on the Lake Poopó issues and a clear statement of the study aims.

The temporal changes of Lake Poopó have been depicted in Fig.4, on P14-L17 of the revised manuscript and included below.

[Figure]

**Figure 1:** Evolution and temporal trends of annual ET and precipitation and mean NDVI over the entire DP system, and volume change in Lake Poopó.

2. **Earlier in the manuscript, it was assumed that ET losses have increased due to increased cultivated crops. However, the analysis of ET changes in Figure 6 focuses on the ET rate for different crop types at given locations. In Figure 1, only representative ROIs of land cover types are indicated. Showing the land cover land use change (LCLUC) during the study period Will help to justify the argument that the area of cultivated crops has increased. Otherwise one can argue that the increased ET is primarily due to increased precipitation, instead of LCLUC.**

Actually, the study analysed the increase in ET and NDVI over the entire catchment and, additionally, per land cover type. But we did not attempt to quantify agricultural intensification. In order to associate mean ET and NDVI magnitudes and trends to different land cover types and explore their relative trends, we defined a number of polygons characteristic of those types (regions of interest, ROIs). Special care was taken in the definition of the ROIs, so that their land use type didn't change over the study period. Consequently, the observed land-cover ET or NDVI trends are associated to a stationary, land cover use.

We revised Section 3.2 of the manuscript (P10-L6), to clarify this point by adding the paragraph as follows:

"Regions of interest (ROIs) were delineated for each of these categories (Fig. 1) and validated by site visits and by visual inspection of multi-temporal, high resolution Google Earth images. Google Earth Engine Timelapse (Gorelick et al., 2017), was used to ensure that the ROI's land cover types had not changed during the study period. The ROIs were used to extract per-class NDVI and ET annual values and trends as explained in Section 3.3." (P10-L6)

The statement shown in the introduction of the increase of ET due to an increase of the extension of cultivated crops is based on literature review. The total cultivation area of quinoa in Bolivia has increased in the last 2 decades (INE, 2019). Jacobsen et al., 2012 shows an increase of the extension of quinoa crops in the southern part of the system. Moreover, Satgé et al., 2017

suggests that an increase of the extension of cropland areas is one of the main causes of the increase of ET. However, the highest increases of ET found by Satgé (2016) do not spatially match with the highest expansion of cropland areas found by Jacobsen (2012).

- Instituto Nacional de Estadística. Estadísticas económicas; Instituto Nacional de Estadística, Ed.; The Bolivian National Institute of Statistics: La Paz, Bolivia, 2015. (In Spanish).
- Jacobsen, S.: The Situation for Quinoa and Its Production in Southern Bolivia: From Economic Success to Environmental Disaster. J. Agron.Crop.Sci., 197(5), 390-399, doi: 10.1111/j.1439-037x.2011.00475.x, 2011.
- Satgé, F., Espinoza, R., Zolá, R. P., Roig, H., Timouk, F., Molina, J., Garnier, J., Calmant, F., Seyler, F. and Bonnet, M. P.: Role of climate variability and human activity on Poopó Lake droughts between 1990 and 2015 assessed using remote sensing data. Remote Sens-Basel., 9(3), 218. doi: 10.3390/rs9030218, 2017.

3. **Results from Figure 4 suggest that the overall precipitation has increased more than that from the ET. Consequently, it was concluded that there is no clear link between agricultural intensification and the lake recession. Given these, I don´t think the NDVI trend analyses (Figure 5 and Figure 6) are very relevant in this study. Rather, readers would likely be more interested in finding out the true driver of the depletion of the increasing trends from precipitation and ET?**

Regarding the interest of monitoring NDVI, we expected that, if the ET increase was mainly due to increased farming, a spatial correlation between NDVI and ET trends would be observed. But this was not the case. We considered that this lack of consistency between vegetation and ET trends supports our result that ET is not a main driver of the lake water loss. Some sentences have been added to the abstract and conclusion to highlight this role of the NDVI observation:

**Abstract**

"Consequently, this study found no clear link between vegetation development and ET indicating that the NDVI is not the single main driver for the increase in ET losses in the Poopó catchment during the period analysed." (P1-L29)

**Results 4.2**

"No clear spatial correlations between the NDVI and ET trends have been identified, suggesting that the increase in vegetation is not the single main driver for the increase in ET losses." (P14-L14)

"However, the limited degree of correlation between NDVI and ET trends reinforces the hypothesis that NDVI is not the single main driver of ET changes in the DP system." (P16-L8)

**Conclusions**

"On the other hand, no neat spatial correlations between NDVI and ET trends have been identified, indicating that the NDVI is not the single main driver for the increase in ET losses."(P20-L12)

In our opinion, quantifying the drivers for the depletion of Lake Poopo is a paramount work, requiring a great deal of hydrological data, part of which are unavailable. For example, there is limited data on aquifer level evolution in the study period and on the groundwater exchange

with the lake. The time series of flow discharge in the Desaguadero River, the main water input into Lake Poopó, presents significant gaps and uncertainties. Other challenges for the lake water balance are related to the contribution from minor ungauged tributaries and to the evaporation losses. We are currently working on the quantification and simulation of some of those water balance terms, but the study presented here focused on one piece of the puzzle, on the detailed temporal analysis of the spatially distributed terms: ET, precipitation and vegetation. We don't attempt to explain all issues affecting Lake Poopó, but to exploit state-of-the-art spatiotemporal data to improve our quantitative knowledge of one of the factors that has been highlighted by recent literature as a possible driver of the lake's decline: the increased crop ET losses. Our revised Introduction tries to clarify this goal of our study, humbler than an overall explanation to the issue, as it may have seemed. Also, we have added a final remark to the Conclusions:

"… despite the intensification of agriculture in the DP system between 2002 and 2014 increased the ET losses, these cannot be the directly linked to the Lake Poopó shrinkage. These results urge the investigation of trends in other lake water balance terms, such as the flow discharge in the Desaguadero River other tributaries, the evaporation losses and the water exchange between the lake and the aquifer." (P20-L14)

4. **From a water budget perspective, it is expected that the river discharge into the lake should have also increased. Therefore, the authors are suggested to show the observed river discharge, which directly affects the lake size. Has the river discharge increased? If so, why did the lake shrink? If not, how to explain its trend against the increasing trends from precipitation and ET?**

We think that the answer to comment 3 responds to comment 4, too. We would like to add that we are currently udnergoing reseacrh on the flow discharge evolution. But within the remit of this manuscript, we only atempt to contribute rigourous information about one of the water balance parameters that has been blamed by the literature, which is ET. The study was timely, since it exploited the recently improved NASA's ET product, and recent cloud-based processing capabilities, which enabled an updated and more detailed (data richer) spatio-temporal analysis of ET.

5. **Since the ET values in the cultivated agriculture areas have increased more than the other areas, it is conceivable that the irrigation withdraw has played an important role in the depletion. Some discussion on this would help to better understand the results from this study.**

A better discussion of these results has been added in the new manuscript (P17-L20).

"The difference between cropland areas and the rest of the land covers could be related to non-environmental factors. Bolivia is one of the largest exporting countries of quinoa. In the last two decades, the production of quinoa in Bolivia has increased significantly (INE, 2019) responding to a growing international demand. Because the study focuses on static ROIs, the larger increases of ET in cropland areas than in the rest of the land covers may be related to changes in the irrigation management, in order to meet international demand." (P17-L20)

6. **Page 3, Line 9: the acronym TDPS needs to be defined.**

We apologize for the omissions. All the acronyms have been checked and defined.

"The highest increases in ET were located in the central and northern part of the Desaguadero-Poopó (DP) system..." (P4-L3)

"The region includes the Titicaca-Desaguadero-Poopó-Salares (TDPS) system with a total area of 143,900 km²..." (P4-L30)

"Regions of interest (ROIs) were delineated for each of these categories..." (P10-L6)

"...and Mann-Kendall (MK) test (Mann, 1957; Kendall, 1975) as explained in the Section 3.4.1." (P10-L25)

**Page 6, Line 31: The '5,111 daily rainfall observations' doesn't match the number '5,113' in Table 1.**

As we changed the study period from 2001-2014 to 2002-2014 in order to avoid confusions, the total number of CHIRPS products has changed to 4,748. This number has been corrected throughout the text.

**7. Page 6, Lines 32-33: change "both" to "the two".**

The change has been applied.

"Fig. 2 depicts the scatter plot for the two data sets…" (P8-L11)

"Given the agreement between the two data sets…" (P8-L13)

**8. Page 7, Lines 6-9: what is the basis of choosing the MODIS reflectance data on 05/16/2012 for the mask? How does the reflectance based mask differ from the ET based mask?**

The text as follows has been added (P10-L16) to the revised manuscript to clarify this point.

"The May 2012 image was selected for the derivation of a water mask because it was acquired at the end of an extremely wet season, so it is expected to capture water bodies and prone-to-flood areas at a close-to-peak water extent condition. A constant water mask, as opposed to a time-varying one, was used for simplicity, although it may miss few, sporadic water-logged pixels." (P10-L16)

The difference between the MNDWI mask and the ET mask differ in some specific pixels that MODIS ET does not mask. Higher number of masked pixels have been observed in the MNDWI product than in the ET product related with rivers and some pixels located in humid areas.

**9. Page 12, first paragraph: As the authors pointed out, the results from Stage et al. (2017) and this study are different because different versions of MODIS ET at two temporal resolutions were used. To what degree would the results and conclusions be affected by the uncertainties associated with the MODIS NDVI and ET products? It would be interesting to add another set of results using MODIS monthly ET v6.**

The ET MODIS dataset used by Satgé et al., 2017 (version 5) and ours (version 6) is different not only in the temporal resolution but also in the spatial resolution:

"Some differences between the studies by Satgé et al. (2017) and the one presented here that may explain the different observations include: Satge et al. (2017) used approximately 180 ET products at monthly intervals for the period 2000–2014, while this study used a total of 599 ET products at eight-day intervals, from 2002 to 2014. Furthermore, Satge et al. (2017)'s analysis used the MODIS ET product version 5, at a spatial resolution of 1000 m, whereas this research

benefitted from the latest version 6 of the same product, at a spatial resolution of 500 m. " (P15-L10)

In this study, only good quality pixels have been selected for ET as well as NDVI. Good quality pixels are all those pixels non-contaminated by aerosols, clouds and physical disturbances that can produce noise in the final image. A quality mask has been applied for ET and NDVI choosing only quality pixels per each observation.

The validation of ET MOD16A2 was carried out at 46 eddy flux towers, global 232 watersheds as well as global meteorological data (Running et al., 2017) extracted from Global Modelling Assimilation Office (GMAO). The results show an acceptable Mean Absolute Error (MAE) of about 0.33 mm day$^{-1}$ (towers), 0.31 mm day$^{-1}$ (GMAO meteorology) with a daily biases of about -0.11 mm day$^{-1}$ (towers) and -0.02 mm day$^{-1}$ (GMAO).

A better explanation of how the validation was carried out together with the uncertainties of the product can be found in the User's Guide MOD16A2 (Running et al., 2017):

https://lpdaac.usgs.gov/documents/378/MOD16_User_Guide_V6.pdf

The validation of MODIS NDVI (MOD13Q1 and MYD13Q1) was carried out based on comparisons with radiometric field worldwide measurements and space and airborne sensors. The results showed an accuracy of about ± 0.025.

The uncertainties of MODIS NDVI (MOD13Q1, MYD13Q1) appear when there are pixels with low or not acceptable quality available within a compositing period.

A better explanation of how the validation was carried out can be find in the User's Guide MOD13Q1 (Didan et al., 2015). The following link shows the reliability/uncertainties of MOD13Q1/MYD13Q1 together with a list of supporting studies validating the product.

https://landval.gsfc.nasa.gov/ProductStatus.php?ProductID=MOD13

The information regarding the uncertainties of the MODIS NDVI and MODIS ET products have been added in the manuscript:

"The accuracy of the MOD13Q1 and MYD13Q1 NDVI values for clear pixels is reported as ± 0.025 (Gao et al., 2003). Table 1 provides an overview of the used dataset." (P7-L8)

"This study used the ET product MOD16A2 version 6, which improves the previous version 5 in spatial resolution, from 1000 m to 500 m, and in the accuracy of near real-time surface weather data.  It has been validated for 232 watersheds and 46 eddy flux towers (Running et al., 2017), and its accuracy has been assessed with an average mean absolute error of about 24 %, within the 10-30% range of accuracy of ET observations (Running et al., 2017; Jian et al., 2004; Kalma et al., 2008). Regarding the ET and NDVI products' accuracy, it is important to note that they undergo strict, frequent calibrations to ensure consistent measurements, and hence provide robust, reliable temporal trends." (P7-L14)

**References:**

Running, S.W., Mu, Q., Zhao, M. and Moreno, A., 2017. User's Guide MODIS Global Terrestrial Evapotranspiration (ET) Product (NASA MOD16A2/A3). NASA Earth Observing System MODIS Land Algorithm. Version, 1.

Jiang, L., Islam, S., and Carlson, T. N.: Uncertainties in latent heat flux measurement and estimation: implications for using a simplified approach with remote sensing data. Can. J. Remote Sens., 30(5), 769-787, doi: 10.5589/m04-038, 2004.

Kalma, J. D., McVicar, T. R., and McCabe, M. F.: Estimating land surface evaporation: A review of methods using remotely sensed surface temperature data. Surv.Geophys., 29(4-5), 421-469, doi: 10.1007/s10712-008-9037-z, 2008.

Didan, K.: MOD13Q1 MODIS/Terra Vegetation Indices 16-Day L3 Global 250m SIN Grid V006 [Data set]. NASA EOSDIS LP DAAC, doi: 10.5067/MODIS/MOD13Q1.006, 2015a.

Didan, K.: MYD13Q1 MODIS/Aqua Vegetation Indices 16-Day L3 Global 250m SIN Grid V006 [Data set]. NASA EOSDIS LP DAAC, doi: 10.5067/MODIS/MOD13Q1.006, 2015b.

**10. Page 12, section number '4.3.1' should be '4.3'.**
The numeric change of the section has been applied:

"4.3 Per land cover analysis of ET and NDVI trends" (P17-L8)

**11. Please discuss the errors and uncertainties associated with the datasets employed in this study.**
The errors and uncertainties of MODIS NDVI and MODIS ET have been presented in the response to comment 9.

Some authors have reported that CHIRPS tends to overestimate and underestimate moderate and extreme rainy episodes (Toté et al., 2015; Trejo et al., 2016, Bai et al., 2018; Paredes-Trejo et al., 2017).  Despite the reported uncertainties, CHIRPS precipitation data were successfully validated in areas adjacent to the DP system by Satgé et al. (2019) and Canedo et al. (2018). We also undertook a rigorous and successful validation (see below and on P8-L1).  The rainfall over high altitude areas in the DP system is very poorly gauged, so we had to resort to remote sensing based rainfall data to account for the precipitation over those areas. Our validation provides strong confidence on the consistency between on-site and remote sensing measurements. However, the main conclusions of our study are based on the temporal trends, not on the absolute values of the parameters. Beyond inaccuracies of satellite measurements, they undergo strict frequent calibrations to ensure consistent measurements, and hence, consistent temporal trends.

**3.1.2 Precipitation data**

"Precipitation data were collected from on-site meteorological stations and from satellite rainfall estimation products. Monthly precipitation data were obtained from twenty-one meteorological stations of the Servicio Nacional de Meteorología e Hidrología de Bolivia (SENAMHI). Figure 1 shows the location of these stations, which is sparse and not well representative of high elevation areas. In order to retrieve measurements from areas poorly gauged by meteorological stations, satellite precipitation estimations from CHIRPS were used. CHIRPS is a global rainfall dataset at 0.05 arc degree resolution, derived from satellite and in-situ precipitation measurements, developed by the U.S. Geological Survey (USGS) and the

Climate Hazards Group (Funk et al., 2015).  CHIRPS precipitation data were successfully validated in areas adjacent to the DP system by Satgé et al. (2019) and Canedo et al. (2018). Here, monthly accumulated precipitation from each on-site meteorological station was compared to the one provided by its corresponding CHIRPS pixel. Fig. 2 depicts the scatter plot for the two data sets, with showing a Spearman's correlation coefficient (CC) of 0.87 with significant results and bias of 2.25 mm month$^{-1}$ (Fig. 2).  Given the agreement between the two data sets, the regional analysis of precipitation was carried out using CHIRPS data, which provides full coverage of the DP system. A bias correction was not applied since only the precipitation temporal trends, and not their absolute values, were needed for the purpose of this study. The lack of precipitation ground data for the validation of the CHIRPS product over the highest parts of the DP system is acknowledged as a source of uncertainty, and current efforts are in place to improve their gauging." (P8-L1)

[Figure]

**Figure 2.** Scatter plot of monthly satellite rainfall estimations (CHIRPS) versus rain gauges (SENAMHI).

**References:**

Bai, L., Shi, C., Li, L., Yang, Y. and Wu, J., 2018. Accuracy of CHIRPS satellite-rainfall products over mainland China. *Remote Sensing*, *10*(3), p.362.

Paredes-Trejo, F.J., Barbosa, H.A. and Kumar, T.L., 2017. Validating CHIRPS-based satellite precipitation estimates in Northeast Brazil. *Journal of arid environments*, *139*, pp.26-40.

Toté, C., Patricio, D., Boogaard, H., Van Der Wijngaart, R., Tarnavsky, E. and Funk, C., 2015. Evaluation of satellite rainfall estimates for drought and flood monitoring in Mozambique. *Remote Sensing*, *7*(2), pp.1758-1776.

Trejo, F.J.P., Barbosa, H.A., Peñaloza-Murillo, M.A., Moreno, M.A. and Farías, A., 2016. Intercomparison of improved satellite rainfall estimation with CHIRPS gridded product and rain gauge data over Venezuela. *Atmósfera*, *29*(4), pp.323-342.

Funk, C., Peterson, P., Landsfeld, M., Pedreros, D., Verdin, J., Shukla, S., Husak, G., Rowland, J., Harrison, L., Hoell, A. and Michaelsen, J., 2015. The climate hazards infrared precipitation with stations—a new environmental record for monitoring extremes. *Scientific data*, *2*, p.150066.

12. **Figure 5: It is unclear how the 'ET annual mean change' was calculated for the wet season and the dry season. Were they first calculated for two months and then scaled up to annual?**

Yes. The seasonal ET annual mean change was calculated with daily ET values for January-February and July-August and the scaled up to annual. A total of 105 and 104 observations for the wet and dry period respectively were used in the regression. The magnitude of each observation was in mm day$^{-1}$ and the results were scaled up to mm yr$^{-1}$.

**Mapping evapotranspiration, vegetation and precipitation trends in the catchment of the shrinking Lake Poopó.**

Juan Torres-Batlló[1], Belén Martí-Cardona[1], Ramiro Pillco-Zolá[2]

[1]Department of Civil and Environmental Engineering, University of Surrey, Guildford, UK
[2]Instituto de Hidráulica e Hidrología, Universidad Mayor de San Andrés, La Paz, Bolivia

**Abstract.**

Lake Poopó is located in the Andean Mountain Range Plateau or Altiplano. A general decline in the lake water level has been observed in the last two decades, coinciding roughly with an intensification of agriculture exploitation such as quinoa crops. Several factors have been blamed for the shrinkage in the extent of the lake, including climate change, increased farming, mining abstractions and population growth. Being an endorheic catchment, evapotranspiration (ET) losses are expected to be the main water output mechanism. This study used a time series of more than 5000 satellite data-based products to map ET, vegetation index (NDVI) and precipitation trends in the Poopó catchment between 2002 and 2014. The aim was to explore links between ET, vegetation, precipitation, land use and the lake decline. The years 2015 and 2016 were excluded from the analysis due to the strong impact of El Niño phenomenon over the study area, which could have masked long term temporal trends related to land use.

We quantified the ET and NDVI magnitudes and trends over the lake Poopó catchment and for areas characteristic of the main cover types during the study period. It became clear that cultivated areas were the ones which had experienced the largest increase in water consumption, although they were not in all instances the land covers with the largest losses. This quantification provides essential information for the sustainable planning of water resources and land uses in the catchment.

We also collected on-site and satellite precipitation data. When integrated over the entire catchment, the overall ET losses showed a sustained increasing trend at an average rate of 4.3 mm yr$^{-1}$. Rainfall water inputs followed a similar trend, with a higher increasing rate of 5.2 mm yr$^{-1}$. Based on these results and from the point of view of the catchment water resources, the ET loss intensification derived from crop expansion has been compensated by the increase in precipitation. Furthermore, no clear spatial correlation was found between ET and NDVI temporal trends. Consequently, this study found no clear link between vegetation development and ET indicating that the NDVI is not the single main driver for the increase in ET losses in the Poopó catchment during the period analysed.

**1. Introduction**

Endorheic or terminal catchments, i.e. those with no surface drainage to rivers or oceans, constitute 25% of the world's continental area (excluding Greenland and Antarctica) and are mostly located in semi-arid and arid climates (Hostetler, 1995). Given the lack of outlet drainage, the surface water storage in endorheic catchments depends critically on the equilibrium among precipitation, evapotranspiration (ET) and groundwater exchange (Hammer, 1986; McCarthy et al., 2001; Wang et al., 2018). This equilibrium is affected, for example, by increases in agricultural land and irrigation, which intensifies ET water losses. ET and evaporation from open water surfaces are also exacerbated by warming temperatures. Climate change combined with human activity has caused severe perturbances in the water balance of several endorheic basins around the world, leading to the shrinkage of terminal lakes in these basins (Wurtsbaugh et al., 2017; Wang et al., 2018), such as Lake Urmia in Iran (Eimanifar and Mohebbi, 2005; AghaKouchak et al., 2015), the Aral Sea in Kazahkstan and Uzbekistan (Micklin, 1988; Micklin, 2007), Lake Chad in Central Africa (Gao et al., 2011) and the Great Salt Lake in the USA (Wurtsbaugh et al., 2017).

The Altiplano, in the Andean Mountain Range, is an endorheic, semi-arid catchment and the second largest mountain plateau in the world. Several studies have highlighted the vulnerability of the Altiplano's water resources to climate change. For example, López-Moreno et al. (2015) and Hunziker et al., (2018) documented temperature increases in the Altiplano in the order of 0.20°C decade$^{-1}$ for the periods 1965-2012 and 1981-2010, respectively. Hoffmann and Requena (2012) estimated the impact of long-term temperature increments on water resources in the northern part of the Altiplano and concluded that it would lead to a dramatic reduction of water in lakes, rivers, glaciers and wetlands, especially during the dry season. Using time-series of satellite images, Cook et al. (2016) revealed a 43% reduction of the total glacial cover in the Bolivian Andean Mountains in the period 1986-2014, which has given rise to new and potentially dangerous proglacial lakes. Vuille et al. (2008) projected a decrease in water availability due to the retreat of Tropical Andean glaciers and the consequent decrease of their reservoir effect. Using global and regional climate models, Thibeault et al. (2010), Seth et al., (2010) and Urrutia and Vuille (2009) predicted changes in precipitation patterns that would lead to the intensification of floods and droughts in the area. In addition to climate change, anthropogenic activities have increased the pressure on the Altiplano's water resources and are likely to exacerbate it further in the future. For example, annual quinoa yield, for which the Altiplano is a major producer, escalated from 28,500 to 75,500 tons in Bolivia between 2008 and 2014 (INE, 2019). Satgé et al. (2017) showed increasing ET rates over cropland in the Altiplano between 2000 and 2014, suggesting its connection to the agriculture intensification. According to Buytaert. and De Bièvre (2012) the projected population growth in the tropical Andes will increase water demand by up to 50% in 2050. Based on population projections and glacio-hydrological simulations, Kinouchi et al., (2019) revealed several scenarios of increase in water demand in Bolivia from 2011 to 2039 of magnitudes ranging from 15% to 53%. Mining,

which is one of the main economic activities in the Altiplano, has increased significantly during the last decade leading to the release of heavy metals and other pollutants into the water bodies thus impairing soil productivity, local fauna and human water use (Perreault, 2013; Andreucci and Radhuber, 2017).

5    Lake Poopó, at the South East of the Andean Altiplano, is a shallow saline lake which supports livestock and fishing activities of local communities (Zola and Bengtsson, 2006; Zola and Bengtsson, 2007; Satgé et al., 2017). More than half of Lake Poopo's annual input comes from the Desaguadero River, which originates from Lake Titicaca's surcharge. Apart from rare overspills towards the Laca Jahuira River, at the south end of the lake (Revollo, 2001; Zola and Bengtsson, 2007), the major water loss mechanism of Lake Poopó is evaporation. The Desaguadero River and Poopó's catchments undergo pronounced

10   wet and dry seasons, causing a rapid recharge of Lake Poopó during the wet season, and a slower decline of water levels as the evaporation exceeds the inflows in the dry season (Zola and Bengtsson, 2006). The annual maximum and minimum lake extents strongly depends on the annual precipitation, as shown by Satgé, 2017. As a result of this dependency, Lake Poopó dried completely on several occasions in the past after severe dry episodes, e.g. in the early 1940´s, 1970s, mid-1990´s and late 2015 (Zola and Bengtsson, 2006).

Apart from the above mentioned specific extreme dry periods, some authors have claimed a declining trend in Lake Poopó water storage in the last two decades (Arsen et al., 2014; Satgé et al., 2017). The latter author highlighted the clear decoupling of the lake extent from the annual precipitation in 2013 and 2014. Several studies have related the lake's decline to the abovementioned pressures over the Altiplano water resources: rapid expansion of arable land in the catchment and an

20   associated increase of ET losses (Satgé et al., 2017); the decreasing flow discharge in the Desaguadero River, presumably linked to irrigation uses (Zola and Bengtsson, 2006; Abarca- Del- Rio et al., 2012; Molina Carpio et al., 2014), and the increased evaporation due to climate warming (Abarca- Del- Rio et al., 2012; López-Moreno et al., 2015).

The lack of substantial ground-based hydro-meteorological observations and the large extent of the study area lead to the use

25   of remote sensing data techniques to map and monitor the Altiplano water resources trends. Earth observation satellites have imaged the Earth periodically since the 1970s. These images can be processed to derive a wide range of spatial information of paramount relevance for water resource monitoring, such as the extent of water bodies (Campbell and Wynne, 2011; Alsdorf et al., 2007; Marti-Cardona et al., 2010; Marti- Cardona et al., 2013; Huang et al., 2018), the spatial distribution of rainfall (Kidd, 2001; Espinoza et al., 2009; Bookhagen and Burbank, 2010), inland water surface temperature (Schneider and Hook,

30   2010; Crosman and Horel, 2009; Liu et al., 2015; Marti-Cardona et al., 2019), vegetation indexes (Chen et al., 2006; Xie et al., 2008; Hermann et al., 2005; de Jong et al., 2011; Eckert et al., 2015; Duethmann and Blöschl, 2018) and ET dynamics (Courault et al., 2005; Li et al., 2009; Karimi and Bastiaanssen, 2015; Marti- Cardona et al., 2016; Mo et al., 2017; Running et al., 2017) among others.

Satgé et al. (2017) used monthly satellite ET products to conduct a spatio-temporal assessment of ET losses in the Altiplano for the period 2000-2014 and pointed at the ET increase from cropland as a key driver of Lake Poopó's decline. The highest increases in ET were located in the central and northern part of the Desaguadero-Poopó (DP) system, despite the maximum cropland expansion being reported at the south and southwest of Lake Poopó by Jacobsen et al. (2011) and Ayaviri and Vallejos (2014). So far, there is no integrated annual and seasonal assessment of precipitation, ET and vegetation tendencies for the study area.

The study presented here undertook for the first time an integrated assessment of vegetation, ET losses and precipitation spatio-temporal trends over the DP system. The study exploits the improved ET product from NASA's Moderate Resolution Imaging Spectroradiometer (MODIS, MOD16A2.006 Global Terrestrial Evapotranspiration 8-Day Global 500m; Running et al., 2017) together with MODIS Terra and Aqua vegetation index product (MOD13Q1.006 and MYD13Q1.006 16-Day Global 250m; Didan, 2015) and Climate Hazards Group InfraRed Precipitation with Station Data (CHIRPS ;Funk et al., 2014) precipitation product. The processing was conducted using the new cloud-based Google Earth Engine, which enabled the integration of the complete time series of daily and 8-day products for the period of analysis, involving over 5000 raster data sets. Temporal trends of the vegetation amount, ET and precipitation were mapped for the entire study period, which encompassed the sharp increase in quinoa yield. The trends for the wet and dry seasons were also analysed independently for the first time, as these were as these were expected to intensify by Thibeault et al. (2010), Seth et al., (2010) and Urrutia and Vuille (2009). An additional novelty of this study is the characterization of ET and vegetation amount trends and absolute values for the broad land uses and geographical areas in the Altiplano. Although absolute values must be taken with caution due to their limited validation over the study area, their temporal trends are robust, and they represent the relative water consumption among different land covers. This information is of paramount relevance for sustainable land use planning and management of water resources in the Altiplano. It is worth noting that, although final figures of precipitation inputs and ET outputs are provided for the entire system, a water balance was not attempted and the main conclusions are drawn from the comparison of the reliable temporal trends of those parameters.

**2. Study area and period**

The Andean Altiplano is the largest plateau on the Earth, after the Tibetan Plateau. The Altiplano has an approximate area of 192,000 km², distributed over the countries as follows: Chile (4%), Peru (26%) and Bolivia (70%) and has an average elevation of 4,000 m. The highest point is Mount Sajama (6,542 m) and the lowest is at the bottom of Lake Titicaca (3,533 m) (OEA, 1996). It represents one of the poorest regions in South America with a total population of 2.2 million (OEA, 1996). The region includes the Titicaca-Desaguadero-Poopó-Salares (TDPS) system with a total area of 143,900 km² (Satgé et al., 2015) at an average elevation of 3,810 m. The TDPS system is an endorheic catchment that comprises a physiographic region of South America located between latitudes and longitudes from 14° S to 21° S and from 71° W to 66° W. The geographical limits of the system are: the Cordillera Occidental to the west, the Cordillera Real (separates the

catchments of the Amazon) to the east, Lake Titicaca to the north and Salar de Uyuni to the south. This research will focus on the DP system within the TDPS system, which is the surface catchment contributing runoff to Lake Poopó with an area of approximately 58,000 km$^2$. The DP is an endorheic system with rare overspill discharge events towards Salar de Uyuni (Revollo, 2001; Zola and Bengtsson, 2007). Below Lake Titicaca, the Desaguadero River conveys water from Titicaca to Poopó, which is the adjacent area to this, the lowest point of the DP system, at an altitude of approximately 3,600 m.

Annual precipitation varies with the latitude, ranging from about 750 mm yr$^{-1}$ in the north to 160 mm yr$^{-1}$ in the arid south. Rainfall is very unevenly distributed over the year with 95% of the annual precipitation falling between October and March, while July and August are the driest months. The average annual temperatures are around 10°C, achieving their minimum during the months of June and July and their peaks in the months of December and January. As it is a plain situated at high altitude it undergoes large thermal amplitudes from -5 °C at night to 25°C at noon. Because of the altitude, the high solar radiation generates intense potential evapotranspiration of 1,000-1,500 mm yr$^{-1}$ (OEA, 1996). According to Pillco- Zolá et al., (2019), the monthly average relative humidity varies between 52% and 68%. In terms of vegetation, 65% of the total vegetation area is covered by grassland followed by alpine vegetation (12%), crops (11%), bareland (9%) and wetland (3%). The most common crops in the Altiplano are potatoes and quinoa (Jacobsen, 2011; Canedo et al., 2016) being the largest quinoa production area in the planet.

This study analysed the vegetation, evapotranspiration and precipitation spatio-temporal trends in the DP system for the period 2002-2014. The years 2015 and 2016 were excluded from the analysis due to the occurrence of an extreme El Niño phenomenon, which significantly altered the hydrological trends. The trends were analysed considering the entire time series, and also focusing on the wettest months, January and February, and the driest ones, July and August. The overall and seasonal trends were mapped over the entire DP system, and also examined independently for the main land cover types.

[Figure]

**Figure 1.** Study area with (a) the location of the TDPS system within the South American continent; (b) DP system within the TDPS and (c) DP system, ROIs representative of land cover types, and location of meteorological stations.

**3. Methodology**

**3.1 Experimental data**

The remote sensing products and on-site records used in this study are listed in Table 1. Most of the analysis were carried out using the Google Earth Engine (GEE) platform (Gorelick et al., 2017), which enabled the effective access and processing of large spatio-temporal data sets.

**3.1.1 Satellite data**

Two identical MODIS instruments are installed on board the Terra and Aqua satellites of the National American Space Agency (NASA), each imaging the entire Earth twice daily. Optimal quality pixels (i.e. with no sub-pixel cloud, low aerosol nor sensor view angle < 30 degrees) from consecutive MODIS images are selected and used to produce vegetation index maps at 8- day intervals, made available from products MOD13Q1 and MYD13Q1 (Didan, 2015a, b). For the analysis of vegetation trends in the DP system, this study used the Normalized Difference Vegetation Index (NDVI, Rouse Jr, 1974). The accuracy of the MOD13Q1 and MYD13Q1 NDVI values for clear pixels is reported as ± 0.025 (Gao et al., 2003).

Global evapotranspiration (ET) estimates are also derived from MODIS images combined with ancillary datasets made available at 8-day intervals as product MOD16A2 (Mu et al., 2011; Running et al., 2017). These ET estimates account for the actual (as opposed to potential) evaporation from canopy and soil, and for the real transpiration through plant´s stomata, and are provided in mm day$^{-1}$. This study used the ET product MOD16A2 version 6, which improves the previous version 5 in spatial resolution, from 1000 m to 500 m, and in the accuracy of near real-time surface weather data. It has been validated for 232 watersheds and 46 eddy flux towers (Running et al., 2017), and its accuracy has been assessed with an average mean absolute error of about 24 %, within the 10-30% range of accuracy of ET observations (Running et al., 2017; Jian et al., 2004; Kalma et al., 2008). Regarding the ET and NDVI products' accuracy, it is important to note that they undergo strict, frequent calibrations to ensure consistent measurements, and hence provide robust, reliable temporal trends.

MODIS surface reflectance (Vermote, 2015) data was acquired for 16 May 2012, at the end of the wettest season in the study period, to generate a water mask. Table 1 provides an overview of the satellite data used.

**Table 1.** Datasets used in this study.

| | DATASETS | | | | | |
|---|---|---|---|---|---|---|
| PRODUCT NAME | MOD16A2 | MOD13Q1 | MYD13Q1 | MOD09A1 | SRTM | CHIRPS |
| Physical magnitude | ET | NDVI | NDVI | Reflectance | Terrain elevation | Precipitation |
| Spatial resolution | 500 m | 250 m | 250 m | 500 m | 30 m | 0.05 arc degrees |
| Temporal interval | 8 days | 16 days | 16 days | 8 days | N/A | Daily |
| Number of products used | | | | 16-May 2012 | Mosaic of 10 granules | |
| Entire study period | 599 | 299 | 288 | | | 4748 |
| Wet season | 105 | 52 | 48 | | | 770 |
| Dry season | 104 | 52 | 52 | | | 806 |

**3.1.2 Precipitation data**

Precipitation data were collected from on-site meteorological stations and from satellite rainfall estimation products. Monthly precipitation data were obtained from twenty-one meteorological stations of the Servicio Nacional de Meteorología e

5    Hidrología de Bolivia (SENAMHI). Figure 1 shows the location of these stations, which is sparse and not well representative of high elevation areas. In order to retrieve measurements from areas poorly gauged by meteorological stations, satellite precipitation estimations from CHIRPS were used. CHIRPS is a global rainfall dataset at 0.05 arc degree resolution, derived from satellite and in-situ precipitation measurements, developed by the U.S. Geological Survey (USGS) and the Climate Hazards Group (Funk et al., 2015). CHIRPS precipitation data were successfully validated in areas adjacent to the DP system

10   by Satgé et al. (2019) and Canedo et al. (2018). Here, monthly accumulated precipitation from each on-site meteorological station was compared to the one provided by its corresponding CHIRPS pixel. Fig. 2 depicts the scatter plot for the two data sets, with showing a Spearman's correlation coefficient (CC) of 0.87 with significant results and bias of 2.25 mm month[-1] (Fig. 2). Given the agreement between the two data sets, the regional analysis of precipitation was carried out using CHIRPS data, which provides full coverage of the DP system. A bias correction was not applied since only the precipitation temporal

15   trends, and not their absolute values, were needed for the purpose of this study. The lack of precipitation ground data for the validation of the CHIRPS product over the highest parts of the DP system is acknowledged as a source of uncertainty, and current efforts are in place to improve their gauging.

[Figure]

**Figure 2.** Scatter plot of monthly satellite rainfall estimations (CHIRPS) versus rain gauges (SENAMHI).

**3.1.3 Topography model and land cover cap**

Topographic data for the DP system was obtained from the Shuttle Radar Topography Mission (SRTM) terrain elevation model
(Farr et al., 2007). The terrain topography was used to delineate the DP watershed boundaries, to inform the delineation of
control regions and to interpret the study results. The study used the version 3.0 with 30 m of spatial resolution and an absolute
vertical accuracy of ~9 m (90 % confidence) or better (Rodriguez et al. 2005).

The land cover map *Mapa de Cobertura y Uso Actual de la Tierra en Bolivia*, COBUSO-2010 (UTNIT, 2011), was used to
inform the delineation of the control regions as explained in Section 3.2. Commissioned by the Bolivian Government,
COBUSO-2010 is a country-wide, 30 m spatial resolution land cover map derived from 60 Landsat 5 scenes acquired between
2006 and 2010. This map contains 21 different land cover types within the DP system.

**3.2 Definition of control areas for the main land cover types**

In order to associate the mean ET and NDVI magnitudes and trends to the main land cover types in the DP system and to
explore their relative trends, eight broad land cover categories were defined based on the COBUSO-2010 classes. These
categories and the COBUSO-2010 classes they correspond to are briefly described below.

- Northern Crops corresponds to cropland areas located in the northern part of the system, which were under
  exploitation long before 2002, the beginning of the study period. Northern Crops comprises Multiple Crops extracted
  from COBUSO-2010.
- Central Crops represents agricultural areas located in the central valley of the DP system on alluvial soils and rather
  flat topography following the course of the Desaguadero River, grouping Multiple Crops from COBUSO-2010.
  Besides Multiple Crops, some residual pixels belonging to Central Crops are classified as Semi-arid Grassland and
  Wetland in COBUSO-2010.
- Southern Crops are located south of Lake Poopó and encompass the COBUSO-2010 land cover of Multiple Crops.
- Cordillera Real groups Multiple Crops, Sub-humid Andean Forest, Scattered Vivacious High-Andean Vegetation and
  Semi-arid Grassland located above 4200 m. All the covers are included in the same category, since their mixture
  occurs at a scale difficult to separate at the ET product resolution. An abundance of irrigation systems has been
  reported in this area by Canedo et al., 2016 together with an expansion of the mining activity (Perreault, 2013).
- Cordillera Occidental includes Scattered Vivacious High-Andean Vegetation and Scattered Puna in Sand areas in the
  North West mountain range.
- Autochthonous Flora encompasses Semi-arid Grassland, Scattered Puna in Sand areas, Scattered Vivacious High-
  Andean Vegetation and Sand Deposits in rare occurrences.

- Wetland areas are located in the northern and north-western part of the system due to the high availability of water. These areas represent flood zones with peculiar ecosystems and encompasses Wet Grasslands.
- Bare land groups Sand, Sault and Lacustrine deposits with almost no vegetation. The highest extent of bareland is found in the arid south and southwest of the DP system.

Regions of interest (ROIs) were delineated for each of these categories (Fig. 1) and validated by site visits and by visual inspection of multi-temporal, high resolution Google Earth images.  Google Earth Engine Timelapse (Gorelick et al., 2017), was used to ensure that the ROI's land cover types had not changed during the study period. The ROIs were used to extract per-class NDVI and ET annual values and trends as explained in Section 3.3.

**3.3 Masking water pixels**

A MODIS surface reflectance image from 16 May 2012 was used to calculate the Modified Normalized Difference Water Index (MNDWI) (Xu, 2006) over the entire DP system. A safe threshold of -0.3 was applied in order to identify likely

15   waterlogged pixels and produce the MNDWI water mask. This mask was applied to the MODIS NDVI products prior to any analysis. It was not applied to the ET products since a water mask is already incorporated in their derivation.  The May 2012 image was selected for the derivation of a water mask because it was acquired at the end of an extremely wet season, so it is expected to capture water bodies and prone-to-flood areas at a close-to-peak water extent condition. A constant water mask, as opposed to a time-varying one, was used for simplicity, although it may miss few, sporadic water-logged pixels.

**3.4 Retrieval and mapping of vegetation, evapotranspiration and precipitation dynamics**

Per pixel annual averages of NDVI, ET losses and precipitation were calculated for the study period, using the entire data set of the corresponding MODIS products. Per pixel ET, NDVI and precipitation temporal trends and their significance were

25   assessed by calculating the Sen´s slope estimator (Sen, 1968) and Mann-Kendall (MK) test (Mann, 1957; Kendall, 1975) as explained in the Section 3.4.1. Both, trend slope and MK test were applied to per-pixel data using the entire study period, and also considering the wet and dry seasons independently. The approach of the study is based on how physical processes change over time. This is the reason why the trend analysis was carried out under the assumption that the datasets follow a long-term persistence viewpoint.

NDVI, ET and precipitation average values and trends were mapped over the entire DP system and analysed together with the digital elevation model to identify physiographical regions exhibiting different temporal patterns. Average values and trends of NDVI, ET and precipitation were averaged within the ROIs representative of the main land cover types, to identified differential behaviours among them.

**3.4.1 Calculation of trends and their significance**

Tests for the detection of trends in time series can be classified as parametric and non-parametric methods. Non-parametric tests do not presume the data distribution, only require them to be statistically independent (Gocic and Trajkovic, 2013). The Sen´s slope is a non-parametric method to measure data trends, robust to outlier values (Yue et al., 2002; Neeti and Eastman, 2011; Guay et al., 2014) which has become a common alternative to the Ordinary Least Squares (OLS) method in the last decades for spatio-temporal vegetation and hydro-climatological analysis (Yue et al., 2002; Fernandes and Leblanc, 2005; Wu et al., 2013; Gocic and Trajkovic, 2013). The Sen´s slope is calculated as follows:

$$Q = Median\left\{\frac{x_j - x_i}{j - i}\right\}, \text{for } i = 1, \dots, N \tag{1}$$

Where $Q$ is the slope, $x_j$ and $x_i$ are the values at time $j$ and $i$ ($j > i$) respectively and N is the number of time periods representing the totality of the observations per pixel. Each point is compared with all the next data points and the median of the slopes is used to characterize the trend. The significance of the time series trends was evaluated by the non-parametric MK test (Mann, 1957; Kendall, 1975). The MK is one of the widely used non-parametric test to detect monotonic trends of hydro-meteorological data series (Dinpashoh et al., 2011; Shadmani et al., 2012; Fensholt and Proud, 2012; Fang et al., 2016, Fathian et al., 2016). Due to the study analyses hydrological data which is not normally distributed, the non-parametric methods are well suited for the detection of monotonic trends. The following equations show how to calculate the test statistic $S$ and the test statistic $Z$ from the MK test:

$$S = \sum_{i=1}^{n-1}\sum_{j=i+1}^{n} sgn(x_j - x_i) \tag{2}$$

$$sgn\,(x_j - x_i) = \begin{cases} -1, & if \quad x_j - x_i < 0 \\ 0, & if \quad x - x_i = 0 \\ +1, & if \quad x - x_i > 0 \end{cases} \tag{3}$$

where $n$ is the length of the dataset, $x_j$ and $x_i$ represent data points in time series $j$ and $i$, respectively ($j > i$). The difference of the magnitude from each pair of values is compared ($x_j$ - $x_i$ ) when ($j > i$) following Eq.(3). $Sgn\,(x_j - x_i)$ is the function sign whose value can be -1, 0 or 1. A positive $S$ value means that the trend is increasing while a negative $S$ value means that the trend is decreasing. The variance of $S$ is estimated by the following equation:

$$Var\,(S) = \frac{\left[n(n-1)(2n+5) - \sum_{j-1}^{g} t_j(t_j-1)(2t_j+5)\right]}{18} \tag{4}$$

where $g$ is the number of the tied groups in the data and $t_j$ is the number of data points in each tied group. The significance of the trend can be found calculating $Z$ value which is acquired as follows:

$$Z = \begin{cases} \frac{S-1}{\sqrt{Var(S)}}, & S > 0 \\ 0, & S = 0 \\ \frac{S+1}{\sqrt{Var(S)}}, & S < 0 \end{cases} \qquad (5)$$

In the present study, the significant level of $Z$ was assessed at 99%, 95% and 90% interval of confidence for $|Z|$ values higher than 2.58, 1.96 and 1.64 respectively. Goodman (2001) shows in Table 1 "Bayesian Interpretation of P-values" the correspondence between Z-score and the p-value. The MK test was assessed in all the NDVI, ET and precipitation spatio-temporal trend analysis.

**4. Results and discussion**

**4.1 Average NDVI, ET and precipitation maps**

NDVI values averaged for the study period (Fig. 3a) range between 0.10 and 0.35 for virtually the entire DP system. The greenest area, with NDVI values from 0.25 to 0.35, is located in the northern part of the catchment and contains dense agricultural developments and areas prone to flooding. The smallest vegetation indices are found in the lowest elevation areas of the central and southern plateau, and also in the mountainous north-west, contrasting with the greenness in the mountainous eastern part of the system.

Average ET losses range from 145 to 550 mm yr$^{-1}$ (Fig. 3b). They appear to be closely related to the terrain elevation, increasing more rapidly with elevation over the Cordillera Real. The maximum ET values are observed in high elevation areas of the Cordillera Real, followed by the northernmost part, where NDVI values are highest. The lowest ET losses occur in the drier south-west, with average annual values under 250 mm yr$^{-1}$.

A north–south gradient is observed in the spatial distribution of precipitation, with annual values ranging from 750 mm yr$^{-1}$ (in the north to 230 mm yr$^{-1}$ in the south (Fig. 3c). Detail analysis of precipitation patterns in the DP system can be found in Garreaud, et al., 2003; Pillco et al., 2007.

[Figure]

**Figure 3.** Spatial distribution of mean NDVI, ET and precipitation for the period 2002-2014 and terrain elevation of the DP system. (a) Mean NDVI; (b) Mean ET; (c) Mean precipitation and (d) Terrain elevation.

**4.2. Spatio-temporal trends in NDVI, ET and precipitation**

Fig. 4 depicts the mean annual NDVI, the annual accumulated ET and annual precipitation integrated over the entire DP system for every year in the study period together, with the change of water volume in Lake Poopó. The ET and precipitation time

series show a mean increasing trend of 4.3 mm yr$^{-1}$ and 5.2 mm yr$^{-1}$ at a significance confidence levels of 90% and 95%, respectively. This result strongly suggests that, from a catchment point of view, the increased ET water losses have been compensated by increases in precipitation during the analysed period. The catchment-wide NDVI exhibits a slower and not-significant increasing trend, mimicking to some degree the rainfall temporal pattern.

Fig. 5 illustrates the spatial distribution of the analysed temporal changes and their per-pixel significance. Fig. 5.a shows that per-pixel vegetation changes across the entire period were small over most of the catchment. Two clusters of NDVI trends can be identified: one corresponds to the areas around Charahuaito (see Fig 5.a) in the southwest part of Aroma province, exhibiting the largest positive NDVI change rate; the second cluster is located at the north end of the system, close to Machaca, and

10   exhibits a negative trend. Both clusters correspond to the areas with the highest mean NDVI (Fig. 3).

The spatial distribution of ET temporal trends reveals highly significant increments over a large central part of the system (Fig. 5.b), while the clearer negative trends are concentrated on the mountainous north-west area. Higher increasing trends were observed in precipitation than in ET over the DP system, especially in the eastern and central parts of the system. No

15   clear spatial correlations between the NDVI and ET trends have been identified, suggesting that the increase in vegetation is not the single main driver for the increase in ET losses.

[Figure]

**Figure 4.** Evolution and temporal trends of annual ET and precipitation and mean NDVI over the entire DP system, and volume change in Lake Poopó.

[Figure]

**Figure 5.** Spatial distribution of NDVI, ET and precipitation trends during the study period. Mean NDVI change (a) was calculated multiplying the trend by the time period. The spatial distribution of ET (b) and precipitation (b) represents annual changes (mm yr$^{-1}$). Figures
5  (d), (e) and (f) show the spatial distribution of the significance level of the trends for the period 2002-2014 in the DP system. Green colour represents pixels without significance in their results. The difference in the pixel size is due to the different spatial resolution of the products.

Some differences were observed in the spatial distribution of increasing ET trends when compared with those reported by Satgé et al. (2017). The latter authors observed statistically significant ET increases in the north of the catchment (see hatched area in Fig. 5b). In this research, the highest increases were observed over a larger, central part of the system. Significant
10  increases were also observed in the south, where the expansion of quinoa crops was reported by Jacobsen et al. (2011). Some

differences between the studies by Satgé et al. (2017) and the one presented here that may explain the different observations include: Satge et al. (2017) used approximately 180 ET products at monthly intervals for the period 2000–2014, while this study used a total of 599 ET products at eight-day intervals, from 2002 to 2014. Furthermore, Satge et al. (2017)'s analysis used the MODIS ET product version 5, at a spatial resolution of 1000 m, whereas this research benefitted from the latest

5    version 6 of the same product, at a spatial resolution of 500 m.

When considering seasonal temporal trends, the wet season presents statistically significant increases in NDVI, ET and precipitation whereas the dry season shows no changes or slight decreases (Table 3, Fig. 6). In the wet season, NDVI increments occur largely over areas of high mean NDVI. However, the limited degree of correlation between NDVI and ET trends reinforces the hypothesis that NDVI is not the single main driver of ET changes in the DP system. The greatest

10    precipitation increases were observed in mountainous areas over the wet season.  The increased rainfall is partially transported via the stream network or aquifer to the lowlands, which could explain the observed ET increase in the central part.

[Figure]

**Figure 6.** NDVI, ET and precipitation spatio-seasonal trends over the DP system for the period 2002-2014: NDVI mean change for (a) wet season and (d) dry season; ET mean annual change (mm yr$^{-1}$) for (b) wet season and (e) dry season and precipitation mean annual change (mm yr$^{-1}$) for (c) wet season and (f) dry season.

**Table 2.** Regional annual ET mean changes in the DP System from 2002-2014. NDVI mean change for the study period 2002-2014. Statistics obtained from the non-parametric Mann Kendall test.

| | STATISTICS OF SEASONAL SPATIO-TEMPORAL ANALYSIS | | | | | |
| --- | --- | --- | --- | --- | --- | --- |
| | Wet period | | | Dry period | | |
| | NDVI | ET | Precipitation | NDVI | ET | Precipitation |
| Sen's slope | 0.049 | 10.7 mm yr$^{-1}$ | 13.26 mm yr$^{-1}$ | 0.002 | -0.7 mm yr$^{-1}$ | 0.26 mm yr$^{-1}$ |
| *p* value | 0.06* | 0.03** | 0.06* | 0.02** | 0.24 | 0.15 |
| % of significant pixels | 51% | 66% | 40% | 70% | 20% | 28% |

5 * Significant at 90% confidence level

\*\*Significant at 95% confidence level

**4.3 Per land cover analysis of ET and NDVI trends**

10 ET trends were positive in most of the scenarios except for the dry season (Fig. 7). The land cover with the highest increases was Central Crops, with trends of 5.7 mm yr$^{-1}$, 16.3 mm yr$^{-1}$ for the entire period and wet season, respectively. Southern Crops and Cordillera Real showed the next highest results after Central Crops, with significant increases during the entire period and wet season, contrasting with no changes during the dry season. Northern Crops experienced the highest decreases during the dry season. The analysis of the entire period revealed almost no changes for Wetlands and Cordillera Occidental. The trend of

15 these land covers changed slightly during the wet months, with increases around 7–8.3 mm yr$^{-1}$. During the dry season, Wetland and Cordillera Occidental showed decreasing trends. Significant increases of ET during the wet season were observed in Bareland and Autochthonous Flora. Table 3 shows the significance confidence level of the land cover trends.

The positive trends observed over natural land covers is probably related to environmental factors, such as warmer temperatures (Hunziker et al., 2018; López-Moreno et al., 2016), increases in glacial meltwater runoff (Vuille et al., 2008;

20 Cook et al., 2016) or changes in the rainfall patterns discussed in previous sections (see Fig. 4). The difference between cropland areas and the rest of the land covers could be related to non-environmental factors. Bolivia is one of the largest exporting countries of quinoa. In the last two decades, the production of quinoa in Bolivia has increased significantly (INE, 2019) responding to a growing international demand. Because the study focuses on static ROIs, the larger increases of ET in cropland areas than in the rest of the land covers may be related to changes in the irrigation management, in order to meet

25 international demand. However, there is a significant general increase of ET affecting all the land covers. The analysis of mean annual water losses per land cover type showed Eastern Cordillera, Cordillera Occidental and Northern Crops as the land covers with the highest ET mean annual values (Fig. 7).

**Table 3.** Result of the MK test.

| LAND COVERS | STATISTICS OF LAND COVER SPATIO-TEMPORAL ANALYSIS | | | | | |
|---|---|---|---|---|---|---|
| | NDVI | | | ET | | |
| | Entire period | Wet period | Dry period | Entire period | Wet period | Dry period |
| **Bareland** | 0.42 | 0.34 | 0.36 | 0.01** | 0.03** | 0.38 |
| **Wetlands** | 0.3 | 0.33 | 0.32 | 0.21 | 0.33 | 0.44 |
| **Cordillera Occ.** | 0.24 | 0.28 | 0.01** | 0.36 | 0.32 | 0.24 |
| **Cordillera Orient.** | 0.03** | 0.06* | 0.00** | 0.03** | 0.26 | 0.36 |
| **Autochthonous Flora** | 0.01** | 0.16 | 0.03** | 0.04** | 0.09* | 0.21 |
| **Northern Crops** | 0.14 | 0.09* | 0.22 | 0.11 | 0.18 | 0.34 |
| **Central Crops** | 0.00** | 0.00** | 0.08* | 0.00** | 0.02** | 0.48 |
| **Southern Crops** | 0.00** | 0.00** | 0.00** | 0.02** | 0.02** | 0.42 |

* Significant at 90% confidence level

** Significant at 95% confidence level

[Figure]

**Figure 7.** ET and NDVI annual land cover trends with a mean of the values corresponding to the entire period and the significance level. (a) Mean ET annual changes (mm yr$^{-1}$) and mean ET (mm yr$^{-1}$) over different land covers and (b) Mean NDVI change and mean NDVI for the same land covers as in ET analysis.

The results of the vegetation land cover analysis revealed interesting and comparable patterns of seasonal vegetation trends. Over the period 2002–2014, land cover NDVI trends were generally positive across the DP system (see Fig. 7b), especially during the wet season. As in the ET analysis, the land cover presenting the highest increases was Central Crops. Greater increases for Northern Crops and Southern Crops than for Cordillera Real were noticed during the wet season. Wetland was the land cover without human activity that presented higher increases during the wet season, followed by Autochthonous flora. Cordillera Occidental and Bareland did not present significant changes in their dynamics for the entire period. The situation could be related to the lack of vegetation in those areas. No remarkable changes were noticed during the dry season in all the land covers. Wetland and Northern Crops were the only land covers showing negative trends during the entire period. The higher increases in cropland areas, especially during the wet season when vegetation is developing and needs a greater amount of water, may be related to some changes in the water management of the area.

**5. Conclusions**

With the aim of exploring catchment changes that may have impacted the Lake Poopó water balance, this study has undertaken a comprehensive spatiotemporal analysis of changes in vegetation, evapotranspiration losses and precipitation over the DP system for the period 2002-2014, using more than 5000 satellite products. Temporal trends of these parameters have also been quantified for the main land cover classes in the area.

The analysis of vegetation changes throughout the entire study period shows slight positive and negative trends distributed over the catchment. Two clusters of consistent trends can be identified: one corresponds to areas around Charahuaito and exhibits the largest positive NDVI change rates; the second cluster, close to Machaca, shows decreasing trends. Both clusters correspond to the highest NDVI areas, indicating that the clearest, large-scale vegetation changes occurred in the most vegetated areas. The spatial distribution of ET temporal trends reveals significant increases over a large central part of the system. The clearest negative trends concentrate on the mountainous northwest area, consistent with the finding of Satgé et al., (2017).

The seasonal analysis of temporal trends revealed striking differences between wet and dry seasons. The trends of NDVI, ET and precipitation in the dry season were either slightly negative or approximately null, whereas the wet season showed increasing trends which are significant over more than 60% of the catchment area. This increasingly uneven seasonal dynamics are presumably related to comparable trends in weather and precipitation patterns and are currently under investigation.

ET losses and their trends have been estimated for the main land covers in the DP catchment. Their values indicate that the land covers with higher water consumption are: Cordillera Real, Cordillera Occidental, Northern Crops, Wetlands and Central

Crops with average values of 500, 410, 410, 370 and 310 mm yr$^{-1}$ respectively. This quantification of water consumption per cover type provides crucial information for the sustainable planning of agriculture exploitation and water resource use in the DP system.

5    Among the analysed land cover classes, only those including crops, i.e. Central and Southern Crops, plus Cordillera Real, have experienced an increase in NDVI and evapotranspiration losses, while natural covers showed either constant or decreasing NDVI trends together with increases in ET. The larger increase in vegetation and ET losses over agricultural regions, strongly suggest that cropping practices exacerbated water losses in these areas.

10   The NDVI and ET values averaged annually over the DP system increased at a mean rate of 0.001 yr$^{-1}$ and 4.3 mm yr$^{-1}$, which yields a mean NDVI and annual ET increments of 0.14 and 56 mm for the 14-year study period. Water inputs into the system due to precipitation increased at a mean rate of 5.2 mm yr$^{-1}$, exceeding the ET rise rate. On the other hand, no neat spatial correlations between NDVI and ET trends have been identified, indicating that the NDVI is not the single main driver for the increase in ET losses. These results indicate that, despite the intensification of agriculture in the DP system between 2002 and 15   2014 increased the ET losses, these cannot be the directly linked to the Lake Poopó shrinkage. These results urge the investigation of trends in other lake water balance terms, such as the flow discharge in the Desaguadero River other tributaries, the evaporation losses and the water exchange between the lake and the aquifer.

20   **Acknowledgements** The authors express their gratitude to USGS, Climate Hazard Group, and Google Earth Engine for the availability of the MODIS and CHIRPS datasets, and to SENAMHI (Bolivia) for the in situ precipitation data.

**References**

25   Abarca-Del-Rio, R., Crétaux, J. F., Berge-Nguyen, M., & Maisongrande, P.: Does Lake Titicaca still control the Lake Poopó system water levels? An investigation using satellite altimetry and MODIS data (2000–2009). Remote sens. lett., 3(8), 707-714, doi: 10.1080/01431161.2012.667884  2012.

AghaKouchak, A., Norouzi, H., Madani, K., Mirchi, A., Azarderakhsh, M., Nazemi, A., Nasrollahi, N., Farahmand, A., 30   Mehran, A. and Hasanzadeh, E.: Aral Sea syndrome desiccates Lake Urmia: call for action. J.Great Lakes Res., 41(1), 307-311, doi: 10.1016/j.jglr.2014.12.007, 2015.

Alsdorf, D.E., Rodríguez, E. and Lettenmaier, D.P.: Measuring surface water from space. Rev. Geophys, 45(2), doi: 10.1029/2006RG000197 2007

Andreucci, D., & Radhuber, I.: Limits to "counter-neoliberal" reform: Mining expansion and the marginalisation of post-extractivist forces in Evo Morales's Bolivia. Geoforum., 84, 280-291, doi: 10.1016/j.geoforum.2015.09.002, 2017.

5    Arsen, A., Crétaux, J. F., Berge-Nguyen, M., & del Rio, R. A.: Remote sensing-derived bathymetry of Lake Poopó. Remote Sens-Basel., 6(1), 407-420, doi: 10.3390/rs6010407, 2014.

Bookhagen, B. and Burbank, D.W.: Toward a complete Himalayan hydrological budget: Spatiotemporal distribution of snowmelt and rainfall and their impact on river discharge. J. Geophys. Res., 115(F3), doi:10.1029/2009JF001426 2010

Buytaert, W. and De Bièvre, B.: Water for cities: The impact of climate change and demographic growth in the tropical Andes. Water Resources Res., 48(8), doi: doi.org/10.1029/2011WR011755 2012.

Campbell, J.B. and Wynne, R.H.:. Introduction to remote sensing. Guilford Press, USA, 2011.

Canedo, C., Pillco Zolá, R., & Berndtsson, R.: Role of Hydrological Studies for the Development of the TDP System. Water-Sui., 8(4), 144. doi: 10.3390/w8040144, 2016.

Canedo Rosso, C., Hochrainer-Stigler, S., Pflug, G., Condori, B., and Berndtsson, R.: Early warning and drought risk
20    assessment for the Bolivian Altiplano agriculture using high resolution satellite imagery data, Nat. Hazards Earth Syst. Sci. Discuss., https://doi.org/10.5194/nhess-2018-133, 2018.

Chen, J., Liao, A., Cao, X., Chen, L., Chen, X., He, C., Han, G., Peng, S., Lu, M. and Zhang, W.: Global land cover mapping at 30 m resolution: A POK-based operational approach. Int. Soc. Photogramme., 103, 7-27, doi:
25    doi.org/10.1016/j.isprsjprs.2014.09.002, 2015.

Chen, X.L., Zhao, H.M., Li, P.X. and Yin, Z.Y.: Remote sensing image-based analysis of the relationship between urban heat island and land use/cover changes. Remote Sens. of Environment, 104(2), pp.133-146, doi: 10.1016/j.rse.2005.11.016, 2006.

30    Cook, S. J., Kougkoulos, I., Edwards, L. A., Dortch, J., and Hoffmann, D.: Glacier change and glacial lake outburst flood risk in the Bolivian Andes. Cryosphere., 10, 2399-2413, https://doi.org/10.5194/tc-10-2399-2016, 2016.

Courault, D., Seguin, B. and Olioso, A.: Review on estimation of evapotranspiration from remote sensing data: From empirical to numerical modeling approaches. Irrig. Drain, 19(3-4), pp.223-249, 2005.

Crétaux, J., Jelinski, W., Calmant, S., Kouraev, A., Vuglinski, V., & Bergé-Nguyen, M. et al.: SOLS: A lake database to monitor in the Near Real Time water level and storage variations from remote sensing data. Adv. Space Res., 47(9), 1497-1507, doi: 10.1016/j.asr.2011.01.004, 2011.

Crosman, E.T. and Horel, J.D.: MODIS-derived surface temperature of the Great Salt Lake. Remote Sens. of Environment, 113(1), pp.73-81, doi: 10.1016/j.rse.2008.08.013,2009.

de Jong, R., de Bruin, S., de Wit, A., Schaepman, M. E., & Dent, D. L.: Analysis of monotonic greening and browning trends from global NDVI time-series. Remote Sens. of Environment, 115(2), 692-702, doi: 10.1016/j.rse.2010.10.011, 2011.

D. Ayaviri Nina and P. Vallejos Mamani.: Cambio climático y seguridad alimentaria, un análisis en la producción agrícola. CienciAgro, 3(1): 59 – 70, 2014.

Didan, K.: MOD13Q1 MODIS/Terra Vegetation Indices 16-Day L3 Global 250m SIN Grid V006 [Data set]. NASA EOSDIS LP DAAC, doi: 10.5067/MODIS/MOD13Q1.006, 2015a.

Didan, K.: MYD13Q1 MODIS/Aqua Vegetation Indices 16-Day L3 Global 250m SIN Grid V006 [Data set]. NASA EOSDIS LP DAAC, doi: 10.5067/MODIS/MOD13Q1.006, 2015b.

Dinpashoh, Y., Jhajharia, D., Fakheri-Fard, A., Singh, V.P. and Kahya, E.: Trends in reference crop evapotranspiration over Iran. J. Hydrol, 399(3-4), pp.422-433, doi: https://doi.org/10.1016/j.jhydrol.2011.01.021 2011.

Duethmann, D. and Blöschl, G.: Why has catchment evaporation increased in the past 40 years? A data-based study in Austria, Hydrol. Earth Syst. Sci., 22, 5143-5158, https://doi.org/10.5194/hess-22-5143-2018, 2018.

Eckert, S., Hüsler, F., Liniger, H., & Hodel, E.: Trend analysis of MODIS NDVI time series for detecting land degradation and regeneration in Mongolia. J. Arid. Environments, 113, 16-28, doi: 10.1016/j.jaridenv.2014.09.001, 2015.

Eimanifar, A. and Mohebbi, F.: Urmia Lake (northwest Iran): a brief review. Saline systems.,3(1), 5, doi: 10.1186/1746-1448-3-5 2005.

Espinoza Villar, J.C., Ronchail, J., Guyot, J.L., Cochonneau, G., Naziano, F., Lavado, W., De Oliveira, E., Pombosa, R. and Vauchel, P.: Spatio-temporal rainfall variability in the Amazon basin countries (Brazil, Peru, Bolivia, Colombia, and

Ecuador). International Journal of Climatology: Int. J. Climatol., 29(11), pp.1574-1594, doi: https://doi.org/10.1002/joc.1791, 2009.

Fang, N. F., Chen, F. X., Zhang, H. Y., Wang, Y. X., and Shi, Z. H.: Effects of cultivation and reforestation on suspended
5   sediment concentrations: a case study in a mountainous catchment in China, Hydrol. Earth Syst. Sci., 20, 13-25, https://doi.org/10.5194/hess-20-13-2016, 2016.

Farr, T.G., Rosen, P.A., Caro, E., Crippen, R., Duren, R., Hensley, S., Kobrick, M., Paller, M., Rodriguez, E., Roth, L. and Seal, D.,: The shuttle radar topography mission. Rev. Geophys., 45(2), doi: 10.1029/2005RG000183, 2007.

10   Fathian, F., Dehghan, Z., Bazrkar, M.H. and Eslamian, S.: Trends in hydrological and climatic variables affected by four variations of the Mann-Kendall approach in Urmia Lake basin, Iran. Hydrolog. Sci. J, 61(5), pp.892-904, doi: https://doi.org/10.1080/02626667.2014.932911, 2016.

Fensholt, R. and Proud, S.R.: Evaluation of earth observation based global long term vegetation trends—Comparing GIMMS and MODIS global NDVI time series. Remote Sens. Environ., 119, 131-147, doi: 10.1016/j.rse.2011.12.015, 2012.

Fernandes, R. and Leblanc, S.G.: Parametric (modified least squares) and non-parametric (Theil–Sen) linear regressions for predicting biophysical parameters in the presence of measurement errors. Remote Sens. Environ., 95(3), 303-316, doi: 10.1016/j.rse.2005.01.005, 2005.

20   Funk, Chris, Pete Peterson, Martin Landsfeld, Diego Pedreros, James Verdin, Shraddhanand Shukla, Gregory Husak, James Rowland, Laura Harrison, Andrew Hoell & Joel Michaelsen. "The climate hazards infrared precipitation with stations—a new environmental record for monitoring extremes". Scientific Data 2, 150066, doi:10.1038/sdata.2015.66, 2015.

Gao, X., Huete, A.R. and Didan, K.: Multisensor comparisons and validation of MODIS vegetation indices at the semiarid
25   Jornada experimental range. IEEE T. Geosci. Remote, 41(10), pp.2368-2381, doi: 10.1109/TGRS.2003.813840, 2003.

Gao, H., Bohn, T. J., Podest, E., McDonald, K. C., & Lettenmaier, D. P.: On the causes of the shrinking of Lake Chad. Environm. Res. Lett. 6(3), 034021, doi: 10.1088/1748-9326/6/3/034021 2011.

30   Garreaud, R., Vuille, M. and Clement, A.C.: The climate of the Altiplano: observed current conditions and mechanisms of past changes. Palaeogeogr. Palaeocl, 194(1-3), pp.5-22, doi: 10.1016/S0031-0182(03)00269-4 ,2003.

Gocic, M. and Trajkovic, S.: Analysis of changes in meteorological variables using Mann-Kendall and Sen's slope estimator statistical tests in Serbia. Global Planet. Change, 100, pp.172-182, doi: https://doi.org/10.1016/j.gloplacha.2012.10.014, 2013.

Goodman, S. N.: Of P-values and Bayes: a modest proposal. Epidemiology, 12(3), 295-297, 2001.

Gorelick, N., Hancher, M., Dixon, M., Ilyushchenko, S., Thau, D. and Moore, R.: Google Earth Engine: Planetary-scale geospatial analysis for everyone. Remote Sens. Environ., 202,18-27, doi: 10.1016/j.rse.2017.06.031, 2017.

Guay, K.C., Beck, P.S., Berner, L.T., Goetz, S.J., Baccini, A. and Buermann, W.: Vegetation productivity patterns at high
10 northern latitudes: a multi-sensor satellite data assessment. Glob. Change Biol, 20(10), pp.3147-3158, doi: https://doi.org/10.1111/gcb.12647 ,2014.

Hammer, U.T.: Saline lake ecosystems of the world (Vol. 59). Springer Science & Business Media, Netherlands, 1986.

15 Herrmann, S.M., Anyamba, A. and Tucker, C.J.: Recent trends in vegetation dynamics in the African Sahel and their relationship to climate. Global Environ. Change, 15(4), pp.394-404, doi: https://doi.org/10.1016/j.gloenvcha.2005.08.004, 2005.

Hoffmann, D. and Requena, C.: Bolivia en un mundo 4 grados más caliente: Escenarios sociopolíticos ante el cambio climático
20 para los años 2030 y 2060 en el altiplano norte. Fundación PIEB, Programa de Investigación Estratégica en Bolivia, Bolivia, 2012.

Hostetler, S.W. Hydrological and Thermal Response of Lakes to Climate: Description and Modeling; Springer: Berlin/Heidelberg, Germany, 1995.
25
Huang, C., Chen, Y., Zhang, S., & Wu, J.: Detecting, extracting, and monitoring surface water from space using optical sensors: A review. Rev. Geophys., 56(2), 333-360, doi: 10.1029/2018RG000598, 2018.

Hunziker, S., Brönnimann, S., Calle, J., Moreno, I., Andrade, M., Ticona, L., Huerta, A. and Lavado-Casimiro, W.: Effects of
30 undetected data quality issues on climatological analyses. Clim. Past, 14(1), pp.1-20, doi:, https://doi.org/10.5194/cp-14-1-2018, 2018.

Instituto Nacional de Estadística. Estadísticas económicas; Instituto Nacional de Estadística, Ed.; The Bolivian National Institute of Statistics: La Paz, Bolivia, 2015. (In Spanish).

Jacobsen, S.: The Situation for Quinoa and Its Production in Southern Bolivia: From Economic Success to Environmental Disaster. J. Agron.Crop.Sci., 197(5), 390-399, doi: 10.1111/j.1439-037x.2011.00475.x, 2011.

Jiang, L., Islam, S., and Carlson, T. N.: Uncertainties in latent heat flux measurement and estimation: implications for using a simplified approach with remote sensing data. Can. J. Remote Sens., 30(5), 769-787, doi: 10.5589/m04-038, 2004.

5   Kalma, J. D., McVicar, T. R., and McCabe, M. F.: Estimating land surface evaporation: A review of methods using remotely sensed surface temperature data. Surv.Geophys., 29(4-5), 421-469, doi: 10.1007/s10712-008-9037-z, 2008.

Karimi, P. and Bastiaanssen, W. G. M.: Spatial evapotranspiration, rainfall and land use data in water accounting – Part 1: Review of the accuracy of the remote sensing data, Hydrol. Earth Syst. Sci., 19, 507-532, https://doi.org/10.5194/hess-19-507-2015, 2015.

Kendall, M. G.: Rank Correlation Methods – Griffin, London, UK, 202 pp., 1975.

Kidd, C.: Satellite rainfall climatology: A review. Int. J. Climat, 21(9), pp.1041-1066, doi: https://doi.org/10.1002/joc.635, 2001.

Li, Z.L., Tang, R., Wan, Z., Bi, Y., Zhou, C., Tang, B., Yan, G. and Zhang, X.: A review of current methodologies for regional
15   evapotranspiration     estimation     from     remotely     sensed     data. Ah.     S.     Sens., 9(5),     pp.3801-3853,     doi: https://doi.org/10.3390/s90503801, 2009.

López-Moreno, J., Morán-Tejeda, E., Vicente-Serrano, S., Bazo, J., Azorin-Molina, C., & Revuelto, J. et al.: Recent temperature variability and change in the Altiplano of Bolivia and Peru. Int. J. Climatol., 36(4), 1773-1796. doi:
20   10.1002/joc.4459, 2015.

Liu, G., Ou, W., Zhang, Y., Wu, T., Zhu, G., Shi, K., & Qin, B.: Validating and mapping surface water temperatures in Lake Taihu: Results from MODIS land surface temperature products. IEEE. J. Sel. Top. Appl. 8(3), 1230-1244, doi: 10.1109/JSTARS.2014.2386333, 2015.

Martí-Cardona, B., Prats, J., & Niclòs, R.:. Enhancing the retrieval of stream surface temperature from Landsat data. Remote Sens. of Environment, 224, 182-191, doi: 10.1016/j.rse.2019.02.007, 2019

Martí-Cardona, B., Pipia, L., Rodríguez Máñez, E., & Hans Sánchez, T.): Teledetección de la Evapotranspiración y Cambio
30   de Cubiertas en la Cuenca del Río Locumba, Perú. In XXVII Congreso Latinoamericano de Hidráulica, IAHR, Lima, Peru, 28-30 Sep. 2016.

Martí-Cardona, B., Dolz-Ripollés, J., & López-Martínez, C.: "Wetland inundation monitoring by the synergistic use of ENVISAT/ASAR imagery and ancillary spatial data", Remote Sens. of Environment, 139(12), 171–184. doi:10.1016/j.rse.2013.07.028, 2013.

5  Martí-Cardona, B., López-Martínez, C., Dolz-Ripollés, J., & Bladé-Castellet, E.: "ASAR polarimetric, multi- incidence angle and multitemporal characterization of Doñana wetlands for flood extent monitoring", Remote Sens. of Environment, 114(11), 2802–2815. doi:10.1016/j.rse.2010.06.015, 2010.

Mann, H.: Nonparametric Tests Against Trend. Econometrica., 13(3), 245. doi: 10.2307/1907187, 1957.

10  McCarthy, J.J., Canziani, O.F., Leary, N.A., Dokken, D.J. and White, K.S. eds.: Climate change 2001: impacts, adaptation, and vulnerability: contribution of Working Group II to the third assessment report of the Intergovernmental Panel on Climate Change (Vol. 2). Cambridge University Press., UK, 2001.

Micklin, P.: Desiccation of the Aral Sea: A Water Management Disaster in the Soviet Union. Sci., 241, 1170-1176. doi: 10.1126/science.241.4870.1170, 1988.

Micklin, P.: The Aral sea disaster. Annu. Rev. Earth Planet. Sci., 35, 47-72. doi: 10.1146/annurec.earth.35.031306.140120, 2007.

Mo, X., Chen, X., Hu, S., Liu, S., and Xia, J.: Attributing regional trends of evapotranspiration and gross primary productivity
20  with remote sensing: a case study in the North China Plain, Hydrol. Earth Syst. Sci., 21, 295-310, https://doi.org/10.5194/hess-21-295-2017, 2017.

Molina Carpio, J., Satgé, F. and Pillco Zola, R..: Los recursos hídricos del sistema TDPS. Available online: http://horizon.documentation.ird.fr/exl-doc/pleins_textes/divers14-09/010062840.pdf, 2014.

Monteith, J. L. et al.: Evaporation and environment, in: Symp. Soc. Exp. Biol, vol. 19, 4, 1965.

Mu, Q., Zhao, M., and Running, S. W.: Improvements to a MODIS global terrestrial evapotranspiration algorithm, Remote Sens. Environ., 115, 1781–1800, doi: 10.1016/j.rse.2011.02.019, 2011.

Neeti, N. and Eastman, J.R.: A contextual mann-kendall approach for the assessment of trend significance in image time series. T. GIS., 15(5), 599-611, doi: doi.org/10.1111/j.1467-9671.2011.01280.x, 2011.

OEA (Organización de los Estados Americanos).: Diagnóstico ambiental del sistema Titicaca-Desaguadero-Poopó-Salar de Coipasa (sistema TDPS) Bolivia-Perú. Departamento Regional y Medio Ambiente. OEA, Washington, D.C. USA, 1996.

Penman, H. L.: Natural evaporation from open water, bare soil and grass, in: Proc. R. Soc. Lond. A, vol. 193, 120–145, The Royal Society, 1948.

Perreault, T.: Dispossession by accumulation? Mining, water and the nature of enclosure on the Bolivian Altiplano. Antipode, 45(5), pp.1050-1069, doi: https://doi.org/10.1111/anti.12005 2013.

Pillco, R., Uvo, C.B., Bengtsson, L. and Villegas, R.: Precipitation variability and regionalization over the Southern Altiplano, Bolivia. *Int. J. Climatol.*, pp.149-164, 2007.

Pillco Zolá, R., Bengtsson, L., Berndtsson, R., Martí-Cardona, B., Satgé, F., Timouk, F., Bonnet, M.-P., Mollericon, L., Gamarra, C., and Pasapera, J.: Modelling Lake Titicaca's daily and monthly evaporation, Hydrol. Earth Syst. Sci., 23, 657-668. doi: 10.5194/hess-23-657-2019, 2019.

Revollo, M.M.: Management issues in the Lake Titicaca and Lake Poopo system: importance of developing a water budget. Lakes & Reservoirs: Research & Management, 6(3), pp.225-229, doi: https://doi.org/10.1046/j.1440-1770.2001.00151.x, 2001.

Rodriguez, E., C.S. Morris, J.E. Belz, E.C. Chapin, J.M. Martin, W. Daffer, S.Hensley.: An assessment of the SRTM topographic products, Technical Report JPL D-31639, Jet Propulsion Laboratory, Pasadena, California, 143 pp. available on http://www2.jpl.nasa.gov/srtm/SRTM_D31639.pdf, 2005.

Rouse, J.W., Haas, R.H., Schell, J.A., Deering, D.W.: Monitoring vegetation systems in the great plains with erts. In: Freden, S.C., Mercanti, E.P., Becker, M.A.: Third Earth Resources Technology Satellite 1 Symposium. NASA, Washington, DC, 1974.

Running, S., Mu, Q., Zhao, M. (2017). MOD16A2 MODIS/Terra Net Evapotranspiration 8-Day L4 Global 500m SIN Grid V006 [Data set]. NASA EOSDIS Land Processes DAAC, doi: 10.5067/MODIS/MOD16A2.006

Satgé, F., Bonnet, M. P., Timouk, F., Calmant, S., Pillco, R., Molina, J., ... & Garnier, J.: Accuracy assessment of SRTM v4 and ASTER GDEM v2 over the Altiplano watershed using ICESat/GLAS data. Int. J. Remote Sens., 36(2), 465-488.doi: 10.1080/01431161.2014.999166, 2015

Satgé, F., Espinoza, R., Zolá, R. P., Roig, H., Timouk, F., Molina, J., Garnier, J., Calmant, F., Seyler, F. and Bonnet, M. P.: Role of climate variability and human activity on Poopó Lake droughts between 1990 and 2015 assessed using remote sensing data. Remote Sens-Basel., 9(3), 218. doi: 10.3390/rs9030218, 2017.

5    Satgé, F., Ruelland, D., Bonnet, M.-P., Molina, J., and Pillco, R.: Consistency of satellite-based precipitation products in space and over time compared with gauge observations and snow- hydrological modelling in the Lake Titicaca region, Hydrol. Earth Syst. Sci., 23, 595-619, https://doi.org/10.5194/hess-23-595-2019, 2019.

Sen, P.K.: Estimates of the regression coefficient based on Kendall's tau. J. Am. Stat. Assoc., 63(324), 1379-1389.doi: 10.2307/2285891, 1968.

Schneider, P. and Hook, S.J.: Space observations of inland water bodies show rapid surface warming since 1985. Geophys. Res. Lett., 37(22), doi: https://doi.org/10.1029/2010GL045059, 2010.

Seth, A., Thibeault, J., Garcia, M. and Valdivia, C.: Making sense of twenty-first-century climate change in the Altiplano: Observed trends and CMIP3 projections. Ann.Assoc Am. Geogr., 100(4), pp.835-847, doi: https://doi.org/10.1080/00045608.2010.500193, 2010.

Shadmani, M., Marofi, S. and Roknian, M.: Trend analysis in reference evapotranspiration using Mann-Kendall and Spearman's Rho tests in arid regions of Iran. Water Resourc. Manag., 26(1), pp.211-224, doi: 10.1007/s11269-011-9913-z, 2012.

20   Thibeault, J.M., Seth, A. and García, M.: Changing climate in the Bolivian Altiplano: CMIP3 projections for temperature and precipitation extremes. J. Geophys. Res-Atmos. 115(D8), doi: https://doi.org/10.1029/2009JD012718, 2010.

Urrutia, R. and Vuille, M.: Climate change projections for the tropical Andes using a regional climate model: Temperature and precipitation simulations for the end of the 21st century. J. Geophys. Res-Atmos., 114(D2), doi: https://doi.org/10.1029/2008JD011021, 2009.

25   UTNIT.: Mapa de cobertura y uso actual de la tierra, Bolivia. COBUSO-2010. Unidad Tecnica Nacional de Informacion de la Tierra. Retreived from http://cdrnbolivia.org/geografia-fisica-nacional.htm, 2011.

Vermote, E.: MOD09A1 MODIS/Terra Surface Reflectance 8-Day L3 Global 500m SIN Grid V006 [Data set]. NASA EOSDIS LP DAAC. doi: 10.5067/MODIS/MOD09A1.006, 2015.

Vuille, M., Francou, B., Wagnon, P., Juen, I., Kaser, G., Mark, B.G. and Bradley, R.S.: Climate change and tropical Andean glaciers: Past, present and future. Earth-sci. Rev., 89(3-4), pp.79-96, doi: https://doi.org/10.1016/j.earscirev.2008.04.002, 2008.

5 Wang, J., Song, C., Reager, J.T., Yao, F., Famiglietti, J.S., Sheng, Y., MacDonald, G.M., Brun, F., Schmied, H.M., Marston, R.A. and Wada, Y.: Recent global decline in endorheic basin water storages. Nat. Geosci. 11(12), p.926, doi: https://doi.org/10.1038/s41561-018-0265-7, 2018.

Wu, Z., Wu, J., Liu, J., He, B., Lei, T. and Wang, Q.: Increasing terrestrial vegetation activity of ecological restoration program 10 in the Beijing–Tianjin Sand Source Region of China. Ecol. Eng., 52,37-50, doi: 10.1016/j.ecoleng.2012.12.040, 2013.

Wurtsbaugh, W. A., Miller, C., Null, S. E., DeRose, R. J., Wilcock, P., Hahnenberger, M.,... & Moore, J.: Decline of the world's saline lakes. Nat. Geosci. 10(11), 816. doi: 10.1038/ngeo3052, 2017.

Xie, Y., Sha, Z. and Yu, M.: Remote sensing imagery in vegetation mapping: a review. J. Plant Ecol., 1(1), pp.9-23, doi: 15 https://doi.org/10.1093/jpe/rtm005, 2008.

Xu, H.: Modification of normalised difference water index (NDWI) to enhance open water features in remotely sensed imagery. Int. J. Remote Sens., 27(14), 3025-3033. doi: 10.1080/01431160600589179, 2006

20 Yue, S., Pilon, P. and Cavadias, G.: Power of the Mann–Kendall and Spearman's rho tests for detecting monotonic trends in hydrological series. J. Hydrol., 259(1-4), pp.254-271, doi: https://doi.org/10.1016/S0022-1694(01)00594-7,2002.

Zola, R.P. and Bengtsson, L.: Long-term and extreme water level variations of the shallow Lake Poopó, Bolivia. Hydrolog. Sci. J., 51(1), 98-114. doi: 10.1623/hysj.51.1.98, 2006.

Zolá, R.P. and Bengtsson, L.: Three methods for determining the area–depth relationship of Lake Poopó, a large shallow lake in Bolivia. Lakes & Reservoirs: Research & Management, 12(4), pp.275-284, doi:https://doi.org/10.1111/j.1440-1770.2007.00344.x, 2007.